# Asymptotically Optimal Fixed-Budget Best Arm Identification with Variance-Dependent Bounds

**Masahiro Kato** [1]   **Masaaki Imaizumi** [1]   **Takuya Ishihara** [2]   **Toru Kitagawa** [3]

## Abstract

We investigate the problem of fixed-budget *best arm identification* (BAI) for minimizing expected simple regret. In an adaptive experiment, a decision maker draws one of multiple treatment arms based on past observations and observes the outcome of the drawn arm. After the experiment, the decision maker recommends the treatment arm with the highest expected outcome. We evaluate the decision based on the *expected simple regret*, which is the difference between the expected outcomes of the best arm and the recommended arm. Due to inherent uncertainty, we evaluate the regret using the *minimax* criterion. First, we derive asymptotic lower bounds for the worst-case expected simple regret, which are characterized by the variances of potential outcomes (leading factor). Based on the lower bounds, we propose the *Two-Stage* (TS)-*Hirano-Imbens-Ridder* (HIR) strategy, which utilizes the HIR estimator (Hirano et al., 2003) in recommending the best arm. Our theoretical analysis shows that the TS-HIR strategy is asymptotically minimax optimal, meaning that the leading factor of its worst-case expected simple regret matches our derived worst-case lower bound. Additionally, we consider extensions of our method, such as the asymptotic optimality for the probability of misidentification. Finally, we validate the proposed method's effectiveness through simulations.

---

[*]Equal contribution [1]Department of Basic Science, the University of Tokyo [2]Graduate School of Economics and Management, Tohoku University [3]Department of Economics, Brown University and Department of Economics, University College London. Correspondence to: Masahiro Kato <mkato.csecon@gmail.com>.

*ICML (International Conference on Machine Learning) Workshop on Counterfactuals in Minds and Machines*, Honolulu, Hawaii, USA. Copyright 2023 by the author(s).

## 1. Introduction

We consider adaptive experiments with multiple treatment arms, such as slot machine arms, different therapies, and distinct unemployment assistance programs. In industrial applications, interactive learning with human feedback for identifying a treatment arm with the highest expected outcome (best treatment arm) is of great interest. This problem is known as the best arm identification (BAI) problem and is a variant of the stochastic multi-armed bandit (MAB) problem (Thompson, 1933; Lai & Robbins, 1985). In this study, we investigate *fixed-budget BAI*, where we aim to minimize the expected simple regret after a fixed number of rounds of an adaptive experiment known as a *budget* (Bubeck et al., 2009; 2011; Audibert et al., 2010). In particular, we focus on the worst-case performance of BAI strategies to reflect the uncertainty of human feedback.

In our setting, a decision maker sequentially draws one of the treatment arms based on past observations in each round of an adaptive experiment and recommends an estimated best treatment arm for future experimental subjects after the experiment. We measure the performance of the decision maker's strategy using the expected simple regret, which is the difference between the maximum expected outcome that could be achieved with complete knowledge of the distributions and the expected outcome of the treatment arm recommended by the strategy. Due to the inherent uncertainty, we evaluate the performance using the worst-case criterion among a given class of bandit models (Bubeck et al., 2011). For nonparametric bandit models with finite variances, we derive worst-case lower bounds. The lower bound's leading factor is characterized by the variances of potential outcomes. We then propose the Two-Stage (TS)-Hirano-Imbens-Ridder (HIR) strategy and show that it is asymptotically minimax optimal, meaning that the leading factor of its worst-case expected simple regret matches that of the lower bound.

In Bubeck et al. (2011), worst-case lower bounds for the expected simple regret are derived for bandit models with bounded supports. These lower bounds only rely on the boundedness of the bandit models and do not depend on any other distributional information. In this study, we derive lower bounds that depend on the variances of the bandit

models, which require the use of distributional information for an optimal strategy. Our lower bounds are based on change-of-measure arguments using the Kullback-Leibler (KL) divergence, which has been used to derive tight lower bounds in existing studies (Kaufmann et al., 2016). Note that the variance appears as the second-order expansion of the KL divergence.

Furthermore, to improve efficiency in this task, we consider a scenario where a decision-maker can draw a treatment arm based on contextual information. We assume that the contextual information can be observed just before drawing a treatment arm and that arm draws can be optimized using this information. Unlike in the conventional setting of contextual bandits, our goal is to identify the treatment arm with the highest unconditional expected outcome rather than the conditional expected outcome. This setting is motivated by the goals of ATE estimation (van der Laan, 2008; Hahn et al., 2011) and BAI with fixed confidence and contextual information (Russac et al., 2021; Kato & Ariu, 2021). Our findings indicate that utilizing contextual information can reduce the expected simple regret, even if our focus is on the unconditional best treatment arm. Note that this setting is a generalization of fixed-budget BAI without contextual information, and our result holds novelty even in the absence of contextual information.

In BAI, target sample allocation ratios play a critical role in determining the proportion of times each treatment arm is drawn. In many BAI settings, target allocation ratios do not have closed-form solutions and require numerical analysis. However, in our analysis, we derive analytical solutions for several specific cases characterized by the standard deviations or variances of the outcomes. When there are only two treatment arms, the target allocation ratio is the ratio of the standard deviation of outcomes. When there are three or more treatment arms without contextual information, the target allocation ratio is the ratio of the variance of outcomes. This contrasts with the findings of Bubeck et al. (2011), which explores the minimax evaluation for bandit models with bounded outcome supports and finds that a strategy of drawing each treatment arm with an equal ratio and recommending a treatment arm with the highest sample average of observed outcomes is minimax optimal. Our results differ from theirs and suggest drawing treatment arms based on the ratio of the standard deviations or variances of outcomes.

Our problem is also closely related to theories of statistical decision-making (Wald, 1949; Manski, 2000; 2002; 2004), limits of experiments (Le Cam, 1972; van der Vaart, 1998), and semiparametric theory (Hahn, 1998), not only to BAI. Among them, semiparametric theory plays an essential role because it allows us to characterize the lower bounds with the semiparametric analogue of the Fisher in-

formation (van der Vaart, 1998). A more detailed survey is presented in Appendix A.

In summary, our contributions include: (i) a lower bound for the worst-case expected simple regret; (ii) an analytical solution for the target allocation ratio, characterized by the variances of outcomes; (iii) the TS-HIR strategy; (iv) an asymptotic minimax optimality of the TS-HIR strategy; These findings contribute to a variety of subjects, including statistical decision theory and causal inference, in addition to BAI.

**Organization.** In Section 2, we formulate our problem. In Section 3, we establish lower bounds for the worst-case expected simple regret and a target allocation ratio. In Section 4, we introduce the TS-HIR strategy. Then, in Section 5, we demonstrate upper bounds of the strategy and its asymptotic minimax optimality. Finally, we discuss related literature in Section 6.

## 2. Problem Setting

We investigate the following setup of fixed-budget BAI. Given a fixed number of rounds $T$, referred to as a budget, in each round $t = 1, 2, \ldots, T$, a decision maker observes contextual information (covariate) $X_t \in \mathcal{X}$ and draws a treatment arm $A_t \in [K] = \{1, 2, \ldots, K\}$. Here, $\mathcal{X} \subset \mathbb{R}^D$ is a space of contextual information[1]. The decision maker then immediately observes an outcome (or reward) $Y_t$ linked to the drawn treatment arm $A_t$. This setting is referred to as the bandit feedback or Neyman-Rubin causal model (Neyman, 1923; Rubin, 1974), in which the outcome in round $t$ is $Y_t = \sum_{a \in [K]} \mathbb{1}[A_t = a] Y_t^a$, where $Y_t^a \in \mathbb{R}$ is a potential independent (random) outcome, and $Y_t^1, Y_t^2, \ldots, Y_t^K$ are conditionally independent given $X_t$. We assume that $X_t$ and $Y_t^a$ are independent and identically distributed (i.i.d.) over $t \in [T] = \{1, 2, \ldots, T\}$. Our goal is to find a treatment arm with the highest expected outcome marginalized over the contextual distribution of $X_t$ with a minimal expected simple regret after observing the outcome in the round $T$.

We define our goal formally. Let $P$ be a joint distribution of $(Y^1, Y^2, \ldots, Y^K, X)$, and $(Y_t^1, Y_t^2, \ldots, Y_t^K, X_t)$ be an i.i.d. copy of $(Y^1, Y^2, \ldots, Y^K, X)$ in round $t$. We refer to the distribution of the potential outcome $(Y^1, Y^2, \ldots, Y^K, X)$ a full-data bandit model (Tsiatis, 2007). For a given full-data bandit model $P$, let $\mathbb{P}_P$ and $\mathbb{E}_P$ denote the probability law and expectation with respect to $P$, respectively. Besides, let $\mu^a(P) = \mathbb{E}_P[Y_t^a]$ denote the expected outcome marginalized over $X$. Let $\mathcal{P}$ denote the set of all $P$. An algorithm in BAI is referred to as a *strategy*, which recommends a treatment arm $\widehat{a}_T \in [K]$ after se-

---

[1] BAI without contextual information corresponds to case where $\mathcal{X}$ is a singleton.

quentially drawing treatment arms in $t = 1, 2, \ldots, T$. With the sigma-algebras $\mathcal{F}_t = \sigma(X_1, A_1, Y_1, \ldots, X_t, A_t, Y_t)$, we define a BAI strategy as a pair $((A_t)_{t \in [T]}, \widehat{a}_T)$, where $(A_t)_{t \in [T]}$ is a sampling rule and $\widehat{a}_T$ is a recommendation rule. The sampling rule draws a treatment arm $A_t \in [K]$ in each round $t$ based on the past observations $\mathcal{F}_{t-1}$ and observed context $X_t$. The recommendation rule returns an estimator $\widehat{a}_T$ of the best treatment arm $\widehat{a}^*(P)$ based on observations up to round $T$. Here, $\widehat{a}_T$ is $\mathcal{F}_T$-measurable. For a bandit model $P \in \mathcal{P}$ and a strategy $\pi \in \Pi$, let us define the simple regret as

$$r_T(\pi)(P) := \max_{a \in [K]} \mu^a(P) - \mu^{\widehat{a}_T}(P).$$

Our goal is to find a strategy that minimizes the worst-case expected simple regret $\sup_{P \in \widetilde{\mathcal{P}}} \mathbb{E}_P[r_T(\pi)(P)]$, where the expectation is taken over the randomness of $\widehat{a}_T \in [K]$ and $\widetilde{\mathcal{P}} \subseteq \mathcal{P}$ is a specific class of bandit models. First, we derive the worst-case lower bounds in Section 3. Then, we propose an algorithm in Section 4 and show the minimax optimality for the expected simple regret in Section 5.

**Notation.** Let $\mu^a(P)(x) := \mathbb{E}_P[Y^a | X = x]$ be the conditional expected outcome given $x \in \mathcal{X}$ for $a \in [K]$. Let $(\sigma^a(P))^2$ be the variance of $Y^a$ under $P \in \mathcal{P}$. Let $\Delta^a(P)$ be a gap $\max_{a \in [K]} \mu^a(P) - \mu^{\widehat{a}_T}(P)$.

# 3. Lower Bounds for the Worst-case Expected Simple Regret

In this section, we derive lower bounds for the worst-case expected simple regret. The expected simple regret can be expressed as

$$\mathbb{E}_P[r_T(\pi)(P)] = \sum_{b \in [K]} \Delta^b(P) \mathbb{P}_P(\widehat{a}_T = b).$$

Here, for each fixed $\mu^a(P), P \in \widetilde{\mathcal{P}}$, independent of $T$, the *probability of misidentification*,

$$\mathbb{P}_P\left(\widehat{a}_T \notin \arg\max_{a \in [K]} \mu^a(P)\right),$$

converges to zero with an exponential rate, while $\Delta^{\widehat{a}_T}(P)$ is the constant. Therefore, we disregard $\Delta^{\widehat{a}_T}(P)$, and the convergence rate of $\mathbb{P}_P\left(\widehat{a}_T \notin \arg\max_{a \in [K]} \mu^a(P)\right)$ dominates the expected simple regret. In this case, to evaluate the convergence rate of $\mathbb{P}_P\left(\widehat{a}_T \notin \arg\max_{a \in [K]} \mu^a(P)\right)$, we utilize large deviation upper bounds. In contrast, if we examine the worst case among $\widetilde{\mathcal{P}}$, which includes a bandit model such that the gaps between the expected outcomes converge to zero with a certain order of the sample size $T$, a bandit model $P$ whose gaps converge to zero at a rate of

$O(1/\sqrt{T})$ dominates the evaluation of the expected simple regret. In general, for the gap $\Delta^b(P)$, the worst-case simple regret is approximately given as

$$\sup_{P \in \widetilde{\mathcal{P}}} \mathbb{E}_P[r_T(\pi)(P)]$$
$$\approx \sup_{P \in \widetilde{\mathcal{P}}} \sum_{b \in [K]} \Delta^b(P) \exp\left(-T\left(\Delta^b(P)\right)^2 / C^b\right),$$

where $(C^b)_{b \in [b]}$ are constants (Bubeck et al., 2011). This is because $\mathbb{P}_P(\widehat{a}_T = b)$ is approximately $\exp\left(-T\left(\Delta^b(P)\right)^2 / C^b\right)$. Then, the maximum is obtained when $\Delta^b(P) = \sqrt{\frac{C^b}{T}}$ for a constant $C^b > 0$, which balances the regret caused by the gap $\Delta^b(P)$ and probability of misidentification $\mathbb{P}_P(\widehat{a}_T = b)$.

## 3.1. Recap: Lower Bounds for Bandit Models with Bounded Supports

Bubeck et al. (2011) shows a (non-asymptotic) lower bound for bandit models with bounded support, where a strategy with the uniform sampling rule is optimal. Let us denote the class of bandit models with bounded outcomes by $P^{[0,1]}$, where each potential outcome $Y_t^a$ is in $[0, 1]$. Then, the authors show that for all $T \geq K \geq 2$, any strategy $\pi \in \Pi$ satisfies $\sup_{P \in P^{[0,1]}} \mathbb{E}_P[r_T(\pi)(P)] \geq \frac{1}{20}\sqrt{\frac{K}{T}}$. For this worst-case lower bound, the authors show that a strategy with the uniform sampling rule and empirical best arm (EBA) recommendation rule is optimal, where we draw $A_t = a$ with probability $1/K$ for all $a \in [K]$ and $t \in [T]$ and recommend a treatment arm with the highest sample average of the observed outcomes, which is referred to as the uniform-EBA strategy. Under the uniform-EBA strategy $\pi^{\text{Uniform-EBM}}$, for $T = K\lfloor T/K \rfloor$, $\sup_{P \in \mathcal{P}^{[0,1]}} \mathbb{E}_P\left[r_T\left(\pi^{\text{Uniform-EBM}}\right)(P)\right] \leq 2\sqrt{\frac{K \log K}{T+K}}$. Thus, the upper bound matches the lower bound if we ignore the $\log K$ and constant terms.

Although the uniform-EBA strategy is nearly optimal, a question remains whether more knowledge about the class of bandit models could be used to derive a tight lower bound and propose another optimal strategy consistent with the novel lower bound. As the worst-case lower bound in Bubeck et al. (2011) is referred to as a distribution-free lower bound, the lower bound does not utilize distributional information, such as variance. In this study, although we still consider worst-case expected simple regret, we develop lower and upper bounds depending on distributional information. Specifically, we characterize the bounds by the variances of potential outcomes. In a worst-case analysis of simple regret, the evaluation of the simple regret is dominated by an instance of a bandit model such that the gaps between the best and suboptimal treatment arms

are $O(1/\sqrt{T})$. Here, recall that tight lower bounds depend on the KL divergence (Lai & Robbins, 1985; Kaufmann et al., 2016). Additionally, the second-order Taylor series expansion of the KL divergence with regard to the gaps can be interpreted as the variance (inversed Fisher information). Therefore, the tight lower bounds employing distributional information in the worst-case expected simple regret should be characterized by the variances (the second-order Taylor series expansion of the KL divergence). In the following sections, we consider worst-case lower bounds characterized by the variances of bandit models.

## 3.2. Local Location-Shift Bandit Models

In this section, we derive asymptotic lower bounds for the worst-case expected simple regret. To derive lower bounds, we often utilize an alternative hypothesis. We consider a bandit model whose expected outcomes are the same among the $K$ treatment arms. We refer it to as the null bandit model.

**Definition 3.1** (Null bandit models). *A bandit model $P \in \mathcal{P}^*$ is called a null bandit model if the expected outcomes are equal: $\mu^1(P) = \mu^2(P) = \cdots = \mu^K(P)$.*

Then, we consider a class of bandit models with unique fixed variances for null bandit models, called local location-shift bandit models. Furthermore, we assume that the moments of potential outcomes are bounded. We define our bandit models as follows.

**Definition 3.2** (Local location-shift bandit models). *A class of bandit models $\mathcal{P}^*$ contains all bandit models that satisfy the following conditions:*
*(i) **Absolute continuity** For all $P, Q \in \mathcal{P}^*$ and $a \in [K]$, let $P^a$ and $Q^a$ be the joint distributions of $(Y^a, X)$ of a treatment arm $a$ under $P$ and $Q$, respectively. The distributions $P^a$ and $Q^a$ are mutually absolutely continuous and have density functions with respect to the Lebesgue measure.*
*(ii) **Invariant contextual information.** For all $P \in \mathcal{P}^*$, the distributions of contextual information $X$ are the same. Let $\mathbb{E}^X$ be an expectation operator over $X$.*
*(iii) **Lipschitz continuity.** For all $a \in [K]$, $x \in \mathcal{X}$, there exists a constant $C > 0$ independent of $T$ such that for all $P, P' \in \mathcal{P}^*$, $\left| (\sigma^a(P)(x))^2 - (\sigma^a(P')(x))^2 \right| < C |\mu^a(P)(x) - \mu^a(P')(x)|$.*
*(iv) **Unique conditional variance.** For all null bandit models $P^\sharp \in \mathcal{P}^*$ such that $\mu^1(P^\sharp) = \mu^2(P^\sharp) = \cdots = \mu^K(P^\sharp)$, the conditional variance is uniquely determined and lower bounded by a positive constant; that is, for all $P^\sharp \in \mathcal{P}^*$, there exists a constant $\sigma^a(x) > C$ such that for all $P^\sharp \in \mathcal{P}^*$, $\left( \sigma^a(P^\sharp)(x) \right)^2 = (\sigma^a(x))^2$ and $\zeta_P(x) = \zeta(x)$, where $C > 0$ is a constant independent of $T$.*
*(v) **Boundedness of the moments.** There exists a constant $C > 0$ independent of $T$ such that for all $P \in \mathcal{P}^*$, $a \in [K]$,*

*and $\gamma \in \mathbb{N}$, $\mathbb{E}_P [|Y^a|^\gamma] < C$.*
*(vi) **Parallel shift.** There exists a constant $B > 0$, independent from $P$, such that for all $P \in \mathcal{P}^*$ and $a, b \in [K]^2$, $\left| (\mu^a(P)(x) - \mu^b(P)(x)) \right| \leq B |\mu^a(P) - \mu^b(P)|$.*

We refer to this class of bandit models as local location-shift bandit models. Our lower bounds are characterized by $\sigma^a(x)$, a conditional variance of null bandit models.

Local location-shift models are a commonly employed assumption in statistical analysis (Lehmann & Casella, 1998). Two key examples are Gaussian and Bernoulli distributions. Under Gaussian distributions with fixed variances, for all $P$, the variances are fixed and only mean parameters shift. Such models are generally called location-shift models. Additionally, we can consider Bernoulli distributions if $\mathcal{P}^*$ includes one instance of $\mu^1(P) = \cdots = \mu^K(P)$ to specify one fixed variance $\sigma^a(x)$. Furthermore, our bandit models are nonparametric within the class and include a more wide range of bandit models, similar to the approach of Barrier et al. (2022).

When contextual information is unavailable, condition (v) can be omitted. Although condition (v) may seem restrictive, its inclusion is not essential for achieving efficiency gains through the utilization of contextual information; that is, the upper bound can be smaller even when this condition is not met. However, it is required in order to derive a matching upper bound for the following lower bounds. Note that the same lower bounds can be derived without condition (v).

## 3.3. Asymptotic Lower Bounds for Local Location-Shift Bandit Models

We consider a restricted class of strategies such that under null bandit models, any strategy in this class recommends one of the arms with an equal probability $(1/K)$.

**Definition 3.3** (Null consistent strategy). *For any $P \in \mathcal{P}^*$ such that $\mu^1(P) = \mu^2(P) = \cdots = \mu^K(P)$, any null consistent strategy satisfies that for any $a, b \in [K]$, $\left| \mathbb{P}_P (\widehat{a}_T = a) - \mathbb{P}_P (\widehat{a}_T = b) \right| \to 0$ as $T \to \infty$. This implies that $\left| \mathbb{P}_P (\widehat{a}_T = a) - 1/K \right| = o(1)$.*

For each $x \in \mathcal{X}$, let us define an average allocation ratio under a bandit model $P \in \mathcal{P}$ and a strategy $\pi \in \Pi$ as $\kappa_{T,P}(a|x) = \frac{1}{T} \sum_{t=1}^{T} \mathbb{E}_P [\mathbb{1}[A_t = a]|X_t = x]$. This quantity represents the average sample allocation to each treatment arm $a$ under a strategy. Let $\mathcal{W}$ be a set of all measurable functions $\kappa_{T,P} : \mathcal{X} \to (0,1)$ such that $\sum_{a \in [K]} \kappa_{T,P}(a|x) = 1$ for each $x \in \mathcal{X}$. Then, we prove the following lower bound. The proof is shown in Appendix D.

**Theorem 3.4.** *Under any null consistent strategy, as $T \to$*

$\infty$,

$$\sup_{P \in \mathcal{P}^*} \sqrt{T} \mathbb{E}_P[r_T(\pi)(P)]$$

$$\geq \frac{1}{12} \inf_{w \in \mathcal{W}} \max_{a \in [K]} \sqrt{\mathbb{E}^X \left[ (\sigma^d(X))^2 / w(a|X) \right]} + o(1).$$

We refer to $w^* = \arg\min_{w \in \mathcal{W}} \max_{a \in [K]} \sqrt{\mathbb{E}^X \left[ \frac{(\sigma^a(X))^2}{w(d|X)} \right]}$ as the target allocation ratio, which is an optimal sample allocation ratio that an optimal strategy aims to achieve. The target sample allocation ratios are used to define a sampling rule in our proposed strategy. In some specific cases, we can obtain analytical solutions for $w^*$. In this study, we consider two cases. As an example, in the following section, we consider a case without contextual information.

**Remark** (Asymptotic Lower Bounds with Allocation Constraints). *When we restrict target allocation ratio, we can restrict the class $\mathcal{W}$. For example, when we need to draw specific treatment arms at a predetermined ratio, we consider a class $\mathcal{W}^\dagger$ such that for all $w \in \mathcal{W}^\dagger$, $\sum_{a \in [K]} w(a|x) = 1$ and $w(b|x) = C$ for all $x \in \mathcal{X}$, some $b \in [K]$, and a constant $C \in (0, 1)$.*

### 3.4. Lower Bounds without Contextual Information

Our result generalizes BAI without contextual information, where $\mathcal{X}$ is a singleton. Let $(\sigma^a)^2$ be the unconditional variance of $Y_t^a$. When there is no contextual information, we can obtain the following lower bound with an analytical solution of the target allocation ratio $w^* \in \mathcal{W}$.

**Corollary 3.5.** *Under any null consistent strategy,*

$$\sup_{P \in \mathcal{P}^*} \sqrt{T} \mathbb{E}_P[r_T(\pi)(P)] \geq \frac{1}{12} \sqrt{\sum_{a \in [K]} (\sigma^a)^2} + o(1)$$

*as $T \to \infty$, where the target allocation ratio is $w^*(a) = \frac{(\sigma^a)^2}{\sum_{b \in [K]} (\sigma^b)^2}$.*

When contextual information is available, for $w(a|x) = \frac{(\sigma^a(x))^2}{\sum_{b \in [K]} (\sigma^b(x))^2}$, the lower bound is $\frac{1}{12} \sqrt{\sum_{a \in [K]} \mathbb{E}^X \left[ (\sigma^a(X))^2 \right]} + o(1)$. It is worth noting that by using the law of total variance, $(\sigma^a)^2 \geq \mathbb{E}^X \left[ (\sigma^a(X))^2 \right]$. Therefore, by utilizing contextual information, we can tighten the lower bounds as $\sqrt{\sum_{a \in [K]} (\sigma^a)^2} \geq \sqrt{\sum_{a \in [K]} \mathbb{E}^X \left[ (\sigma^a(X))^2 \right]} \geq \min_{w \in \mathcal{W}} \max_{d \in [K]} \sqrt{\mathbb{E}_X \left[ \frac{(\sigma^d(X))^2}{w(d|X)} \right]}$. This improvement is efficiency gain by using contextual information.

### 3.5. Refined Lower Bounds for Two-Armed Bandits

For two-armed bandits ($K = 2$), we can refine the lower bound as follows. In this case, we can also obtain an analytical solution of the target allocation ratio even when there is contextual information.

**Theorem 3.6.** *When $K = 2$, under any null consistent strategy,*

$$\sup_{P \in \mathcal{P}^*} \sqrt{T} \mathbb{E}_P[r_T(\pi)(P)]$$

$$\geq \frac{1}{12} \sqrt{\mathbb{E}^X \left[ (\sigma^1(X) + \sigma^2(X))^2 \right]} + o(1)$$

*as $T \to \infty$, where the target allocation ratio is $w^*(a|x) = \sigma^a(x)/(\sigma^1(x) + \sigma^2(x))$ for all $x \in \mathcal{X}$.*

Here, note that $\sqrt{\sum_{a \in [2]} (\sigma^a)^2} \geq \sqrt{(\sigma^1 + \sigma^2)^2}$. This target allocation ratio is the same as that in Kaufmann et al. (2016) for the probability of misidentification minimization. Also see Section 6.1. The proofs is shown in Appendix F.

Note that for $K \geq 3$, we have an analytical solution of the target allocation ratio only when there is no contextual information, and the target allocation ratio is the ratio of the variances. In contrast, for $K = 2$, we can obtain analytical solutions of the target sample allocation ratio even when there is contextual information, and it is the ratio of the (conditional) standard deviation.

## 4. The TS-HIR Strategy

This section introduces our strategy, which is inspired by the adaptive experimental design of Hahn et al. (2011). Our strategy comprises the following sampling and recommendation rules: First, we divide the budget into two parts. In the first stage, we uniformly randomly draw a treatment arm. In the second stage, we draw treatment arms to achieve the target allocation ratio. After the second stage, we recommend the best arm using the HIR estimator. We refer to this strategy as the TS-HIR strategy.

### 4.1. Target allocation ratio

First, we define a target allocation ratio, which is used to determine our sampling rule. As discussed in Section 3, for certain cases, the target allocation ratio has analytical solutions: for $a \in [K]$, $w^*(a|x) = \frac{\sigma^a(x)}{\sigma^1(x) + \sigma^2(x)}$ if $K = 2$, and $w^*(a) = \frac{(\sigma^a)^2}{\sum_{b \in [K]} (\sigma^b)^2}$ if $K \geq 3$ and there is no contextual information. When the variances are unknown, this target allocation ratio is also unknown. Therefore, we estimate it during the adaptive experiment and use the estimator as the probability of drawing a treatment arm.

## 4.2. The TS-HIR Strategy

The TS-HIR strategy consists of the following two stage experiments. For each $a \in [K]$ and $x \in \mathcal{X}$, let $w^{(1)}(a|x)$ and $w^{(2)}(a|x)$ be sample allocation ratios in the first and second stages, respectively. When there is no contextual information, let $w^{(s)}(a|x)$ be $w^{(s)}(a)$ fore each $s \in \{1, 2\}$. We first fix $r \in (0, 1)$ independent from $T$ and $w^{(1)}(a|x)$ such that $w^{(1)}(a|x) > C$ and $\sum_{a \in [K]} w^{(1)}(a|x) = 1$, where $C$ is independent from $T$ [2]. In Stage 1, after drawing each treatment arm $1, 2, \ldots, K$ at each round $1, 2, \ldots, K$, we draw treatment arm $A_t = a$ with probability $w^{(1)}(a)$ until $\lceil rT \rceil$.

After Stage 1, we draw treatment arms with probability $w^{(2)}$, chosen to minimize empirical version of the lower bounds as follows:

$$w^{(2)} = \underset{w \in \mathcal{H}}{\arg\min}$$

$$\begin{cases} \sqrt{\dfrac{(\widehat{\sigma}^1(X_t))^2}{rw^{(1)}(1|X_t) + (1-r)w(1|X_t)} + \dfrac{(\widehat{\sigma}^2(X_t))^2}{rw^{(1)}(2|X_t) + (1-r)w(2|X_t)}} \\ \qquad\qquad\qquad\qquad\qquad\qquad \text{if } K = 2 \\ \max_{d \in [K]} \sqrt{\dfrac{1}{\lceil rT \rceil} \sum_{t=1}^{\lceil rT \rceil} \left\{ \dfrac{(\widehat{\sigma}^d(X_t))^2}{rw^{(1)}(d|X_t) + (1-r)w(d|X_t)} \right\}} \\ \qquad\qquad\qquad\qquad\qquad\qquad \text{if } K \geq 3 \end{cases}, \tag{1}$$

where $\mathcal{H}$ is a class of models of $w^{(2)}$ and can be constrained if there are constraints on the target allocation ratio. Suppose that $r$ is sufficiently small. If $K = 2$ or $K \geq 3$ and there are not contextual information, we have the following analytical solutions for each $a \in [K]$:

$$w^{(2)}(a|x) = \left\{ \frac{\widehat{\sigma}^a(x)}{\widehat{\sigma}^1(x) + \widehat{\sigma}^2(x)} - rw^{(1)}(a|x) \right\} / (1 - r)$$

if $K = 2$, and

$$w^{(2)}(a) = \left\{ \frac{(\widehat{\sigma}^a)^2}{\sum_{b \in [K]} (\widehat{\sigma}^b)^2} - rw^{(1)}(a) \right\} / (1 - r)$$

if $K \geq 3$ and there is no contextual information. Specifically, at each $t \in [T]$, we first draw $\gamma_t$ from a uniform distribution with support $[0, 1]$. Then, we draw $A_t = 1$ if $\gamma_t \leq w^{(2)}(1|X_t)$ and $A_t = a$ for $a \geq 2$ if $\gamma_t \in \left( \sum_{b=1}^{a-1} w^{(2)}(b|X_t), \sum_{b=1}^{a} w^{(2)}(b|X_t) \right]$.

After Stage 2 (after round $T$), for each $a \in [K]$, we estimate $\mu^a$ for each $a \in [K]$ and recommend the maximum. To estimate $\mu^a$, we use the HIR (Hirano-Imbens-Ridder, Hirano et al., 2003; Hahn et al., 2011) estimator defined as

$$\widehat{\mu}_T^{\text{HIR},a} = \frac{1}{T} \sum_{s=1}^{T} \frac{\mathbb{1}[A_s = a]Y_s^a}{\widehat{\pi}(a|X_s)}, \tag{2}$$

[2]Although $r$ is assumed to be independent from $T$, we use $r$ such that $\lceil rT \rceil > K + 1$ in the following sections.

**Parameter:** Positive constants $\gamma \in (0, 1)$.
**Initialization: for** $t = 1$ **do** Draw $A_t = t$. For each $a \in [K]$, set $\widehat{w}_t(a|X_t) = 1/K$. **end for**
**Stage 1:**
**for** $t = K + 1$ to $\lceil rT \rceil$ **do**
    Draw a treatment arm $a$ with probability $w^{(1)}$, irrespective of the contextual information $X_t$.
**end for**
Construct $w^{(2)}$ by using the estimators of the variances as (1).
**Stage 2:**
**for** $t = \lceil rT \rceil + 1$ to $T$ **do**
    Observe $X_t$.
    Draw $\gamma_t$ from a uniform distribution with a support $[0, 1]$.
    $A_t = 1$ if $\gamma_t \leq w^{(2)}(1|X_t)$ and $A_t = a$ for $a \geq 2$ if $\gamma_t \in \left( \sum_{b=1}^{a-1} w^{(2)}(b|X_t), \sum_{b=1}^{a} w^{(2)}(b|X_t) \right]$.
**end for**
Construct $\widehat{\mu}_T^{\text{HIR},a}$ for each $a \in [K]$ as (2).
Recommend $\widehat{a}_T^{\text{HIR}} = \arg\max_{a \in [K]} \widehat{\mu}_T^{\text{HIR},a}$.

---

where $\quad \widehat{\pi}(a|x) = \dfrac{\sum_{s=1}^{T} \mathbb{1}[A_s = a, X_s = x]}{\sum_{s=1}^{T} \mathbb{1}[X_s = x]}.$

Then, we recommend $\widehat{a}_T^{\text{HIR}} \in [K]$ as

$$\widehat{a}_T^{\text{HIR}} = \underset{a \in [K]}{\arg\max} \, \widehat{\mu}_T^{\text{HIR},a}.$$

Note that when there is no contextual information, the HIR estimator is equivalent to the following Sample Average (SA) estimator: $\widehat{\mu}_T^{\text{SA},a} = \frac{1}{\sum_{s=1}^{T} \mathbb{1}[A_s = a]} \sum_{s=1}^{T} \mathbb{1}[A_s = a]Y_s^a$.

## 5. Asymptotic Minimax Optimality of the TS-HIR Strategy

In this section, we derive upper bounds for the worst-case expected simple regret under local location-shift models with our proposed TS-HIR strategy. Let us define $\Delta^{a,b}(P) = \mu^a(P) - \mu^b(P)$, $\Delta^{a,b}(P)(x) = \mu^a(P)(x) - \mu^b(P)(x)$, and $\widehat{\Delta}_T^{\text{HIR},a,b} = \widehat{\mu}_T^{\text{HIR},a} - \widehat{\mu}_T^{\text{HIR},b}$ for all $a, b \in [K]$ and $x \in \mathcal{X}$. The asymptotic normality of the HIR estimator, which is derived by Hahn et al. (2011) using arguments of empirical process, plays an important role in our proof.

**Proposition 5.1** (Asymptotic normality of the HIR estimator. From Theorem 1 of Hahn et al. (2011))**.** *Assume that $w^*$ smoothly depends on $\sigma^a(x)$. Then,*

$$\sqrt{T} \left( \widehat{\Delta}_T^{\text{HIR},a,b} - \Delta^{a,b}(P) \right) \xrightarrow{\text{d}} \mathcal{N}\left(0, V^{a,b}(P)\right),$$

*where*

$$V^{a,b}(P) = \mathbb{E}^X \left[ \frac{(\sigma^a(P)(X))^2}{w^*(a|X)} + \frac{(\sigma^b(P)(X))^2}{w^*(b|X)} + \left( \Delta^{a,b}(P)(X) - \Delta^{a,b}(P) \right)^2 \right].$$

Then, to prove the upper bound for the expected simple regret, we first derive the that for the probability of misidentification by the Chernoff bound an Proposition 5.1. After that, we derive the worst-case upper bound as well as Bubeck et al. (2011).

### 5.1. Upper Bounds for the Probability of Misidentification

First, we show the upper bound for the probability of misidentification, which is needed for deriving the upper bound for the expected simple regret. Let $V^a(P) = V^{a^*(P),a}(P)$ for each $a \in [K]$.

**Theorem 5.2** (Upper bound for the probability of misidentification). *For each $P \in \mathcal{P}^*$, $a \in [K]\backslash\{a^*(P)\}$, and any $\varepsilon > 0$, there exists $T_0 > 0$ such that for all $T > T_0$,*

$$\mathbb{P}_P \left( \widehat{\mu}_T^{\mathrm{HIR},a^*(P)} \leq \widehat{\mu}_T^{\mathrm{HIR},a} \right)$$
$$\leq \exp \left( -\frac{T(\Delta^a(P))^2}{2V^a(P)} + \left\{ \frac{\sqrt{T}\Delta^a(P)}{\sqrt{V^a(P)}} + \frac{T(\Delta^a(P))^2}{2V^a(P)} \right\} \varepsilon \right).$$

The asymptotic optimality for the probability of misidetification is also intriguing problem. There are several arguments for the lower bounds (Kaufmann, 2020). In Section 6.1, we show that the upper bound in Theorem 5.2 is asymptotically optimal when $K = 2$ and $\Delta^{1,2}(P) \to 0$. Also see Section A. The proof is shown in Appendix G, which utilizes the asymptotic normality of $\widehat{\Delta}_T^{\mathrm{HIR},a,b}$.

### 5.2. Upper Bounds for the Expected Simple Regret

Theorem 5.2 yields the following theorem on the upper bound for the expected simple regret.

**Theorem 5.3** (Upper bound for the expected simple regret). *For the TS-HIR strategy $\pi^{\mathrm{HIR}}$ and any $\varepsilon > 0$, there exists $T_0 > 0$ such that for all $T > T_0$,*

$$\mathbb{E}_P \left[ r_T \left( \pi^{\mathrm{HIR}} \right) (P) \right] \leq \sum_{a \in [K]} \Delta^a(P)$$
$$exp \left( -\frac{T(\Delta^a(P))^2}{2V^a(P)} + \left\{ \frac{\sqrt{T}\Delta^a(P)}{\sqrt{V^a(P)}} + \frac{T(\Delta^a(P))^2}{2V^a(P)} \right\} \varepsilon \right).$$

### 5.3. Asymptotic Minimax Optimality

We first present an asymptotic worst-case upper bound of our proposed TS-HIR strategy. Suppose that there is no contextual information or $\mathcal{X}$ is discrete. We put the following assumption on the convergence rate of estimators of the nuisance parameters. We show the following theorem on the worst-case upper bound and the asymptotic minimax optimality. The proof is shown in Appendix H.

**Theorem 5.4** (Asymptotic minimax optimality of the TS-HIR Strategy). *For the TS-HIR strategy $\pi^{\mathrm{HIR}}$, when $K \geq 3$, as $T \to \infty$,*

$$\sup_{P \in \mathcal{P}^*} \sqrt{T}\mathbb{E}_P \left[ r_T \left( \pi^{\mathrm{HIR}} \right) (P) \right]$$
$$\leq \max_{a,b \in [K]:a \neq b} \sqrt{\log K \mathbb{E}^X \left[ \frac{(\sigma^a(X))^2}{w^*(a|X)} + \frac{(\sigma^b(X))^2}{w^*(b|X)} \right]} + o(1)$$

Because we can obtain an analytical solution of $w^*$ when $K = 2$ with and without contextual information and $K \geq 3$ without contextual information, we have the following upper bound in those case:

$$\sup_{P \in \mathcal{P}^*} \sqrt{T}\mathbb{E}_P \left[ r_T \left( \pi^{\mathrm{HIR}} \right) (P) \right]$$
$$\leq \frac{1}{2} \sqrt{\mathbb{E}^X \left[ \sigma^1(X) + \sigma^2(X) \right]^2} + o(1)$$

when $K = 2$, and

$$\sup_{P \in \mathcal{P}^*} \sqrt{T}\mathbb{E}_P \left[ r_T \left( \pi^{\mathrm{HIR}} \right) (P) \right]$$
$$\leq 2 \log K \left( \sqrt{\sum_{a \in [K]} \mathbb{E}^X \left[ (\sigma^a(X))^2 \right]} \right) + o(1)$$

when $K \geq 3$, and there is no contextual information. Thus, when $T \to \infty$, the upepr and lower bound match.

## 6. Discussion and Related Work

### 6.1. On the Asymptotic Optimality for the Probability of Misidentification

Although we do not discuss the lower bounds for the probability of misidentification, our upper bound for the probability of misidentification in Theorem 5.2 aligns with the lower bound established by Kaufmann et al. (2016) for two-armed Gaussian bandits. Let $\mathcal{P}^G \subset \mathcal{P}^*$ denote two-armed Gaussian bandits defined in Kaufmann et al. (2016) with fixed variance $(\sigma^1)^2 . (\sigma^2)^2 > 0$. Their work demonstrates that $\limsup_{T \to \infty} -\frac{1}{T} \log \mathbb{P}_P(\widehat{a}_T^{\mathrm{HIR}} \neq a^*(P)) \leq (\Delta^{1,2}(P))^2/2(\sigma^1 + \sigma^2)^2$ when $P \in \mathcal{P}^G$. Our upper bound coincides with the lower bound when $\Delta^{1,2}(P) \to 0$.

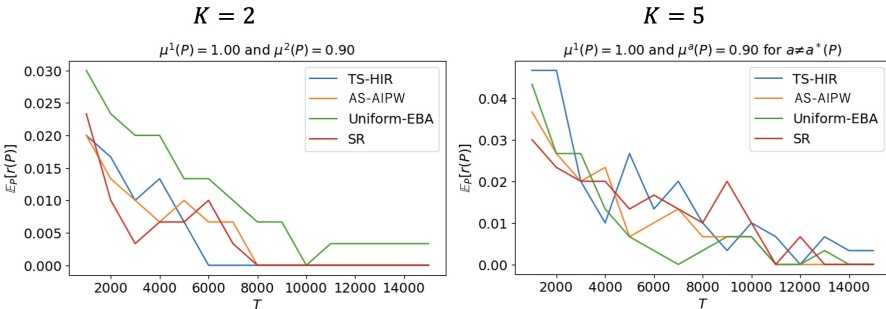

Figure 1: Experimental results. The $y$-axis and $x$-axis denote the expected simple regret $\mathbb{E}_P[r_T(\pi)(P)]$ under each strategy and $T$, respectively.

**Theorem 6.1** (Asymptotic optimally for the probability of misidentification). *For each $P \in \mathcal{P}^{\mathrm{G}}$,*

$$\liminf_{T \to \infty} -\frac{1}{T} \log \mathbb{P}_P \left( \widehat{a}_T^{\mathrm{HIR}} \neq a^*(P) \right)$$

$$\geq \frac{\left( \Delta^{1,2}(P) \right)^2}{2(\sigma^1 + \sigma^2)^2} - o\left( \left( \Delta^{1,2}(P) \right)^2 \right)$$

From Theorem 5.2, $-\frac{1}{T} \log \mathbb{P}_P \left( \widehat{a}_T^{\mathrm{HIR}} \neq a^*(P) \right) \geq \frac{(\Delta^{1,2}(P))^2}{2V^{1,2}(P)} - \left\{ \frac{\Delta^{1,2}(P)}{\sqrt{TV^{1,2}(P)}} + \frac{(\Delta^{1,2}(P))^2}{2V^{1,2}(P)} \right\} \varepsilon.$

When for two-armed Gaussian bandits, the upper bound matches the lower bound in Kaufmann et al. (2016). We do not discuss the details of the lower bounds for probability of misidentification for more general cases because there are technical difficulties and many open problems. Also see Kaufmann (2020), Ariu et al. (2021), Qin (2022), and Degenne (2023) for the details.

### 6.2. The AS-AIPW Strategy

When the contextual information is continuous, we cannot apply the TS-HIR strategy. In this case, we use the AS-AIPW strategy, which consists of the *Adaptive Sampling* (AS) and *Augmented Inverse Probability Weighting* (AIPW) rules. Under the AS rule, at each $t$, we estimate $w^*$ and draw treatment arm $a$ with probability $\widehat{w}_t(a|X_t)$, where $\widehat{w}_t$ is an estimates of $w^*$. Then, at each $T$, we estimate $\mu^a(P)$ by using the AIPW estimator defined in Appendix I and recommend a treatment arm with the highest estimate value as the best arm. The AIPW estimator debiases the sample selection bias resulting from arm draws based on contextual information. The details is shown in Appendix I.

### 7. Experiments

We compare our TS-HIR and AS-AIPW strategies with the Uniform-EBA (Uniform, Bubeck et al., 2011), and Successive Rejection (SR, Audibert et al., 2010) (Gabillon et al., 2012).

In this section, we investigate two setups with $K = 2, 5$ without contextual information. The best treatment arm is arm 1 and $\mu^1(P) = 1$. The expected outcomes of suboptimal treatment arms are equivalent and denoted by $\widetilde{\mu} = \mu^2(P) = \cdots = \mu^K(P)$. We use $\widetilde{\mu} = 0.90$. When $K = 2$, we fix the variances as $((\sigma^1)^2, (\sigma^2)^2) = (5, 1)$; when $K = 5$, we fix the variances as $((\sigma^1)^2, (\sigma^2)^2, (\sigma^3)^2, (\sigma^4)^2, (\sigma^5)^2) = (5, 1, 2, 3, 4)$. We continue the experiments until $T = 5,000$ when $\widetilde{\mu} = 0.80$ and $T = 15,000$ when $\widetilde{\mu} = 0.90$. We conduct 100 independent trials for each setting. At each $t \in \{1000, 2000, 3000, \cdots, 14000, 15000\}$, we plot the empirical simple regret in Figure 1. Additional results with other settings, including contextual information, are presented in Appendix M.

From Figure 1 and Appendix M, we can observe that our proposed strategies perform well when $K = 2$. When there exists a contextual information, our methods tends to show better performances than the others. Although our strategies show preferable performances in many settings, other strategies also perform well. We conjecture that our strategies exhibit superiority against other methods when $K$ is small (mismatching term in the upper bound), the gap between the best and suboptimal arms is small, and the variances significantly vary across arms. As the superiority depends on the situation, we recommend a practitioner to use several strategies in a hybrid way.

### 8. Conclusion

We conducted an asymptotic worst-case analysis of the simple regret in fixed-budget BAI with contextual information. Initially, we obtained lower bounds for local location-shift bandit models, where the variances of potential outcomes characterize the asymptotic lower bounds as a second-order approximation of the KL divergence. Based on these lower bounds, we derived target allocation ratios, which were used to define a sampling rule in the TS-HIR strategy. Finally, we demonstrated that the TS-HIR strategy achieves minimax optimality for the expected simple

regret.

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

# A. Related Work

In this section, we review the literature on the related field.

For the expected simple regret, Bubeck et al. (2011) shows that the Uniform-Empirical Best Arm (EBA) strategy is minimax optimal for bandit models with bounded supports. Kock et al. (2020) extends the results to cases where parameters of interest are functionals of the distribution, and finds that optimal sampling rules are not uniform sampling. Adusumilli (2022; 2021) considers a different minimax evaluation of bandit strategies for both regret minimization and BAI problems, which is based on a formulation using a diffusion process, as proposed by Wager & Xu (2021).

## A.1. Literature of BAI

The MAB problem embodies an abstraction of the sequential decision-making process (Thompson, 1933; Robbins, 1952; Lai & Robbins, 1985). BAI constitutes a paradigm of this problem (Even-Dar et al., 2006; Audibert et al., 2010; Bubeck et al., 2011), with its variants dating back to the 1950s in the context of sequential testing, ranking, and selection problems (Bechhofer et al., 1968). Additionally, ordinal optimization has been extensively studied in the field of operations research, with a modern formulation established in the early 2000s (Chen et al., 2000; Glynn & Juneja, 2004). Most of these studies have focused on determining optimal strategies under the assumption of known target allocation ratios. Within the machine learning community, the problem has been reframed as the BAI problem, with a particular emphasis on performance evaluation under unknown target allocation ratios. (Audibert et al., 2010; Bubeck et al., 2011).

For fixed-budget BAI has been extensively studied, with notable contributions fromBubeck et al. (2011) who demonstrates minimax optimal strategies for the expected simple regret in a non-asymptotic setting, and Audibert et al. (2010) who proposes the UCB-E and Successive Rejects (SR) strategies. Kock et al. (2020) generalizes the results of Bubeck et al. (2011) to cases where parameters of interest are functionals of the distribution and find that target allocation ratios are not uniform, in contrast to the results of Bubeck et al. (2011).

Kaufmann et al. (2016) also contributes to this field by deriving distribution-dependent lower bounds for BAI with fixed confidence and a fixed budget, using the change-of-measure arguments and building upon the work of (Lai & Robbins, 1985). Garivier & Kaufmann (2016) proposes an optimal strategy for BAI with fixed confidence, however, in the fixed-budget setting, there is currently a lack of strategies whose upper bound matches the lower bound established by Kaufmann et al. (2016). Carpentier & Locatelli (2016) examines the lower bound and show the optimality of the method proposed by Audibert et al. (2010) in terms of leading factors in the exponent.

The lower bound of Carpentier & Locatelli (2016) is based on a minimax evaluation of the probability of misidentification under a large gap. Yang & Tan (2022) proposes minimax optimal linear BAI with a fixed budget by extending the result of Carpentier & Locatelli (2016).

Kato & Imaizumi (2023) summarizes our result and discusses the asymptotic optimality of the AS-AIPW strategy for the probabilty of misidentification in more details.

In addition to minimax evaluation, Komiyama et al. (2021) develops an optimal strategy whose upper bound for a simple Bayesian regret lower bound matches their derived lower bound. Atsidakou et al. (2023) proposes a Bayes optimal strategy for minimizing the probability of misidentification, which shows a surprising result that $1/\sqrt{T}$-factor dominates the evaluation.

In Russo (2016), Qin et al. (2017), and Shang et al. (2020) respectively, the authors propose Bayesian BAI strategies that are optimal in terms of posterior convergence rate. However, it has been shown by Kasy & Sautmann (2021) and Ariu et al. (2021) that such optimality does not extend to asymptotic optimality for the probability of misidentification in fixed-budget BAI.

Adusumilli (2022; 2021) present an alternative minimax evaluation of bandit strategies for both regret minimization and BAI, which is based on a formulation utilizing a diffusion process proposed by Wager & Xu (2021). Furthermore, Armstrong (2022) extends the results of Hirano & Porter (2009) to a setting of adaptive experiments. The results of Adusumilli (2022; 2021) and Armstrong (2022) employ arguments on local asymptotic normality (Le Cam, 1960; 1972; 1986; van der Vaart, 1991; 1998), where the class of alternative hypotheses comprises of "local models," in which parameters of interest converge to true parameters at a rate of $1/\sqrt{T}$.

Tekin & van der Schaar (2015), Guan & Jiang (2018), and Deshmukh et al. (2018) consider BAI with contextual informa-

tion, but their analysis and setting are different from those employed in this study.

## A.2. Efficient Average Treatment Effect Estimation

Efficient estimation of ATE via adaptive experiments constitutes another area of related literature. van der Laan (2008) and Hahn et al. (2011) propose experimental design methods for more efficient estimation of ATE by utilizing covariate information in treatment assignment. Despite the marginalization of covariates, their methods are able to reduce the asymptotic variance of estimators. Karlan & Wood (2014) applies the method of Hahn et al. (2011) to examine the response of donors to new information regarding the effectiveness of a charity. Subsequently, Tabord-Meehan (2022) and Kato et al. (2020) have sought to improve upon these studies, and more recently, Gupta et al. (2021) have proposed the use of instrumental variables in this context.

## A.3. Other Related Work

Our arguments are inspired by limit-of-experiments framework (Le Cam, 1986; van der Vaart, 1998; Hirano & Porter, 2009). Within this framework, we can approximate the statistical experiment by a Gaussian distribution using the CLT. Hirano & Porter (2009) relates the asymptotic optimality of statistical decision rules (Manski, 2000; 2002; 2004; Dehejia, 2005) to the framework.

The AIPW estimator has been extensively used in the fields of causal inference and semiparametric inference (Tsiatis, 2007; Bang & Robins, 2005; Chernozhukov et al., 2018). More recently, the estimator has also been utilized in other MAB problems, as seen in Kim et al. (2021) and Ito et al. (2022).

# B. Preliminaries

Let $W_i$ be a random variable with probability measure $P$. Let $\mathcal{F}_n = \sigma(W_1, W_2, \ldots, W_n)$.

**Definition B.1.** *[Uniform integrability, Hamilton (1994), p. 191] Let $W_t \in \mathbb{R}$ be a random variable with a probability measure $P$. A sequence $\{W_t\}$ is said to be uniformly integrable if for every $\epsilon > 0$ there exists a number $c > 0$ such that*

$$\mathbb{E}_P[|W_t| \cdot \mathbb{1}[|W_t| \geq c]] < \epsilon$$

*for all $t$.*

**Proposition B.2** (Sufficient conditions for uniform integrability; Proposition 7.7, p. 191. Hamilton (1994)). *Let $W_t, Z_t \in \mathbb{R}$ be random variables. Let $P$ be a probability measure of $Z_t$. (a) Suppose there exist $r > 1$ and $M < \infty$ such that $\mathbb{E}_P[|W_t|^r] < M$ for all $t$. Then $\{W_t\}$ is uniformly integrable. (b) Suppose there exist $r > 1$ and $M < \infty$ such that $\mathbb{E}_P[|Z_t|^r] < M$ for all $t$. If $W_t = \sum_{j=-\infty}^{\infty} h_j Z_{t-j}$ with $\sum_{j=-\infty}^{\infty} |h_j| < \infty$, then $\{W_t\}$ is uniformly integrable.*

**Proposition B.3** ($L^r$ convergence theorem, p 165, Loeve (1977)). *Let $W_t$ be a random variable with probability measure $P$ and $w$ be a constant. Let $0 < r < \infty$, suppose that $\mathbb{E}_P[|W_t|^r] < \infty$ for all $t$ and that $W_t \xrightarrow{\mathrm{P}} z$ as $n \to \infty$. The following are equivalent:*

*(i) $W_t \to w$ in $L^r$ as $t \to \infty$;*

*(ii) $\mathbb{E}_P[|W_t|^r] \to \mathbb{E}_P[|w|^r] < \infty$ as $t \to \infty$;*

*(iii) $\{|W_t|^r, t \geq 1\}$ is uniformly integrable.*

**Definition B.4.** *For $\mathcal{F}_t$ equal to the $\sigma$-field generated by $\xi_1, \ldots, \xi_t$, $\{W_t, \mathcal{F}_t, t \geq 1\}_{t=1}^{\infty}$ is a martingale if for all $t \geq 1$, we have*

$$\mathbb{E}[W_{t+1}|\mathcal{F}_t] = W_t.$$

*If $\mathbb{E}[W_{t+1}|\mathcal{F}_t] = 0$, $\{W_t, \mathcal{F}_t, t \geq 1\}_{t=1}^{\infty}$ is a martingale difference sequence.*

**Proposition B.5** (Weak Law of Large Numbers for Martingale, Hall et al. (1980)). *Let $\{S_t = \sum_{s=1}^{t} W_s, \mathcal{F}_t, t \geq 1\}$ be a martingale and $\{b_t\}$ a sequence of positive constants with $b_t \to \infty$ as $t \to \infty$. Then, writing $W_{ts} = W_s \mathbb{1}[|W_s| \leq b_t]$, $1 \leq s \leq t$, we have that $b_t^{-1} S_t \xrightarrow{\mathrm{P}} 0$ as $t \to \infty$ if*

**(i)** $\sum_{s=1}^{t} P(|W_s| > b_t) \to 0$;

**(ii)** $b_t^{-1} \sum_{s=1}^{t} \mathbb{E}[W_{ts}|\mathcal{F}_{s-1}] \xrightarrow{\mathrm{P}} 0$, and;

**(iii)** $b_t^{-2} \sum_{s=1}^{t} \left\{ \mathbb{E}[W_{ts}^2] - \mathbb{E}\left[\mathbb{E}\left[W_{ts}|\mathcal{F}_{s-1}\right]\right]^2 \right\} \to 0.$

**Proposition B.6** (Central Limit Theorem for a Martingale Difference Sequence; from Proposition 7.9, p. 194, Hamilton (1994); also see White (1984)). *Let* $\{(S_t = \sum_{s=1}^{t} W_t, \mathcal{F}_t)\}_{t=1}^{\infty}$ *be a martingale with* $\mathcal{F}_t$ *equal to the $\sigma$-field generated by* $W_1, \dots, W_t$. *Suppose that*

**(a)** $\mathbb{E}[W_t^2] = \sigma_t^2$, *a positive value with* $(1/T) \sum_{t=1}^{T} \sigma_t^2 \to \sigma^2$, *a positive value;*

**(b)** $\mathbb{E}[|W_t|^r] < \infty$ *for some $r > 2$;*

**(c)** $(1/T) \sum_{t=1}^{T} W_t^2 \xrightarrow{p} \sigma^2.$

*Then* $S_T \xrightarrow{d} \mathcal{N}(\mathbf{0}, \sigma^2)$.

**Proposition B.7** (Rate of convergence in the CLT; from Theorem 3.8, p 88, Hall et al. (1980)). *Let* $\{(S_t = \sum_{s=1}^{t} W_t, \mathcal{F}_t)\}_{t=1}^{\infty}$ *be a martingale with $\mathcal{F}_t$ equal to the $\sigma$-field generated by $W_1, \dots, W_t$. Let*

$$V_t^2 = \sum_{s=1}^{t} \mathbb{E}[W_s^2|\mathcal{F}_{t-1}] \qquad 1 \le t \le T.$$

*Suppose that for some $\alpha > 0$ and constants $M$, $C$ and $D$,*

$$\max_{t \le T} \mathbb{E}_P[\exp(|\sqrt{t}W_t|^{\alpha})] < M,$$

*and*

$$\mathbb{P}_P\left(|V_t^2 - 1| > D/\sqrt{t}(\log t)^{2+2/\alpha}\right) \le Ct^{-1/4}(\log t)^{1+1/\alpha}.$$

*Then, for $T \ge 2$,*

$$\sup_{-\infty < x < \infty} \left|\mathbb{P}_P(S_T \le x) - \Phi(x)\right| \le AT^{-1/4}(\log T)^{1+1/\alpha},$$

*where the constant $A$ depends only on $\alpha$, $M$, $C$, and $D$.*

**Proposition B.8** (Convergence in distribution and in moments. Lemma 2.1 of Hayashi (2000)). *Let $\alpha_{s,n}$ be the $s$-th moment of $z_n$, and $\lim_{n\to\infty} \alpha_{s,n} = \alpha_s$, where $\alpha_s$ is finite. Suppose that for some $\delta > 0$, $\mathbb{E}\left[|z_n|^{s+\delta}\right] < M < \infty$ for all $n$ and a constant $M > 0$ independent of $n$. If $z_n \xrightarrow{d} z$, then $\alpha_s$ is the $s$-th moment of $z$.*

## C. Non-asymptotic Lower Bounds for Bandit Models with Bounded Supports

First, we introduce an existing lower bound for bounded bandit models. Let us denote the class of bandit models with bounded outcomes by $P^{[0,1]}$, where each potential outcome $Y_t^a$ is in $[0,1]$. Then, Bubeck et al. (2011) proposes the following lower bound, which holds for $P^{[0,1]}$.

**Proposition C.1.** *For all $T \ge K \ge 2$, any strategy $\pi \in \Pi$ satisfies $\sup_{P\in P^{[0,1]}} \mathbb{E}_P[r_T(\pi)(P)] \ge \frac{1}{20}\sqrt{\frac{K}{T}}$.*

This lower bound only requires that the support of the bandit models in $P^{[0,1]}$ is bounded.

For this non-asymptotic lower bound, Bubeck et al. (2011) shows that a strategy with the uniform sampling rule and empirical best arm (EBA) recommendation rule is optimal, where we draw $A_t = a$ with probability $1/K$ for all $a \in [K]$ and $t \in [T]$ and recommend a treatment arm with the highest sample average of the observed outcomes. We call this strategy the uniform-EBA strategy.

**Proposition C.2** (Non-asymptotic optimality of the uniform-EBA strategy). *Under the uniform-EBA strategy* $\pi^{\text{Uniform-EBM}}$, *for* $T = K\lfloor T/K \rfloor$, $\sup_{P \in \mathcal{P}[0,1]} \mathbb{E}_P \left[ r_T \left( \pi^{\text{Uniform-EBM}} \right) \right] \leq 2\sqrt{\frac{K \log K}{T+K}}$.

Thus, the upper bound matches the distribution-free lower bound if we ignore the $\log K$ and constant terms.

Although the uniform-EBA strategy is nearly optimal, a question remains whether more knowledge about the class of bandit models could be used to derive a tight lower bound and propose another optimal strategy consistent with the novel lower bound. To answer this question, we consider asymptotic evaluation and derive a tight lower bound for bandit models with a fixed-variance.

# D. Proof of the Asymptotic Lower Bound for Multi-Armed Bandits (Theorem 3.4)

In this section, we provide the proof of Theorems 3.4. Our lower bound derivation is based on arguments of a change-of-measure and semiparametric efficiency. The change-of-measure arguments have been extensively used in the bandit literature (Lai & Robbins, 1985). The semiparametric efficiency is employed for deriving the lower bound of the KL divergence with a two-order Taylor expansion. Our proof is inspired by van der Vaart (1998), and Murphy & van der Vaart (1997).

We prove the asymptotic lower bound through the following steps. We first introduce lower bounds for the probability of misidentification, shown by Kaufmann et al. (2016). In Appendix D.1, we define observed-data bandit models, which are distributions of observations that differ from full-data bandit models $P \in \mathcal{P}^*$. In Appendix D.2, we define submodels of the observed-data bandit models, which parametrize nonparametric bandit models by using parameters of gaps of the expected outcomes of the best and suboptimal treatment arms. These parameters serve as technical devices for the proof. In Appendix D.3, we then decompose the expected simple regret into the gap parameters and the probability of misidentification, and apply the lower bound of Kaufmann et al. (2016) for the probability of misidentification. The lower bound is characterized by the KL divergence of the observed-data bandit models, which we expand around the gap parameters in Appendix D.4. We then derive the semiparametric efficient influence function, which bounds the second term of the series expansion of the KL divergence in Appendix D.5, and compute the worst-case bandit model in Appendix D.6. Finally, we derive the target allocation from the lower bound in Appendix D.7.

Let $f_P^a(y^a|x)$ and $\zeta_P(x)$ be a density function of $Y_t^a$ and $X_t$ under a model $P$. Kaufmann et al. (2016) derives the following result based on change-of-measure argument, which is the principal tool in our lower bound. Let us define a density of $(Y^1, Y^2, \ldots, Y^K, X)$ under a bandit model $P \in \mathcal{P}^*$ as

$$p(y^1, y^2, \ldots, y^K, x) = \prod_{a \in [K]} f_P^a(y^a|x)\zeta_P(x)$$

**Proposition D.1** (Lemma 1 and Remark 2 in Kaufmann et al. (2016)). *For any two bandit model $P, Q \in \mathcal{P}^*$ with $K$ treatment arms such that for all $a \in [K]$, the distributions $P^a$ and $Q^a$ are mutually absolutely continuous. Then,*

$$\sup_{\mathcal{E} \in \mathcal{F}_T} \left| \mathbb{P}_P(\mathcal{E}) - \mathbb{P}_Q(\mathcal{E}) \right| \leq \sqrt{\frac{\mathbb{E}_P\left[L_T(P, Q)\right]}{2}}$$

Recall that $d(p, q)$ indicates the KL divergence between two Bernoulli distributions with parameters $p, q \in (0, 1)$.

This "transportation" lemma provides the distribution-dependent characterization of events under a given bandit model $P$ and corresponding perturbed bandit model $P'$.

Between two bandit models $P, Q \in \mathcal{P}^*$, following the proof of Lemma 1 in Kaufmann et al. (2016), we define the log-likelihood ratio as

$$L_T(P, Q) = \sum_{t=1}^{T} \sum_{a \in [K]} \mathbb{1}[A_t = a] \log \left( \frac{f_P^a(Y_t^a|X_t)\zeta_P(X_t)}{f_Q^a(Y_t^a|X_t)\zeta_Q(X_t)} \right).$$

We consider an approximation of $\mathbb{E}_Q[L_T]$ under an appropriate alternative hypothesis $Q \in \mathcal{P}^*$ when the gaps between the expected outcomes of the best treatment arm and suboptimal treatment arms are small.

### D.1. Observed-Data Bandit Models

For each $x \in \mathcal{X}$, let us define an average allocation ratio under a bandit model $P, Q \in \mathcal{P}^*$ as

$$\frac{1}{T} \sum_{t=1}^{T} \mathbb{E}_P \left[ \mathbb{1}[A_t = a] | X_t = x \right] = \kappa_{T,P}(a|x)$$

This quantity represents an average sample allocation to each treatment arm $a$ under a strategy.

**Lemma D.2.** *For* $P, Q \in \mathcal{P}^*$,

$$\mathbb{E}_P[L_T(P,Q)] = T \sum_{a \in [K]} \mathbb{E}_P \left[ \mathbb{E}_P \left[ \log \frac{f_P^a(Y^a|X)\zeta_P(X)}{f_Q^a(Y^a|X)\zeta_Q(X)} | X \right] \kappa_{T,P}(a|X) \right].$$

Here, recall that $A_t$ is only based on the past observations $\mathcal{F}_{t-1}$ and observed context $X_t$ and independent from $(Y_t^1, \ldots, Y_t^K)$. According to this proposition, we can consider hypothetical observed-data generated as

$$(\widetilde{Y}_t, \widetilde{A}_t, X_t) \overset{\text{i.i.d}}{\sim} \prod_{a \in [K]} \{f_P^a(y^a|x)\kappa_{T,P}(a|X)\}^{\mathbb{1}[d=a]} \zeta_P(x).$$

We present the proof in Appendix E. Then, the expectation of $L_T(P,Q) = \sum_{t=1}^{T} \sum_{a \in [K]} \mathbb{1}[A_t = a] \log \left( \frac{f_P^a(Y_t^a|X_t)\zeta_P(X_t)}{f_Q^a(Y_t^a|X_t)\zeta_Q(X_t)} \right)$ is the same as that under the original observation $P$. Also see (3). Note that this observed-data is induced by the bandit model $P \in \mathcal{P}^*$. For simplicity, we also denote $(\widetilde{Y}_t, \widetilde{A}_t, X_t)$ by $(Y_t, A_t, X_t)$ without loss of generality.

For a bandit model $P \in \mathcal{P}^*$, we consider observed-data distribution $\overline{R}_P$ with the density function given as

$$\overline{r}_P^\kappa(y,d,x) = \prod_{a \in [K]} \{f_P^a(y|x)\kappa_{T,P}(a|x)\}^{\mathbb{1}[d=a]} \zeta_P(x),$$

Let $\mathcal{R}_{\mathcal{P}^*} = \{\overline{R}_P : P \in \mathcal{P}^*\}$ be a set of all observed-data bandit models $\overline{R}_P$. Then, we have

$$\mathbb{E}_P[L_T(P,Q)] = \mathbb{E}_{\mathcal{R}_{\mathcal{P}^*}}[L_T(P,Q)] \tag{3}$$

### D.2. Parametric Submodels for the Observed-data Distribution and Tangent Set

The purpose of this section is to introduce parametric submodels for observed-data distribution, which is indexed by a real-valued parameter and a set of distributions contained in the larger set $\mathcal{R}$, and define the derivative of a parametric submodel as a preparation for the series expansion of the log-likelihood; that is, we consider approximation of the log-likelihood $L_T = \sum_{t=1}^{T} \sum_{a \in [K]} \mathbb{1}[A_t = a] \log \left( \frac{f_P^a(Y_t^a|X_t)\zeta_P(X_t)}{f_Q^a(Y_t^a|X_t)\zeta_Q(X_t)} \right)$ using $\mu^a(P)$.

This section consists of the following three parts. In the first part, we define parametric submodels as (4). Then, in the following part, we confirm the differentiability (6) and define score functions. Finally, we define a set of score functions, called a tangent set in the final paragraph.

By using the parametric submodels and tangent set, in Section D.4, we demonstrate the series expansion of the log-likelihood (Lemma D.5). In this section and Section D.4, we abstractly provide definitions and conditions for the parametric submodels and do not specify them. However, in Section D.5, we show a concrete form of the parametric submodel by finding score functions satisfying the conditions imposed in this section.

**Definition of parametric submodels for the observed-data distribution** First, we define parametric submodels for the observed-data distribution $\overline{R}_P$ with the density function $\overline{r}_P(y,d,x)$ by introducing a parameter $\boldsymbol{\Delta} = (\Delta^a)_{a \in [K]}$ $\Delta^a \in \Theta$ with some compact space $\Theta$. We denote a set of parametric submodels by $\{\overline{R}_{P,\boldsymbol{\Delta}} : \boldsymbol{\Delta} \in \Theta^K\} \subset \mathcal{R}_{\mathcal{P}^*}$, which is defined as follows: for some $g : \mathbb{R} \times [K] \times \mathcal{X} \to \mathbb{R}^K$ satisfying $\mathbb{E}_P[g^a(Y_t, A_t, X_t)] = 0$ and $\mathbb{E}_P[(g^a(Y_t, A_t, X_t))^2] < \infty$, a parametric submodel $\overline{R}_{P,\boldsymbol{\Delta}}$ has a density such that

$$\overline{r}_{\boldsymbol{\Delta}}^\kappa(y,d,x) := 2c(y,d,x;\boldsymbol{\Delta}) \left(1 + \exp\left(-2\boldsymbol{\Delta}^\top g(y,d,x)\right)\right)^{-1} \overline{r}_P^a(y,d,x), \tag{4}$$

$$\mathbb{E}_{\overline{R}_{P,\mathbf{\Delta}}}[Y_t^d] = \int \int y\overline{r}_{\mathbf{\Delta}}^\kappa(y,d,x)\mathrm{d}y\mathrm{d}x = \mu^a(P) + \Delta^a + O((\Delta^a)^2). \tag{5}$$

where $c(y,d,x;\mathbf{\Delta})$ is some function such that $c((y,d,x;\mathbf{0}) = 1$ and $\frac{\partial}{\partial\Delta^a}\big|_{\mathbf{\Delta}=\mathbf{0}} \log c((y,d,x;\mathbf{\Delta}) = 0$ for all $(y,d,x) \in \mathbb{R} \times [K] \times \mathcal{X}$.[3] Note that the parametric submodels are usually not unique. For $a \in [K]$, the parametric submodel is equivalent to $\overline{r}_P(y,a,x)$ if $\Delta^a = 0$. Let $f_{\Delta^a}^a(y|x)$ and $\zeta_{\mathbf{\Delta}}(x)$ be the conditional density of $\widetilde{Y}_t^a$ given $X_t$ and the density of $X_t$, satisfying (4), as

$$\overline{r}_{\mathbf{\Delta}}^\kappa(y,d,x) = \prod_{a\in[K]} \{f_{\Delta^a}^a(y|x)\kappa(a|x)\}^{\mathbb{1}[d=a]} \zeta_{\mathbf{\Delta}}(x).$$

**Differentiablity and score functions of the parametric submodels for the observed-data distribution.** Next, we confirm the differentiablity of $\overline{r}_{\mathbf{\Delta}}^\kappa(y,d,x)$. From the definition of the parametric submodel (4), because $\sqrt{\overline{r}_{\mathbf{\Delta}}^\kappa(y,d,x)}$ is continuously differentiable for every $y,x$ given $d \in [K]$, and $\int \left(\frac{\dot{\overline{r}}_{\mathbf{\Delta}}^\kappa(y,d,x)}{\overline{r}_{\mathbf{\Delta}}^\kappa(y,d,x)}\right)^2 \overline{r}_{\mathbf{\Delta}}^\kappa(y,d,x)\mathrm{d}m$ are well defined and continuous in $\mathbf{\Delta}$, where $m$ is some reference measure on $(y,d,x)$, from Lemma 7.6 of van der Vaart (1998), we see that the parametric submodel has the score function $g$ in the $L_2$ sense; that is, the density $\overline{r}_{\mathbf{\Delta}}^\kappa(y,d,x)$ is differentiable in quadratic mean:

$$\int \left[\overline{r}_{\mathbf{\Delta}}^{\kappa,1/2}(y,d,x) - \overline{r}_P^{\kappa,1/2}(y,d,x) - \frac{1}{2}\mathbf{\Delta}^\top g(y,d,x)\overline{r}_P^{\kappa,1/2}(y,d,x)\right]^2 \mathrm{d}m = o\left(\|\mathbf{\Delta}\|^2\right). \tag{6}$$

In other words, the parametric submodel $\overline{r}_Q^{\kappa,1/2}$ is differentiable in quadratic mean at $\mathbf{\Delta} = 0$ with the score function $g$.

In the following section, we specify a measurable function $g$ satisfying the conditions (4). For each $\Delta^a$ $a \in [K]$, we define the score as

$$S(y,d,x) = \frac{\partial}{\partial\mathbf{\Delta}}\Big|_{\mathbf{\Delta}=\mathbf{0}} \log\overline{r}_{\mathbf{\Delta}}^\kappa(y,d,x) = \begin{pmatrix} \mathbb{1}[d=1]S_f^1(y|x) + S_\zeta^1(x) \\ \mathbb{1}[d=2]S_f^2(y|x) + S_\zeta^2(x) \\ \vdots \\ \mathbb{1}[d=K]S_f^K(y|x) + S_\zeta^K(x) \end{pmatrix}$$

where for each $a \in [K]$, let $S^a(y,d,x) = \mathbb{1}[d=a]S_f^a(y|x) + S_\zeta(x)$,

$$S_f^a(y|x) = \frac{\partial}{\partial\Delta^a}\Big|_{\mathbf{\Delta}=\mathbf{0}} \log f_{\Delta^a}^a(y|x), \qquad S_\zeta^a(x) = \frac{\partial}{\partial\Delta^a}\Big|_{\mathbf{\Delta}=\mathbf{0}} \log\zeta_{\mathbf{\Delta}}(x).$$

Note that $\frac{\partial}{\partial\Delta^a}\log\kappa_{T,P}(a|x) = 0$. Here, we specify $g$ in (4) as the score function of the parametric submodel as $S(y,d,x) = g(y,d,x)$, where $S^a(y,d,x) = g^a(y,d,x)$. This relationship is derived from

$$\frac{\partial}{\partial\mathbf{\Delta}}\Big|_{\mathbf{\Delta}=\mathbf{0}} \log\frac{1}{1+\exp\left(-2\mathbf{\Delta}^\top g(y,d,x)\right)} = \begin{pmatrix} \frac{2g^1(y,d,x)}{\exp(2\mathbf{\Delta}^\top g(y,d,x))+1} \\ \frac{2g^2(y,d,x)}{\exp(2\mathbf{\Delta}^\top g(y,d,x))+1} \\ \vdots \\ \frac{2g^K(y,d,x)}{\exp(2\mathbf{\Delta}^\top g(y,d,x))+1} \end{pmatrix}\Bigg|_{\mathbf{\Delta}=\mathbf{0}} = \begin{pmatrix} g^1(y,d,x) \\ g^2(y,d,x) \\ \vdots \\ g^K(y,d,x) \end{pmatrix}.$$

**Definition of the tangent set.** Recall that parametric submodels and corresponding score functions are not unique. Here, we consider a set of score functions. For a set of the parametric submodels $\{\overline{R}_{P,\mathbf{\Delta}} : \mathbf{\Delta} \in \Theta^K\}$, we obtain a corresponding set of score functions in the Hilbert space $L_2(\overline{R}_P)$, which we call a tangent set of $\mathcal{R}$ at $\overline{R}_P$ and denote it by $\dot{\mathcal{R}}$. Because $\mathbb{E}_{\overline{R}_P}[g^2]$ is automatically finite, the tangent set can be identified with a subset of the Hilbert space $L_2(\overline{R}_P)$, up to equivalence classes. For our parametric submodels, the tangent set at $\overline{R}_P$ in $L_2(\overline{R}_P)$ is given as

$$\dot{\mathcal{R}} = \left\{\begin{pmatrix} \mathbb{1}[d=1]S_f^1(y|x) + S_\zeta^1(x) \\ \mathbb{1}[d=2]S_f^2(y|x) + S_\zeta^2(x) \\ \vdots \\ \mathbb{1}[d=K]S_f^K(y|x) + S_\zeta^K(x) \end{pmatrix}\right\}.$$

---

[3]In (4), $\overline{r}_{\mathbf{\Delta}}^\kappa(y,d,x)$ satisfies the definition of the probability density as discussed in Example 25.15 of van der Vaart (1998).

A linear space of the tangent set is called a *tangent space*. We also define $\dot{\mathcal{R}}^a = \left\{ \left( \mathbb{1}[d = a] S_f^a(y|x) + S_\zeta^a(x) \right) \right\}$.

### D.3. Change-of-Measure

We consider a set of bandit models $\mathcal{P}^\dagger \subset \mathcal{P}^*$ such that $P \in \mathcal{P}^\dagger$, $a \in [K]$, and $x \in \mathcal{X}$, $\mu^a(P)(x) = \mu^a(P)$. Before a bandit process begins, we fix $P^\sharp \in \mathcal{P}^\dagger$ such that $\mu^1(P^\sharp) = \cdots = \mu^K(P^\sharp) = \mu(P^\sharp)$. We choose one treatment arm $d \in [K]$ as the best treatment arm following a multinomial distribution with parameters $(e^1, e^2, \ldots, e^K)$, where $e^a \in [0, 1]$ for all $a \in [K]$ and $\sum_{a \in [K]} e^a = 1$; that is, the expected outcome of the chosen treatment arm $d$ is the highest among the treatment arms. Let $\mathbf{\Delta}$ be a set of parameters such that $\mathbf{\Delta} = (\Delta^c)_{c \in [K]}$, where $\Delta^c \in (0, \infty)$. Let $\mathbf{\Delta}^{(d)}$ be a set of parameters such that $\mathbf{\Delta}^{(d)} = (0, \ldots, \Delta^d, \ldots, 0)$. Then, for each chosen $d \in [K]$, let $Q_{\mathbf{\Delta}^{(d)}} \in \mathcal{P}^\dagger$ be another bandit model such that $d = \arg\max_{a \in [K]} \mu^a(Q_{\mathbf{\Delta}^{(d)}})$, $\mu^b(Q_{\mathbf{\Delta}^{(d)}}) = \mu(P^\sharp)$ for $b \in [K] \backslash \{d\}$, and $\mu^d(Q_{\mathbf{\Delta}^{(d)}}) - \mu(P^\sharp) = \Delta^d + O\left((\Delta^d)^2\right)$. For each $d \in [K]$, we consider $\overline{R}_{P^\sharp, \mathbf{\Delta}^{(d)}} \in \mathcal{R}_{\mathcal{P}^\dagger} \subset \mathcal{R}_{\mathcal{P}^*}$ such that the following equation holds:

$$
\begin{aligned}
L_T(P^\sharp, Q_{\mathbf{\Delta}^{(d)}}) &= \sum_{t=1}^T \sum_{a \in [K]} \left\{ \mathbb{1}[A_t = a] \log\left( \frac{f_{P^\sharp}^a(Y_t^a | X_t)}{f_{Q_{\mathbf{\Delta}^{(d)}}}^a(Y_t^a | X_t)} \right) + \log\left( \frac{\zeta_{P^\sharp}(X_t)}{\zeta_{Q_{\mathbf{\Delta}^{(d)}}}(X_t)} \right) \right\} \\
&= \sum_{t=1}^T \left\{ \mathbb{1}[A_t = d] \log\left( \frac{f_{P^\sharp}^d(Y_t^d | X_t)}{f_{Q_{\mathbf{\Delta}^{(d)}}}^d(Y_t^d | X_t)} \right) + \log\left( \frac{\zeta_{P^\sharp}(X_t)}{\zeta_{Q_{\mathbf{\Delta}^{(d)}}}(X_t)} \right) \right\} \\
&= \sum_{t=1}^T \left\{ \mathbb{1}[A_t = d] \log\left( \frac{f_{P^\sharp}^d(Y_t^d | X_t)}{f_{\mathbf{\Delta}^{(d)}}^d(Y_t^d | X_t)} \right) + \log\left( \frac{\zeta_P(X_t)}{\zeta_{\mathbf{\Delta}^{(d)}}(X_t)} \right) \right\}.
\end{aligned}
$$

Then, let $L_T^a(P^\sharp, Q_{\mathbf{\Delta}^{(d)}})$ be $\sum_{t=1}^T \left\{ \mathbb{1}[A_t = d] \log\left( \frac{f_{P^\sharp}^d(Y_t^d | X_t)}{f_{\mathbf{\Delta}^{(d)}}^d(Y_t^d | X_t)} \right) + \log\left( \frac{\zeta_P(X_t)}{\zeta_{\mathbf{\Delta}^{(d)}}(X_t)} \right) \right\}$. Under the class of bandit models, we show the following lemma.

**Lemma D.3.** *Any null consistent BAI strategy satisfies*

$$
\sup_{P \in \mathcal{P}^*} \mathbb{E}_P[r_T(\pi)(P)] \geq \sup_{\mathbf{\Delta} \in (0, \infty)^K} \sum_{d \in [K]} e^d \Delta^d \left\{ 1 - \mathbb{P}_{P^\sharp}(\widehat{a}_T = d) - \sqrt{\frac{\mathbb{E}_{P^\sharp}\left[ L_T^d(P^\sharp, Q_{\mathbf{\Delta}^{(d)}}) \right]}{2}} + O\left(\Delta^d\right) \right\}.
$$

*Proof of Lemma D.3.* First, we decompose the expected simple regret by using the definition of $\mathcal{P}^\dagger$ as

$$
\begin{aligned}
&\sup_{P \in \mathcal{P}^*} \mathbb{E}_P[r_T(\pi)(P)] \\
&= \sup_{P \in \mathcal{P}^*} \sum_{b \in [K]} \left\{ \max_{a \in [K]} \mu^a(P) - \mu^b(P) \right\} \mathbb{P}_P(\widehat{a}_T = b) \\
&\geq \sup_{\mathbf{\Delta} \in (0, \infty)^K} \sum_{d \in [K]} e^d \sum_{b \in [K] \backslash \{d\}} \left( \mu^d(Q_{\mathbf{\Delta}^{(d)}}) - \mu^b(Q_{\mathbf{\Delta}^{(d)}}) \right) \mathbb{P}_{Q_{\mathbf{\Delta}^{(d)}}}(\widehat{a}_T = b) \\
&\geq \sup_{\mathbf{\Delta} \in (0, \infty)^K} \sum_{d \in [K]} e^d \sum_{b \in [K] \backslash \{d\}} \left( \mu^d(Q_{\mathbf{\Delta}^{(d)}}) - \mu(P^\sharp) \right) \mathbb{P}_{Q_{\mathbf{\Delta}^{(d)}}}(\widehat{a}_T = b) \\
&= \sup_{\mathbf{\Delta} \in (0, \infty)^K} \sum_{d \in [K]} e^d \left\{ \sum_{b \in [K] \backslash \{d\}} \Delta^d \mathbb{P}_{Q_{\mathbf{\Delta}^{(d)}}}(\widehat{a}_T = b) + O\left((\Delta^d)^2\right) \right\} \\
&= \sup_{\mathbf{\Delta} \in (0, \infty)^K} \sum_{d \in [K]} e^d \left\{ \Delta^d \mathbb{P}_{Q_{\mathbf{\Delta}^{(d)}}}(\widehat{a}_T \neq d) + O\left((\Delta^d)^2\right) \right\} \\
&= \sup_{\mathbf{\Delta} \in (0, \infty)^K} \sum_{d \in [K]} e^d \left\{ \Delta^d \left( 1 - \mathbb{P}_{Q_{\mathbf{\Delta}^{(d)}}}(\widehat{a}_T = d) \right) + O\left((\Delta^d)^2\right) \right\}.
\end{aligned}
$$

From Propositions D.5 and D.1. and the definition of null consistent strategies,

$$
\sup_{\mathbf{\Delta} \in (0, \infty)^K} \sum_{d \in [K]} e^d \left\{ \Delta^d \left( 1 - \mathbb{P}_{Q_{\mathbf{\Delta}^{(d)}}}(\widehat{a}_T = d) \right) + O\left((\Delta^d)^2\right) \right\}
$$

$$= \sup_{\mathbf{\Delta} \in (0,\infty)^K} \sum_{d \in [K]} e^d \left\{ \Delta^d \left( 1 - \mathbb{P}_{P^\sharp} (\widehat{a}_T = d) + \mathbb{P}_{P^\sharp} (\widehat{a}_T = d) - \mathbb{P}_{Q_{\mathbf{\Delta}(d)}} (\widehat{a}_T = d) \right) + O\left( (\Delta^d)^2 \right) \right\}$$

$$\geq \sup_{\mathbf{\Delta} \in (0,\infty)^K} \sum_{d \in [K]} e^d \left\{ \Delta^d \left\{ 1 - \mathbb{P}_{P^\sharp} (\widehat{a}_T = d) - \sqrt{\frac{\mathbb{E}_{P^\sharp} \left[ L_T^d(P^\sharp, Q_{\mathbf{\Delta}(d)}) \right]}{2}} \right\} + O\left( (\Delta^d)^2 \right) \right\}.$$

The proof is complete. $\qquad\square$

### D.4. Semiparametric Likelihood Ratio

In this section and next section (Appendix D.5), our goal is to prove the following lemma.

**Lemma D.4.**

$$\mathbb{E}_{P^\sharp} \left[ L_T^d(P^\sharp, Q_{\mathbf{\Delta}(d)}) \right] \leq \frac{T \left( \Delta^a \right)^2}{2 \mathbb{E}_P \left[ \frac{(\sigma^d(X))^2}{w(d|X)} \right]} + O\left( T \left( \Delta^a \right)^3 \right).$$

We consider series expansion of the log-likelihood ratio $L_T$ defined between $P, Q \in \mathcal{P}^\dagger$. We consider an approximation of $L_T$ around a parametric submodel. Because there can be several score functions for our parametric submodel due to directions of the derivative, we find a parametric submodel that has a score function with the largest variance, called a least-favorable parametric submodel (van der Vaart, 1998). Our series expansion is upper-bounded by the variance of the score function, which corresponds to the lower bound for the probability of misidentification.

Inspired by the arguments in Murphy & van der Vaart (1997), we define the semiparametric likelihood ratio expansion to characterize the lower bound for the probability of misidentification with the semiparametric efficiency bound. Note again that the details are different from them owing to the difference of the parameters submodels.

As a preparation, we define a parameter $\mathbb{E}_{\overline{R}_{P,\mathbf{\Delta}(a)}}[Y_t^a]$ as a function $\psi^a : \mathcal{R} \mapsto \mathbb{R}$ such that $\psi^a(\overline{R}_{P,\mathbf{\Delta}(a)}) = \mathbb{E}_{\overline{R}_{P,\mathbf{\Delta}(a)}}[Y_t^a]$. The information bound for $\psi^a(\overline{R}_{P,\mathbf{\Delta}(a)})$ of interest is called semiparametric efficiency bound. Let $\overline{\mathrm{lin}}\dot{\mathcal{R}}$ be the closure of the tangent set. Then, $\psi^a(\overline{R}_{P,\mathbf{\Delta}(a)})$ is pathwise differentiable relative to the tangent set $\dot{\mathcal{R}}^a$ if and only if there exists a function $\widetilde{\psi} \in \overline{\mathrm{lin}}\dot{\mathcal{R}}$ such that

$$\frac{\partial}{\partial \Delta^a} \bigg|_{\Delta^a=0} \psi^a(\overline{R}_{P,\mathbf{\Delta}(a)}) = \mathbb{E}_{\overline{R}_{P,\mathbf{\Delta}(a)}} \left[ \widetilde{\psi}_P^a(Y_t, A_t, X_t) S^a(Y_t, A_t, X_t) \right]. \tag{7}$$

This function $\widetilde{\psi}$ is called the *semiparametric influence function*. Note that the RHS of (7) is calculated as follows:

$$\mathbb{E}_{\overline{R}_{P,\mathbf{\Delta}(a)}} \left[ \widetilde{\psi}_P^a(Y_t, A_t, X_t) S^a(Y_t, A_t, X_t) \right] = \int \int y S_f^a(y|x) f_{\mathbf{\Delta}^a}^a(y|x) \zeta_{\mathbf{\Delta}(a)}(x) \mathrm{d}y \mathrm{d}x + \int \mu^a(x) S_\zeta(x) \zeta_{\mathbf{\Delta}(a)}(x) \mathrm{d}x.$$

Then, we prove the following lemma:

**Lemma D.5.** *For $P \in \mathcal{P}^\dagger$,*

$$\mathbb{E}_P[L_T^a(P, Q)] \leq \frac{1}{2} \frac{T \left( \Delta^a \right)^2}{\mathbb{E}_P \left[ \left( \widetilde{\psi}_P^a(Y_t, A_t, X_t) \right)^2 \right]} + O\left( T \left( \Delta^a \right)^3 \right).$$

To prove this lemma, we define

$$\ell_{\mathbf{\Delta}}^a(y, d, x) = \mathbb{1}[d = a] \left\{ \log f_{\mathbf{\Delta}^a}^a(y^a|x) \right\} + \log \zeta_{\mathbf{\Delta}}(x).$$

Then, by using the parametric submodel defined in the previous section,

$$L_T^a(P, Q) = \sum_{t=1}^T \mathbb{1}[A_t = a] \log \left( \frac{f_P^a(Y_t^a|X_t) \zeta_P(X_t)}{f_Q^a(Y_t^a|X_t) \zeta_Q(X_t)} \right)$$

$$= \sum_{t=1}^{T} \mathbb{1}[A_t = a] \log\left( \frac{f_P^a(Y_t^a|X_t)\zeta_P(X_t)}{f_{\Delta^a}^a(Y_t^a|X_t)\zeta_{\Delta}(X_t)} \right)$$

$$= \sum_{t=1}^{T} \left( -\frac{\partial}{\partial \Delta^a}\Big|_{\Delta^a=0} \ell_{\Delta^{(a)}}^a(Y_t, A_t, X_t)\Delta^a - \frac{\partial^2}{\partial(\Delta^a)^2}\Big|_{\Delta^a=0} \ell_{\Delta^{(a)}}^a(Y_t, A_t, X_t)\frac{(\Delta^a)^2}{2} + O\left((\Delta^a)^3\right) \right).$$

Here, note that

$$\frac{\partial}{\partial \Delta^a}\Big|_{\Delta^a=0} \ell_{\Delta^{(a)}}^a(Y_t, A_t, X_t) = S^a(Y_t, A_t, X_t) = g^a(Y_t, A_t, X_t)$$

$$\frac{\partial}{\partial(\Delta^a)^2}\Big|_{\Delta^a=0} \ell_{\Delta^{(a)}}^a(Y_t, A_t, X_t) = -\left(S^a(Y_t, A_t, X_t)\right)^2.$$

By using the expansion, we evaluate $\mathbb{E}_P[L_T^a]$. Here, by definition, $\mathbb{E}_P[S^a(Y_t, A_t, X_t)] = 0$. Therefore, we consider an upper bound of $\frac{1}{\mathbb{E}_P[(S^a(Y_t, A_t, X_t))^2]}$ for $S \in \mathcal{R}$.

Then, we prove the following lemma on the upper bound for $\frac{1}{\mathbb{E}_P[(S^a(Y_t, A_t, X_t))^2]}$:

**Lemma D.6.** *For $P \in \mathcal{P}^\dagger$,*

$$\sup_{S \in \dot{\mathcal{R}}} \frac{1}{\mathbb{E}_P\left[(S^a(Y_t, A_t, X_t))^2\right]} \leq \mathbb{E}_P\left[\left(\widetilde{\psi}_P^a(Y_t, A_t, X_t)\right)^2\right]$$

*Proof.* From the Cauchy-Schwarz inequality, we have

$$1 = \mathbb{E}_P\left[\widetilde{\psi}_P^a(Y_t, A_t, X_t)S^a(Y_t, A_t, X_t)\right] \leq \sqrt{\mathbb{E}_P\left[\left(\widetilde{\psi}_P^a(Y_t, A_t, X_t)\right)^2\right]}\sqrt{\mathbb{E}_P\left[(S^a(Y_t, A_t, X_t))^2\right]}.$$

Therefore,

$$\sup_{S \in \dot{\mathcal{R}}} \frac{1}{\mathbb{E}_P\left[(S^a(Y_t, A_t, X_t))^2\right]} \leq \mathbb{E}_P\left[\left(\widetilde{\psi}_P^a(Y_t, A_t, X_t)\right)^2\right].$$

$\square$

According to this lemma, to derive the upper bound for $\frac{1}{\mathbb{E}_P[(S^a(Y_t, A_t, X_t))^2]}$, let us define the *semiparametric efficient score* $S_{\mathrm{eff}}^a(Y_t, A_t, X_t) \in \overline{\mathrm{lin}}\dot{\mathcal{R}}^a$ as

$$S_{\mathrm{eff}}^a(Y_t, A_t, X_t) = \frac{\widetilde{\psi}_P^a(Y_t, A_t, X_t)}{\mathbb{E}_P\left[\left(\widetilde{\psi}_P^a(Y_t, A_t, X_t)\right)^2\right]}.$$

Then, by using the semiparametric efficient score $S_{\mathrm{eff}}^a(Y_t, A_t, X_t)$, we approximate the likelihood ratio as follows:

*Proof of Lemma D.5.*

$$\mathbb{E}_{P'}[L_T^a(P, Q)] = T\mathbb{E}_P\left[\frac{1}{2}(S^a(Y_t, A_t, X_t))^2(\Delta^a)^2 + O((\Delta^a)^3)\right]$$

$$\leq T\mathbb{E}_P\left[\frac{1}{2}(S_{\mathrm{eff}}^a(Y_t, A_t, X_t))^2(\Delta^a)^2 + O((\Delta^a)^3)\right]$$

$$= \frac{1}{2}\frac{T(\Delta^a)^2}{\mathbb{E}_P\left[\left(\widetilde{\psi}_P^a(Y_t, A_t, X_t)\right)^2\right]} + O\left(T(\Delta^a)^3\right).$$

$\square$

### D.5. Observed-Data Semiparametric Efficient Influence Function

Our remaining is task is to find $\widetilde{\psi}_P^a \in \overline{\mathrm{lin}}\dot{\mathcal{R}}$ in (7). Our derivation mainly follows Hahn (1998). We guess that $\widetilde{\psi}_P^a(Y_t, A_t, X_t)$ has the following form:

$$\widetilde{\psi}_P^a(y, d, x) = \frac{\mathbb{1}[d = a](y - \mu^a(P)(x))}{\kappa_{T,P}(a|X)} + \mu^a(P)(x) - \mu^a(P). \tag{8}$$

Then, as shown by Hahn (1998), the condition $\frac{\partial}{\partial \Delta^a}\Big|_{\boldsymbol{\Delta}^{(a)}=\mathbf{0}} \psi^a(\overline{R}_{P,\boldsymbol{\Delta}^{(a)}}) = \mathbb{E}_{\overline{R}_Q}\left[\widetilde{\psi}_P^a(Y_t, A_t, X_t)S^a(Y_t, A_t, X_t)\right]$ holds under (8) when the score functions are given as

$$S_f^a(y|x) = \frac{(y - \mu^a(P)(x))}{\kappa_{T,P}(a|x)}/\widetilde{V}^a(\kappa_{T,P}), \qquad S_\zeta^a(x) = \left(\mu^a(P)(x) - \mu^a(P)\right)/\widetilde{V}^a(\kappa_{T,P}) \quad \text{for } a \in [K],$$

where

$$\widetilde{V}^a(\kappa_{T,P}) := \mathbb{E}_P\left[\left(\frac{\mathbb{1}[d = a](y - \mu^a(P)(x))}{\kappa_{T,P}(a|X)} + \mu^a(P)(x) - \mu^a(P)\right)^2\right] = \mathbb{E}_P\left[\frac{(\sigma^a(X_t))^2}{\kappa_{T,P}(a|X_t)} + (\mu^a(P)(X_t) - \mu^a(P))^2\right].$$

Therefore,

$$S^a(y, d, x) = \left(\frac{\mathbb{1}[d = a](y - \mu^a(P)(x))}{\kappa_{T,P}(a|X)} + \mu^a(P)(x) - \mu^a(P)\right)/\widetilde{V}^a(\kappa_{T,P}).$$

Our specified score function satisfies (5) because we can confirm that

$$\psi^a(\overline{R}_{P,\mathbf{0}}) = \mu^a(P),$$

and

$$\frac{\partial}{\partial \Delta^a}\Big|_{\Delta^a=0} \psi^a(\overline{R}_{P,\boldsymbol{\Delta}^{(a)}}) = \mathbb{E}_{\overline{R}_Q}\left[\widetilde{\psi}_P^a(Y_t, A_t, X_t)S^a(Y_t, A_t, X_t)\right]$$
$$= \mathbb{E}_{\overline{R}_Q}\left[\left(\frac{\mathbb{1}[d = a](y - \mu^a(P)(x))}{\kappa_{T,P}(a|X)} + \mu^a(P)(x) - \mu^a(P)\right)^2/\widetilde{V}^a(\kappa_{T,P})\right] = 1.$$

Then, from the first-order series expansion of $\psi^a(\overline{R}_{P,\boldsymbol{\Delta}^{(a)}})$ around $\Delta^a = 0$, we obtain

$$\psi^a(\overline{R}_{P,\boldsymbol{\Delta}^{(a)}}) = \psi^a(\overline{R}_{P,\mathbf{0}}) + \Delta^a\frac{\partial}{\partial \Delta^a}\Big|_{\Delta^a=0}\psi^a(\overline{R}_{P,\boldsymbol{\Delta}^{(a)}}) + O((\Delta^a)^2) = \mu^a(P) + \Delta^a + O((\Delta^a)^2).$$

Summarizing the above arguments, we obtain the following lemma.

**Lemma D.7.** *For $P \in \mathcal{P}^\dagger$, the semiparametric efficient influence function is*

$$\widetilde{\psi}_P^a(y, d, x) = \widetilde{V}^a(\kappa_{T,P})\left(\mathbb{1}[d = a]S_f^a(y|x) + S_\zeta(x)\right)$$
$$= \frac{\mathbb{1}[d = a](y - \mu^a(P)(x))}{\kappa_{T,P}(a|x)} + \mu^a(P)(x) - \mu^a(P).$$

Thus, under $g$ with our specified score functions, we can confirm that the semiparametric influence function $\widetilde{\psi}_P^a(y, d, x) = \widetilde{V}^a(\kappa_{T,P})\left(\mathbb{1}[A_t = a]S_f^a(y|x) + S_\zeta(x)\right)$ belongs to $\overline{\mathrm{lin}}\dot{\mathcal{R}}$. Note that $\mathbb{E}_{\overline{R}_Q}[S_{\mathrm{eff}}^a(Y_t, A_t, X_t)] = 0$ and

$$\mathbb{E}_{\overline{R}_Q}\left[\left(S_{\mathrm{eff}}^a(Y_t, A_t, X_t)\right)^2\right] = \left(\mathbb{E}_{\overline{R}_Q}\left[\left(\widetilde{\psi}_P^a(Y_t, A_t, X_t)\right)^2\right]\right)^{-1}.$$

In summary, from Lemmas D.5, D.6, and D.7, we obtain Lemma D.4. Note that because $\mu^a(P)(x) = \mu^a$ for $P \in \mathcal{P}^\dagger$, $\widetilde{V}^a(\kappa_{T,P}) := \mathbb{E}_P\left[\frac{(\sigma^a(X_t))^2}{\kappa_{T,P}(a|X_t)}\right]$.

## D.6. The Worst Case Bandit Model

We show the final step of the proof.

*Proof.* Then, from Lemmas D.3 and D.7, for all $d \in [K]$, and definition of the null consistent strategy, for any $\epsilon > 0$, there exists $T_0 > 0$ such that for all $T > T_0$,

$$
\sup_{P \in \mathcal{P}^*} \mathbb{E}_P[r_T(\pi)(P)] \geq \sup_{\boldsymbol{\Delta} \in (0,\infty)^K} \sum_{d \in [K]} e^d \Delta^d \left\{ 1 - \mathbb{P}_{P^\sharp}(\widehat{a}_T = d) - \sqrt{\frac{\mathbb{E}_{P^\sharp}\left[L_T^d(P^\sharp, Q_{\boldsymbol{\Delta}^{(d)}})\right]}{2}} + O\left(\Delta^d\right) \right\}
$$

$$
\geq \inf_{w \in \mathcal{W}} \sup_{\boldsymbol{\Delta} \in (0,\infty)^K} \sum_{d \in [K]} e^d \Delta^d \left\{ 1 - \frac{1}{K} - \sqrt{\frac{T(\Delta^d)^2}{2\mathbb{E}_{P^\sharp}\left[\frac{(\sigma^d(X_t))^2}{w(d|X_t)}\right]}} + O\left(T((\Delta^d)^3\right) + O\left(\Delta^d\right) \right\} - \epsilon
$$

$$
\geq \inf_{w \in \mathcal{W}} \sup_{\boldsymbol{\Delta} \in (0,\infty)^K} \sum_{d \in [K]} e^d \Delta^d \left\{ \frac{1}{2} - \sqrt{\frac{T\left(\Delta^d\right)^2}{2\mathbb{E}_{P^\sharp}\left[\frac{(\sigma^d(X_t))^2}{w(d|X_t)}\right]}} + O\left(T((\Delta^d)^3\right) + O\left(\Delta^d\right) \right\} - \epsilon.
$$

The maximizer of $\sup_{\boldsymbol{\Delta} \in (0,\infty)^K} \sum_{d \in [K]} e^d \Delta^d \left\{ \frac{1}{2} - \sqrt{\frac{T(\Delta^d)^2}{2\mathbb{E}_{P^\sharp}\left[\frac{(\sigma^d(X_t))^2}{w(d|X_t)}\right]}} \right\}$ is given as $\Delta^a = \frac{1}{4}\sqrt{\frac{2\mathbb{E}_{P^\sharp}\left[\frac{(\sigma^a(X_t))^2}{w(a|X_t)}\right]}{T}}$. Therefore,

$$
\sup_{P \in \mathcal{P}^*} \mathbb{E}_P[r_T(\pi)(P)] \geq \inf_{w \in \mathcal{W}} \sup_{\boldsymbol{\Delta} \in (0,\infty)^K} \sum_{d \in [K]} e^d \Delta^d \left\{ \frac{1}{2} - \sqrt{\frac{T\left(\Delta^d\right)^2}{2\mathbb{E}_{P^\sharp}\left[\frac{(\sigma^d(X_t))^2}{w(d|X_t)}\right]}} + O\left(T\left(\Delta^d\right)^3\right) + O\left(\Delta^d\right) \right\} - \epsilon
$$

$$
\geq \frac{1}{12} \inf_{w \mathcal{W}} \sum_{d \in [K]} e^d \left\{ \sqrt{\frac{\mathbb{E}_{P^\sharp}\left[\frac{(\sigma^d(X_t))^2}{w(d|X_t)}\right]}{T}} + O\left(\frac{2\mathbb{E}_{P^\sharp}\left[\frac{(\sigma^d(X_t))^2}{w(d|X_t)}\right]}{T}\right) \right\} - \epsilon.
$$

As $T \to \infty$, letting $\epsilon \to 0$,

$$
\sup_{P \in \mathcal{P}^*} \sqrt{T} \mathbb{E}_P[r_T(\pi)(P)] \geq \frac{1}{12} \inf_{w \mathcal{W}} \sum_{d \in [K]} e^d \sqrt{\mathbb{E}_{P^\sharp}\left[\frac{(\sigma^d(X_t))^2}{w(d|X_t)}\right]} + o\left(1\right).
$$

$\square$

## D.7. Characterization of the Target Allocation Ratio

*Proof of Theorem 3.4.* We showed that any null consistent BAI strategy satisfies

$$
\sup_{P \in \mathcal{P}^*} \mathbb{E}_P[r_T(\pi)(P)] \geq \frac{1}{12} \inf_{w \mathcal{W}} \sum_{d \in [K]} e^d \sqrt{\frac{(\sigma^d)^2}{w(d)}} + o\left(1\right).
$$

In the tight lower bound, $e^{\widetilde{d}} = 1$ for $\widetilde{d} = \arg\max_{d \in [K]} \frac{1}{12} \sqrt{\frac{(\sigma^d)^2}{w(d)}} + o(1)$[4]. Therefore, we consider solving

$$
\inf_{w \in \mathcal{W}} \max_{d \in [K]} \sqrt{\frac{(\sigma^d)^2}{w(d)}}.
$$

---

[4]If there are multiple candidates of the best treatment arm, we choose one of the multiple treatment arms as the best treatment arm with probability 1.

If there exists a solution, we can replace the inf with the min. We consider the following constrained optimization:

$$\inf_{R \in \mathbb{R}, w \in \mathcal{W}} R$$

$$\text{s.t.} \quad R \geq \frac{\left(\sigma^d\right)^2}{w(d)} \quad \forall d \in [K]$$

$$\sum_{a \in [K]} w(a) = 1.$$

For this problem, we derive the first-order condition, which is sufficient for the global optimality of such a convex programming problem. For Lagrangian multipliers $\lambda^d \in (-\infty, 0]$ and $\gamma \in \mathbb{R}$, we consider the following Lagrangian function:

$$L(\boldsymbol{\lambda}, \gamma; R, w) = R + \sum_{d \in [K]} \lambda^d \left\{ \frac{\left(\sigma^d\right)^2}{w(d)} - R \right\} + \gamma \left\{ \sum_{d \in [K]} w(d) - 1 \right\}.$$

Then, the optimal solutions $w^*$, $\lambda^{*d}$, $\gamma^*$, and $R^*$ of the original problem satisfies

$$1 - \sum_{d \in [K]} \lambda^{d*} = 0 \quad \forall x \in \mathcal{X}$$

$$-\lambda^{d*} \frac{\left(\sigma^d\right)^2}{(w^*(d))^2} = \gamma^* \quad \forall d \in [K],$$

$$\lambda^{d*} \left\{ \frac{\left(\sigma^d\right)^2}{w(d)} - R^* \right\} = 0$$

$$\gamma^*(x) \left\{ \sum_{a \in [K]} w^*(a) - 1 \right\} = 0 \quad \forall a \in [K].$$

Here, the solutions are given as

$$w^*(d) = \frac{\left(\sigma^d\right)^2}{\sum_{b \in [K]} \left(\sigma^b\right)^2},$$

$$\lambda^{d*} = w^*(d),$$

$$\gamma^*(x) = -\sum_{b \in [K]} \left(\sigma^b\right)^2.$$

Therefore,

$$\inf_{\boldsymbol{w}\mathcal{W}} \sum_{a \in [K]} e^a \frac{1}{12} \sqrt{\mathbb{E}_{P^\sharp} \left[ \frac{(\sigma^a(X))^2}{w(a|X)} \right]} + o(1) = \frac{1}{12} \sqrt{\mathbb{E}_{P^\sharp} \left[ \sum_{a \in [K]} (\sigma^a(X))^2 \right]} \sum_{a \in [K]} e^a + o(1).$$

Since $\sum_{a \in [K]} e^a = 1$ and $\zeta_P(x) = \zeta(x)$, the proof is complete.

$\square$

Here, $\widetilde{w}(a|x) = \frac{(\sigma^a(x))^2}{\sum_{b \in [K]}(\sigma^b(x))^2}$ works as a target allocation ratio in implementation of our BAI strategy because it represents the sample average of $\mathbb{1}[A_t = a]$; that is, we design our sampling rule $(A_t)_{t \in [T]}$ for the average to be the target allocation ratio.

Although this lower bound is applicable to a case with $K = 2$, we can tighten the lower bound by changing the definiton of the parametric submodel.

# E. Proof of Lemma D.2

*Proof.*

$$\mathbb{E}_Q[L_T] = \sum_{t=1}^{T} \mathbb{E}_Q \left[ \sum_{a \in [K]} \mathbb{1}\{A_t = a\} \log \frac{f_Q^a(Y_t^a|X_t)\zeta_Q(X_t)}{f_{P_0}^a(Y_t^a|X_t)\zeta_{P_0}(X_t)} \right]$$

$$= \sum_{t=1}^{T} \mathbb{E}_Q^{X_t, \mathcal{F}_{t-1}} \left[ \sum_{a \in [K]} \mathbb{E}_Q^{Y_t^a, A_t} \left[ \mathbb{1}[A_t = a] \log \frac{f_Q^a(Y_t^a|X_t)\zeta_Q(X_t)}{f_{P_0}^a(Y_t^a|X_t)\zeta_{P_0}(X_t)} \middle| X_t, \mathcal{F}_{t-1} \right] \right]$$

$$= \sum_{t=1}^{T} \mathbb{E}_Q^{X_t, \mathcal{F}_{t-1}} \left[ \sum_{a \in [K]} \mathbb{E}_Q \left[ \mathbb{1}[A_t = a]|X_t, \mathcal{F}_{t-1} \right] \mathbb{E}_Q^{Y_t^a} \left[ \log \frac{f_Q^a(Y_t^a|X_t)\zeta_Q(X_t)}{f_{P_0}^a(Y_t^a|X_t)\zeta_{P_0}(X_t)} \middle| X_t, \mathcal{F}_{t-1} \right] \right]$$

$$= \sum_{t=1}^{T} \mathbb{E}_Q^{X_t} \left[ \mathbb{E}_Q^{\mathcal{F}_t} \left[ \sum_{a \in [K]} \mathbb{E}_Q \left[ \mathbb{1}[A_t = a]|X_t, \mathcal{F}_{t-1} \right] \mathbb{E}_Q^{Y_t^a} \left[ \log \frac{f_Q^a(Y_t^a|X_t)\zeta_Q(X_t)}{f_{P_0}^a(Y_t^a|X_t)\zeta_{P_0}(X_t)} \middle| X_t \right] \right] \right]$$

$$= \sum_{t=1}^{T} \int \left( \sum_{a \in [K]} \mathbb{E}_Q^{\mathcal{F}_t} \left[ \mathbb{E}_Q \left[ \mathbb{1}[A_t = a]|X_t = x, \mathcal{F}_{t-1} \right] \right] \mathbb{E}_Q^{Y_t^a} \left[ \log \frac{f_Q^a(Y_t^a|X_t)\zeta_Q(X_t)}{f_{P_0}^a(Y_t^a|X_t)\zeta_{P_0}(X_t)} \middle| X_t = x \right] \right) \zeta_Q(x)\mathrm{d}x$$

$$= \int \sum_{a \in [K]} \left( \mathbb{E}_Q^{Y^a} \left[ \log \frac{f_Q^a(Y^a|X)\zeta_Q(X)}{f_{P_0}^a(Y^a|X)\zeta_{P_0}(X)} \middle| X = x \right] \sum_{t=1}^{T} \mathbb{E}_Q^{\mathcal{F}_t} \left[ \mathbb{E}_Q \left[ \mathbb{1}[A_t = a]|X_t = x, \mathcal{F}_{t-1} \right] \right] \right) \zeta_Q(x)\mathrm{d}x$$

$$= \mathbb{E}_Q^X \left[ \sum_{a \in [K]} \mathbb{E}_Q^{Y^a} \left[ \log \frac{f_Q^a(Y^a|X)\zeta_Q(X)}{f_{P_0}^a(Y^a|X)\zeta_{P_0}(X)} \middle| X \right] \sum_{t=1}^{T} \mathbb{E}_Q^{\mathcal{F}_{t-1}} \left[ \mathbb{E}_Q \left[ \mathbb{1}[A_t = a]|X, \mathcal{F}_{t-1} \right] \right] \right],$$

where $\mathbb{E}_Q^Z$ denotes an expectation of random variable $Z$ over the distribution $Q$. We used that the observations $(Y_t^1, \dots, Y_t^K, X_t)$ are i.i.d. across $t \in \{1, 2, \dots, T\}$. $\qquad\square$

# F. Proof of the Asymptotic Lower Bound for Two-Armed Bandits (Theorem 3.6)

When $K = 2$, we define different parametric submodels from those in Section D.

**Parametric submodels for the observed-data distribution and tangent set.** In a case with $K = 2$, we consider one-parameter parametric submodels for the observed-data distribution $\overline{R}_P$ with the density function $\overline{r}_P(y, d, x)$ by introducing a parameter $\Delta \in \Theta$ with some compact space $\Theta$. We denote a set of parametric submodels by $\{\overline{R}_{P,\Delta} : \Delta \in \Theta\} \subset \mathcal{R}_{\mathcal{P}^*}$, which is defined as follows: for some $g : \mathbb{R} \times [2] \times \mathcal{X} \to \mathbb{R}$ satisfying $\mathbb{E}_P[g(Y_t, A_t, X_t)] = 0$ and $\mathbb{E}_P[(g(Y_t, A_t, X_t))^2] < \infty$, a parametric submodel $\overline{R}_{P,\Delta}$ has a density such that

$$\overline{r}_\Delta^\kappa(y, d, x) := 2c(y, d, x; \Delta) \left(1 + \exp\left(-2\Delta g(y, d, x)\right)\right)^{-1} \overline{r}_P^a(y, d, x),$$

where $c(y, d, x; \Delta)$ is some function such that $c((y, d, x; 0) = 1$ and $\left.\frac{\partial}{\partial \Delta}\right|_{\Delta=0} \log c((y, d, x; \Delta) = 0$ for all $(y, d, x) \in \mathbb{R} \times [2] \times \mathcal{X}$. Note that the parametric submodels are usually not unique. The parametric submodel is equivalent to $\overline{r}_P(y, a, x)$ if $\Delta = 0$.

Let $f_\Delta^a(y|x)$ and $\zeta_\Delta(x)$ be the conditional density of $y$ given $x$ and some density of $x$, satisfying (4) as

$$\overline{r}_\Delta^\kappa(y, d, x) = \prod_{a \in [2]} \{f_\Delta^a(y|x)\kappa(a|x)\}^{\mathbb{1}[d=a]} \zeta_\Delta(x).$$

For this parametric submodel, we develop the same argument in Section D. Note that we consider one-parameter parametric submodel for two-armed bandits, while in Section D, we consider $K$-dimensional parametric submodels for $K$-armed bandits.

**Change-of-measure.** We consider a set of bandit models $\mathcal{P}^{\dagger\dagger} \subset \mathcal{P}^*$ such that for all $P \in \mathcal{P}^{\dagger\dagger}$, $a \in [K]$, and $x \in \mathcal{X}$, $\mu^a(P)(x) = \mu^a$. Before a bandit process begins, we fix $P^{\sharp\sharp} \in \mathcal{P}^{\dagger\dagger}$ such that $\mu^1(P^{\sharp\sharp}) = \mu^2(P^{\sharp\sharp}) = \mu(P^{\sharp\sharp})$. We choose one treatment arm $d \in [2]$ as the best treatment arm following a Bernoulli distribution with parameter $e \in [0, 1]$; that is, the expected outcome of the chosen treatment arm $d$ is the highest among the treatment arms. We choose treatment arm 1 with probability $e$ and treatment arm 2 with probability $1 - e$. Let $\Delta \in (0, \infty)$ be a gap parameter and $Q_\Delta \in \mathcal{P}^{\dagger\dagger}$ be another bandit model such that $d = \arg\max_{a\in[2]} \mu^a(Q_\Delta)$, $\mu^b(Q_\Delta) = \mu(P^{\sharp\sharp})$ for $b \neq d$, and $\mu^d(Q_\Delta) - \mu(P^{\sharp\sharp}) = \Delta + O(\Delta^2)$. For the parameter $\Delta$, we consider $\overline{R}_{P^{\sharp\sharp},\Delta} \in \mathcal{R}_{\mathcal{P}^{\dagger\dagger}} \subset \mathcal{R}_{\mathcal{P}^*}$ such that the following equation holds:

$$
L_T(P, Q) = \sum_{t=1}^{T} \left\{ \mathbb{1}[A_t = 1] \log \left( \frac{f_P^1(Y_t^1|X_t)}{f_Q^1(Y_t^1|X_t)} \right) + \mathbb{1}[A_t = 2] \log \left( \frac{f_P^2(Y_t^2|X_t)}{f_Q^2(Y_t^2|X_t)} \right) + \log \left( \frac{\zeta_P(X_t)}{\zeta_Q(X_t)} \right) \right\}
$$

$$
= \sum_{t=1}^{T} \left\{ \mathbb{1}[A_t = 1] \log \left( \frac{f_P^1(Y_t^1|X_t)}{f_\Delta^a(Y_t^1|X_t)} \right) + \mathbb{1}[A_t = 2] \log \left( \frac{f_P^2(Y_t^2|X_t)}{f_\Delta^2(Y_t^2|X_t)} \right) + \log \left( \frac{\zeta_P(X_t)}{\zeta_\Delta(X_t)} \right) \right\}.
$$

*Proof of Theorem 3.6.* First, we decompose the expected simple regret by using the definition of $\mathcal{P}^{\dagger\dagger}$ as

$$
\sup_{P \in \mathcal{P}^*} \mathbb{E}_P[r_T(\pi)(P)]
$$

$$
= \sup_{P \in \mathcal{P}^*} \sum_{b \in [2]} \left\{ \max_{a \in [2]} \mu^a(P) - \mu^b(P) \right\} \mathbb{P}_P(\widehat{a}_T = b)
$$

$$
\geq \sup_{\Delta \in (0,\infty)} \left\{ e\left(\mu^1(Q_\Delta) - \mu^2(Q_\Delta)\right) \mathbb{P}_{Q_\Delta}(\widehat{a}_T = 2) + (1-e)\left(\mu^2(Q_\Delta) - \mu^1(Q_\Delta)\right) \mathbb{P}_{Q_\Delta}(\widehat{a}_T = 1) \right\}
$$

$$
= \sup_{\Delta \in (0,\infty)} \left\{ e\left(\mu^1(Q_\Delta) - \mu(P^{\sharp\sharp})\right) \mathbb{P}_{Q_\Delta}(\widehat{a}_T = 2) + (1-e)\left(\mu^2(Q_\Delta) - \mu(P^\sharp)\right) \mathbb{P}_{Q_\Delta}(\widehat{a}_T = 1) \right\}
$$

$$
= \sup_{\Delta \in (0,\infty)} \left\{ e\left(\Delta + O(\Delta^2)\right) \mathbb{P}_{Q_\Delta}(\widehat{a}_T = 2) + (1-e)\left(\Delta + O(\Delta^2)\right) \mathbb{P}_{Q_\Delta}(\widehat{a}_T = 1) \right\}
$$

$$
= \sup_{\Delta \in (0,\infty)} \left\{ e\Delta\mathbb{P}_{Q_\Delta}(\widehat{a}_T = 2) + (1-e)\Delta\mathbb{P}_{Q_\Delta}(\widehat{a}_T = 1) + O(\Delta^2) \right\}
$$

$$
= \sup_{\Delta \in (0,\infty)} \left\{ e\Delta\left(1 - \mathbb{P}_{Q_\Delta}(\widehat{a}_T = 1)\right) + (1-e)\Delta\left(1 - \mathbb{P}_{Q_\Delta}(\widehat{a}_T = 2)\right) + O(\Delta^2) \right\}.
$$

From Propositions D.5 and D.1 and definition of the null consistent strategy,

$$
\sup_{\Delta \in (0,\infty)} \left\{ e\Delta\left(1 - \mathbb{P}_{Q_\Delta}(\widehat{a}_T = 1)\right) + (1-e)\Delta\left(1 - \mathbb{P}_{Q_\Delta}(\widehat{a}_T = 2)\right) + O(\Delta^2) \right\}
$$

$$
= \sup_{\Delta \in (0,\infty)} \left\{ e\Delta\left(1 - \mathbb{P}_{P^{\sharp\sharp}}(\widehat{a}_T = 1) + \mathbb{P}_{P^{\sharp\sharp}}(\widehat{a}_T = 1) - \mathbb{P}_{Q_\Delta}(\widehat{a}_T = 1)\right) \right.
$$

$$
\left. \qquad + (1-e)\Delta\left(1 - \mathbb{P}_{P^{\sharp\sharp}}(\widehat{a}_T = 2) + \mathbb{P}_{P^{\sharp\sharp}}(\widehat{a}_T = 2) - \mathbb{P}_{Q_\Delta}(\widehat{a}_T = 2)\right) + O(\Delta^2) \right\}
$$

$$
= \sup_{\Delta \in (0,\infty)} \left\{ e\Delta\left(1 - \mathbb{P}_{P^{\sharp\sharp}}(\widehat{a}_T = 1) - \sqrt{\frac{\mathbb{E}_{P^{\sharp\sharp}}[L_T(P^{\sharp\sharp}, Q_\Delta)]}{2}}\right) \right.
$$

$$
\left. \qquad + (1-e)\Delta\left(1 - \mathbb{P}_{P^{\sharp\sharp}}(\widehat{a}_T = 2) - \sqrt{\frac{\mathbb{E}_{P^{\sharp\sharp}}[L_T(P^{\sharp\sharp}, Q_\Delta)]}{2}}\right) + O(\Delta^2) \right\}
$$

$$
= \sup_{\Delta \in (0,\infty)} \left\{ e\Delta\left(1 - \frac{1}{2} - \sqrt{\frac{\mathbb{E}_{P^{\sharp\sharp}}[L_T(P^{\sharp\sharp}, Q_\Delta)]}{2}}\right) + (1-e)\Delta\left(1 - \frac{1}{2} - \sqrt{\frac{\mathbb{E}_{P^{\sharp\sharp}}[L_T(P^{\sharp\sharp}, Q_\Delta)]}{2}}\right) + O(\Delta^2) \right\}
$$

$$
= \sup_{\Delta \in (0,\infty)} \left\{ \Delta\left(\frac{1}{2} - \sqrt{\frac{\mathbb{E}_{P^{\sharp\sharp}}[L_T(P^{\sharp\sharp}, Q_\Delta)]}{2}}\right) + O(\Delta^2) \right\}
$$

$$
\geq \inf_{w \in \mathcal{W}} \sup_{\Delta \in (0,\infty)} \left\{ \Delta\left(\frac{1}{2} - \sqrt{\frac{T\Delta^2}{2\mathbb{E}_P\left[\frac{(\sigma^1(X_t))^2}{w(1|X)} + \frac{(\sigma^2(X))^2}{w(2|X_t)}\right]}} + O(T\Delta^3)\right) + O(\Delta^2) \right\}.
$$

Let $\Delta = \frac{1}{4}\sqrt{\dfrac{2\mathbb{E}_P\left[\frac{(\sigma^1(X))^2}{w(1|X)} + \frac{(\sigma^2(X))^2}{w(2|X)}\right]}{T}}$. Then,

$$\inf_{w\in\mathcal{W}}\sup_{\Delta\in(0,\infty)}\left\{\Delta\left(\frac{1}{2} - \sqrt{\frac{T\Delta^2}{2\mathbb{E}_P\left[\frac{(\sigma^1(X_t))^2}{w(1|X)} + \frac{(\sigma^2(X))^2}{w(2|X_t)}\right]} + O\left(T\Delta^3\right)}\right) + O(\Delta^2)\right\}$$

$$\geq \frac{1}{12}\inf_{w\mathcal{W}}\sqrt{\frac{\mathbb{E}_P\left[\frac{(\sigma^1(X))^2}{w(1|X_t)} + \frac{(\sigma^2(X_t))^2}{w(2|X_t)}\right]}{T}} + O\left(\Delta^2\right)$$

$$\geq \frac{1}{12}\sqrt{\frac{\mathbb{E}_P\left[(\sigma^1(X) + \sigma^2(X))^2\right]}{T}} + O\left(\frac{\mathbb{E}_P\left[(\sigma^1(X) + \sigma^2(X))^2\right]}{T}\right).$$

Here, the minimizer regarding $w$ is $\widetilde{w}(1|x) = \frac{\sigma^1(x)}{\sigma^1(x)+\sigma^2(x)}$ ($\widetilde{w}(2|x) = 1 - \widetilde{w}(1|x)$) (van der Laan, 2008; Hahn et al., 2011; Kato et al., 2020). Because $\zeta_P(x) = \zeta(x)$, $\sup_{P'\in\mathcal{P}^*}\sqrt{\mathbb{E}_{P'}[r_T(\pi)(P)]} \geq \frac{1}{12}\sqrt{\int(\sigma^1(X) + \sigma^2(X))^2\zeta(x)\mathrm{d}x} + o(1)$. $\square$

## G. Proof of Theorem 5.2

*Proof.* Let

$$\xi_T^{a,b}(P) = \frac{\sqrt{T}\left(\widehat{\Delta}_T^{\mathrm{HIR},a,b} - \Delta^{a,b}(P)\right)}{V^{a,b}(P)}.$$

By applying the Chernoff bound, for any $v \geq 0$ and any $\lambda < 0$,

$$\mathbb{P}_P\left(\widehat{\Delta}_T^{\mathrm{HIR},a,b} - \Delta^{a,b}(P) \leq v\right) \leq \mathbb{E}_P\left[\exp\left(\lambda\sqrt{T}\xi_T^{a,b}(P)\right)\right]\exp\left(-\lambda Tv\right).$$

By applying the Taylor series expansion for $\log\mathbb{E}_P\left[\exp\left(\lambda\sqrt{T}\xi_T^{a,b}(P)\right)\right]$ around $\frac{\lambda}{\sqrt{T}} = 0$,

$$\log\mathbb{E}_P\left[\exp\left(\lambda\sqrt{T}\xi_T^{a,b}(P)\right)\right]$$
$$= \sqrt{T}\lambda\mathbb{E}_P\left[\xi_T^{a,b}(P)\right] + \frac{T\lambda^2}{2}\mathbb{E}_P\left[\left(\xi_T^{a,b}(P)\right)^2\right] + \sum_{n=3}^{\infty}\frac{(\sqrt{T}\lambda)^n}{n!}c_{n,T},$$

where $c_{n,T}$ is the $n$-th cumulant of $(\xi_T^{a,b}(P)$. From Lemma 2.1 of Hayashi (2000) (Proposition B.8 in Appendix), Proposition 5.1, we have $\lim_{T\to\infty}\mathbb{E}\left[\xi_T^{a,b}(P)\right] = 0$, $\lim_{T\to\infty}\mathbb{E}\left[\left(\xi_T^{a,b}(P)\right)^2\right] = 1$, and $\lim_{T\to\infty}\mathbb{E}\left[\left(\xi_T^{a,b}(P)\right)^n\right] = m_n$ for all $n \geq 3$, where $m_n$ is the $n$-th moments. Then, we have $\lim_{T\to\infty}c_{n,T} = 0$ because cumulants of centered normal distributions are zero except for the second-order cumulant. Here, note that $\lim_{T\to\infty}\sum_{n=3}^{\infty}\frac{(\sqrt{T}\lambda)^{n-2}}{n!} = -\frac{1}{2}$. Therefore, for any $v, \varepsilon > 0$, there exist $T_0 > 0$ such that for all $T > T_0$,

$$\mathbb{P}_P\left(\sum_{t=1}^{T}\xi_t^{a,b}(P) \leq v\right) \leq \exp\left(\frac{T\lambda^2}{2} - T\lambda v - \left\{\sqrt{T}\lambda + T\lambda^2/2\right\}\varepsilon\right).$$

By substituting $\lambda = v = -\frac{\Delta^{a,b}(P)}{\sqrt{V^{a,b}(P)}} < 0$, the claim follows. $\square$

## H. Proof of Theorem 5.4

We follow the proof of Corollary 3 of Bubeck et al. (2011). First, from Theorem 5.3,

$$\mathbb{P}_P\left(\widehat{\mu}_T^{\mathrm{HIR},a} \leq \widehat{\mu}_T^{\mathrm{HIR},b}\right) \leq \exp\left(-\frac{T(\Delta^{a,b})^2}{2V^{a,b}(P)} + \left\{\frac{\sqrt{T}\Delta^{a,b}}{\sqrt{V^{a,b}(P)}} + \frac{T(\Delta^a(P))^2}{2V^a(P)}\right\}\varepsilon\right).$$

We consider two cases where a given $\Delta^a$ is more or less than a threshold $\ell^1, \ell^2, \dots, \ell^K > 0$. We have

$$\mathbb{E}_P \left[ r_T \left( \pi^{\mathrm{HIR}} \right) (P) \right] = \sum_{a \in [K]} \Delta^a \mathbb{P}_P \left( \widehat{\mu}_T^{\mathrm{HIR}, a^*(P)} \leq \widehat{\mu}_T^{\mathrm{HIR}, a} \right)$$

$$\leq \sum_{a \in [K]} \left\{ \ell^a \mathbb{P}_P \left( \widehat{\mu}_T^{\mathrm{HIR}, a^*(P)} \leq \widehat{\mu}_T^{\mathrm{HIR}, a} \right) + \mathbb{1}[\Delta^a \geq \ell^a] \Delta^a \mathbb{P}_P \left( \widehat{\mu}_T^{\mathrm{HIR}, a^*(P)} \leq \widehat{\mu}_T^{\mathrm{HIR}, a} \right) \right\}$$

$$\leq \max_{a \in [K]} \ell^a + \sum_{a \in [K]} \left\{ \mathbb{1}[\Delta^a \geq \ell^a] \Delta^a \mathbb{P}_P \left( \widehat{\mu}_T^{\mathrm{HIR}, a^*(P)} \leq \widehat{\mu}_T^{\mathrm{HIR}, a} \right) \right\}$$

Because $x \in [0, C_\Delta] \mapsto z \exp(-Cz^2)$ is decreasing on $[1/\sqrt{2C}, C_\Delta]$, for any $C > 0$ and $C_\Delta$, where $\Delta^a < C_\Delta$ for all $a \in [K]$. Therefore, taking $C = \lfloor \frac{T}{2V^a(P)} \rfloor$, for $\ell^a \geq 1/\sqrt{2 \lfloor \frac{T}{2V^a(P)} \rfloor}$,

$$\mathbb{E}_P \left[ r_T \left( \pi^{\mathrm{HIR}} \right) (P) \right]$$

$$\leq \max_{a \in [K]} \ell^a + \sum_{a \in [K]} \ell^a \exp \left( -\frac{T(\ell^a)^2}{2V^a(P)} + \left\{ \frac{\sqrt{T}\ell^a}{\sqrt{V^a(P)}} + \frac{T(\ell^a)^2}{2V^a(P)} \right\} \varepsilon \right)$$

$$\leq \max_{a \in [K]} \ell^a + (K-1) \max_{a \in [K]} \ell^a \exp \left( -\frac{T(\ell^a)^2}{2V^a(P)} + \left\{ \frac{\sqrt{T}\ell^a}{\sqrt{V^a(P)}} + \frac{T(\ell^a)^2}{2V^a(P)} \right\} \varepsilon \right)$$

$$\leq \max_{a \in [K]} \left\{ \ell^a + (K-1)\ell^a \exp \left( -\frac{T(\ell^a)^2}{2V^a(P)} + \left\{ \frac{\sqrt{T}\ell^a}{\sqrt{V^a(P)}} + \frac{T(\ell^a)^2}{2V^a(P)} \right\} \varepsilon \right) \right\}.$$

Substituting $\ell^a = \sqrt{\log K / \lfloor \frac{T}{2V^a(P)} \rfloor}$, we have

$$\mathbb{E}_P \left[ r_T \left( \pi^{\mathrm{HIR}} \right) (P) \right] \leq \max_{a \in [K]} \left\{ \sqrt{\log K / \left\lfloor \frac{T}{2V^a(P)} \right\rfloor} \right.$$

$$+ (K-1) \sqrt{\log K / \left\lfloor \frac{T}{2V^a(P)} \right\rfloor} \exp \left( -\frac{T \log K / \left\lfloor \frac{T}{2V^a(P)} \right\rfloor}{2V^a(P)} \right)$$

$$\left. \times \exp \left( \left\{ \frac{\sqrt{T \log K / \lfloor \frac{T}{2V^a(P)} \rfloor}}{\sqrt{V^a(P)}} + \frac{T \log K / \lfloor \frac{T}{2V^a(P)} \rfloor}{2V^a(P)} \right\} \varepsilon \right) \right\}.$$

Letting $T \to \infty$ and $\varepsilon \to 0$, we conclude the proof.

## I. Details of the AS-AIPW Strategy

. For $a \in [K]$ and $t \in [T]$, let $\widehat{w}_t(a|x)$ be an estimated target allocation ratio at round $t$. In each round $t$, we obtain $\gamma_t$ from the uniform distribution on $[0, 1]$ and draw a treatment arm $A_t = 1$ if $\gamma_t \leq \widehat{w}_t(1|X_t)$ and $A_t = a$ for $a \geq 2$ if $\gamma_t \in (\sum_{b=1}^{a-1} \widehat{w}_t(b|X_t), \sum_{b=1}^{a} \widehat{w}_t(b|X_t)]$; that is, we draw a treatment arm $a$ with a probability $\widehat{w}_t(a|X_t)$. As an initialization, we draw a treatment arm $A_t$ at round $t \leq K$ with an uniform probability; that is $\widehat{w}_t(a|X_t) = 1/K$ for all $a \in [K]$. In a round $t > K$, for all $a \in [K]$, we estimate the target allocation ratio $w^*$ using past observations $\mathcal{F}_{t-1}$, such that for all $a \in [K]$ and $x \in \mathcal{X}$, $\widehat{w}_t(a|x) > 0$ and $\sum_{a \in [K]} \widehat{w}_t(a|x) = 1$. We show a pseudo-code in Algorithm 1.

To construct an estimator $\widehat{w}_t(a|x)$ for all $x \in \mathcal{X}$ in each round $t$, we use bounded estimators of the conditional expected outcome $\mu^a(P)(x)$ and $\nu^a(P)(x)$, denoted by $\widehat{\mu}_t^a(x)$ and $\widehat{\nu}_t^a(x)$, respectively. We denote an estimator of the conditional variance $(\sigma^a(x))^2$ by $(\widehat{\sigma}_t^a(x))^2$, which are estimated as follows. For $t = 1, 2, \dots, K$, we set $\widehat{\mu}_t^a = \widehat{\nu}_t^a = (\widehat{\sigma}_t^a(x))^2 = 0$. For $t > K$, we estimate $\mu^a(P)(x)$ and $\nu^a(P)(x)$ using only past samples $\mathcal{F}_{t-1}$ and converge to the true parameter almost surely. We use a bounded estimator for $\widehat{\mu}_t^a$ such that $|\widehat{\mu}_t^a| < C_\mu$. Let $(\widehat{\sigma}_t^{\dagger a}(x))^2 = \widehat{\nu}_t^a(x) - (\widehat{\mu}_t^a(x))^2$ for all $a \in [K]$ and $x \in \mathcal{X}$. Then, we estimate the variance $(\sigma^a(x))^2$ for all $a \in [K]$ and $x \in \mathcal{X}$ in a round $t$ as $(\widehat{\sigma}_t^a(x))^2 =$

$\max\{\min\{((\widehat{\sigma}_t^{\dagger a}(x))^2, C_{\sigma^2}\}, 1/C_{\sigma^2}\}$ and define $\widehat{w}_t$ by replacing the variances in $w^*$ with corresponding estimators; that is, when $K = 2$, for each $a \in \{1, 2\}$, $\widehat{w}(a|X_t) = \frac{\widehat{\sigma}_t^1(X_t)}{\widehat{\sigma}_t^1(X_t) + \widehat{\sigma}_t^2(X_t)}$; when $K \geq 3$, for each $a \in [K]$, $\widehat{w}(a|X_t) = \frac{(\widehat{\sigma}_t^a(x))^2}{\sum_{b \in [K]}(\widehat{\sigma}_t^b(x))^2}$.

nonparametric estimators, such as the nearest neighbor regression estimator and kernel regression estimator, can be applied, which have been proven to converge to the true function in probability under a bounded sampling probability $\widehat{w}_t$ by Yang & Zhu (2002) and Qian & Yang (2016). Provided that these conditions are satisfied, any estimator can be used. It should be noted that we do not assume specific convergence rates for estimators for $\mu^a(P)(x)$ and $w^*$ as the asymptotic optimality of the AIPW estimator can be demonstrated without them (van der Laan, 2008; Kato et al., 2020; 2021).

The following part presents our recommendation rule. In the recommendation phase of round $T$, for each $a \in [K]$, we estimate $\mu^a$ for each $a \in [K]$ and recommend the maximum. To estimate $\mu^a$, the AIPW estimator is defined as

$$\widehat{\mu}_T^{\mathrm{AIPW},a} = \frac{1}{T} \sum_{t=1}^T \varphi_t^a\Big(Y_t, A_t, X_t\Big), \tag{9}$$

$$\varphi_t^a(Y_t, A_t, X_t) = \frac{\mathbb{1}[A_t = a]\big(Y_t^a - \widehat{\mu}_t^a(X_t)\big)}{\widehat{w}_t(a|X_t)} + \widehat{\mu}_t^a(X_t).$$

In the final round $t = T$, we recommend $\widehat{a}_T \in [K]$ as

$$\widehat{a}_T = \arg\max_{a \in [K]} \widehat{\mu}_T^{\mathrm{AIPW},a}. \tag{10}$$

The AIPW estimator debiases the sample selection bias resulting from arm draws based on contextual information. Additionally, the AIPW estimator possesses the following properties: (i) its components $\varphi_t^a(Y_t, A_t, X_t)_{t=1}^T$ are a martingale difference sequence, allowing us to employ the martingale limit theorems in derivation of the upper bound; (ii) it has the minimal asymptotic variance among the possible estimators. For example, other estimators with a martingale property, such as the inverse probability weighting (IPW) estimator, may be employed, yet their asymptotic variance would be greater than that of the AIPW estimator. The $t$-th element of the sum in the AIPW estimator utilizes nuisance parameters ($\mu^a(P)(x)$ and $w^*$) estimated from past observations up to round $t - 1$ for constructing a martingale difference sequence (van der Laan, 2008; Hadad et al., 2021; Kato et al., 2020; 2021). A pseudo-code for this process is provided in Algorithm 1.

The AS-AIPW strategy constitutes a generalization of the Neyman allocation (Neyman, 1934), which has been utilized for the efficient estimation of the ATE with two treatment arms (van der Laan, 2008; Hahn et al., 2011; Tabord-Meehan, 2022; Kato et al., 2020)[5] and two-armed fixed-budget BAI without contextual information (Glynn & Juneja, 2004; Kaufmann et al., 2016; Adusumilli, 2022; Armstrong, 2022). For two treatment arms, Adusumilli (2022) demonstrates the minimax optimality for the expected simple regret under the limit-of-experiment framework, utilizing a diffusion process framework. Armstrong (2022) also analyzes the minimax optimal strategy under the limit-of-experiment framework and establishes that the Neyman allocation is minimax optimal. Glynn & Juneja (2004) and Kaufmann et al. (2016) respectively illustrate the asymptotic optimality for each $P \in \mathcal{P}$ when the standard deviations of the potential outcomes are known.

Kato & Imaizumi (2023) summarizes and introduces our result using the AS-AIPW strategy, focusing on the probability of misidentification. They simplify the proof and explain the details of the strategy.

### I.1. Asymptotic Minimax Optimality of the AS-AIPW Strategy

We next present an asymptotic worst-case upper bound of our proposed AS-AIPW strategy. It is enough to provide the asymptotic normality. Under the asymptotic normality, the minimax optimality direct holds from the same procedure as that in the TS-HIR strategy.

For $a, b \in [K]$, define[6]

$$\xi_t^{a,b}(P) = \frac{\varphi_t^a(Y_t, A_t, X_t) - \varphi^b(Y_t, A_t, X_t) - \Delta^b(P)}{\sqrt{TV^{a,b*}(P)}},$$

---

[5]The AS-AIPW strategy is also similar to those proposed for efficient estimation of the ATE with multiple treatment arms (van der Laan, 2008).

[6]More rigorously, $\xi_t^{a,b}(P)$ should be denoted as double arrays such as $\xi_{Tt}^{a,b}(P)$ because they dependent on $T$. However, we omit the subscript $T$ for simplicity.

---

**Algorithm 2** AS-AIPW strategy.

---

**Parameter:** Positive constants $C_\mu$ and $C_{\sigma^2}$.
**Initialization:**
**for** $t = 1$ to $K$ **do**
    Draw $A_t = t$. For each $a \in [K]$, set $\widehat{w}_t(a|x) = 1/K$.
**end for**
**for** $t = K + 1$ to $T$ **do**
    Observe $X_t$.
    Construct $\widehat{w}_t(1|X_t)$ by using the estimators of the variances.
    Draw $\gamma_t$ from the uniform distribution on $[0, 1]$.
    $A_t = 1$ if $\gamma_t \le \widehat{w}_t(1|X_t)$ and $A_t = a$ for $a \ge 2$ if $\gamma_t \in \left( \sum_{b=1}^{a-1} \widehat{w}_t(b|X_t), \sum_{b=1}^{a} \widehat{w}_t(b|X_t) \right]$.
**end for**
Construct $\widehat{\mu}_T^{\mathrm{AIPW},a}$ for each $a \in [K]$ following (9).
Recommend $\widehat{a}_T$ following (10).

---

where

$$V^{a,b*}(P) = \mathbb{E}_P \left[ \frac{(\sigma^a(X))^2}{w^*(a|X)} + \frac{\left(\sigma^b(X)\right)^2}{w^*(b|X)} + \left( \Delta^{a,b}(P)(X) - \Delta^{a,b}(P) \right)^2 \right].$$

Here, note that $\{\xi_t^{a,b}(P)\}_{t=1}^T$ is a martingale difference sequence because $\mathbb{E}_P[\xi_t^{a,b}(P)|\mathcal{F}_{t-1}] = 0$ from

$$\mathbb{E}_P \left[ \frac{\mathbb{1}[A_t = a]\left(Y_t^a - \widehat{\mu}_t^a(X_t)\right)}{\widehat{w}_t(a|X_t)} + \widehat{\mu}_t^a(X_t)|X_t, \mathcal{F}_{t-1} \right] = \frac{\widehat{w}_t(a|X_t)\left(\mu^a(X_t)(P) - \widehat{\mu}_t^a(X_t)\right)}{\widehat{w}_t(a|X_t)} + \widehat{\mu}_t^a(X_t) = \mu^a(X_t)(P),$$

$$\mathbb{E}_P[\xi_t^{a,b}(P)|\mathcal{F}_{t-1}] = \int \left\{ \frac{\mu^a(x)(P) - \mu^b(x)(P) - \Delta^b(P)}{\sqrt{T V^{a,b*}(P)}} \right\} \zeta(x) \mathrm{d}x = 0.$$

We put the following assumption, which hold for a wide class of estimators (Yang & Zhu, 2002; Qian & Yang, 2016).

**Assumption I.1.** *For all $P \in \mathcal{P}^*$, $a \in [K]$, as $t \to \infty$,*

$$\widehat{\mu}_t^a(x) - \mu^a(P)(x) = o_P(1), \ \widehat{\nu}_t^a(x) - \nu^a(P)(x) = o_P(1).$$

Let $\widehat{\Delta}_T^{\mathrm{AIPW},a,b} := \widehat{\mu}_T^{a,\mathrm{AIPW}} - \widehat{\mu}_T^{b,\mathrm{AIPW}}$. Then, by directly applying the martingale CLT, we obtain the following theorem.

**Theorem I.2.** *Under Assumption I.1 and the AS-AIPW strategy,*

$$\sqrt{T} \left( \widehat{\Delta}_T^{\mathrm{AIPW},a,b} - \Delta^{a,b}(P) \right) \xrightarrow{\mathrm{d}} \mathcal{N} \left( 0, V^{a,b}(P) \right),$$

*where recall that*

$$V^{a,b}(P) = \mathbb{E}^X \left[ \frac{(\sigma^a(P)(X))^2}{w^*(a|X)} + \frac{\left(\sigma^b(P)(X)\right)^2}{w^*(b|X)} + \left( \Delta^{a,b}(P)(X) - \Delta^{a,b}(P) \right)^2 \right].$$

For the TS-HIR strategy, we prove the asymptotic optimality for the expected simple regret as follows:

1. Confirm the asymptotic normality of the HIR estimator.

2. Derive the upper bound for the probability of misidentification of the TS-HIR strategy by using the asymptotic normality of the HIR estimator.

3. Derive the upper bound for the expected simple regret of the TS-HIR strategy for each $P \in \mathcal{P}^*$.

4. Obtain the worst-case upper bound for the expected simple regret of the TS-HIR strategy.

We show the worst-case asymptotic optimality for the expeted simple regret of the AS-AIPW strategy as well as the TS-HIR strategy.

Here, unlike the TS-HIR strategy, which requires empirical process technique to show the asymptotic normality (Hirano et al., 2003; Hahn et al., 2011), we can skip the proof of the asymptotic normality to obtain the upper bound for the probability of misidentification of the AS-AIPW strategy. By employing martingale difference sequence arguments and the Chernoff bound, we can directly obtain the upper bound for the probability of misidentification, which greatly simplify the proof procedure. We show the proof in Appendix J.

### I.2. Proof of Theorem I.2

Note that $\sum_{t=1}^{T} \xi_t^{a,b}(P) = \sqrt{T} \left( \widehat{\mu}_T^{a,\text{AIPW}} - \widehat{\mu}_T^{b,\text{AIPW}} - \Delta^{a,b}(P) \right) / \sqrt{V^{a,b}(P)}$.

To prove Theorem I.2, we show the following lemma.

**Lemma I.3.** *Under Assumption I.1 and the AS-AIPW strategy, the following properties hold:*

**(a)** $\sum_{t=1}^{T} \mathbb{E}_P[(\xi_t^{a,b}(P))^2] \to 1$, *a positive value;*

**(b)** $\mathbb{E}_P[|\sqrt{T}\xi_t^{a,b}(P)|^r] < \infty$ *for some $r > 2$ and for all $t \in \mathbb{N}$;*

**(c)** $\sum_{t=1}^{T} (\xi_t^{a,b}(P))^2 \xrightarrow{p} 1$.

Proof of this lemma is shown in Appendix I.3. Then, we prove Theorem I.2.

*Proof.* The second statement directly holds from and large deviation bound. Therefore, we focus on the proof of the first statement, $\mathbb{P}\left( \widehat{\mu}_T^{a,\text{AIPW}} - \widehat{\mu}_T^{c,\text{AIPW}} \le 0 \right) - \exp\left( -\frac{T\left(\Delta^{a,b}(P)\right)^2}{V^{a,b*}(P)} \right) \le o(1)$. This inequality follows from the martingale CLT of White (1984) (Proposition B.6) on $\sum_{t=1}^{T} \xi_t^{a,b}(P)$ because

$$\mathbb{P}\left( \widehat{\mu}_T^{a,\text{AIPW}} - \widehat{\mu}_T^{b,\text{AIPW}} \le 0 \right) = \mathbb{P}\left( \sqrt{T}\left( \widehat{\mu}_T^{a,\text{AIPW}} - \widehat{\mu}_T^{b,\text{AIPW}} \right) - \sqrt{T}\Delta^{a,b}(P) \le -\sqrt{T}\Delta^{a,b}(P) \right)$$

$$= \mathbb{P}\left( \sqrt{\frac{T}{V^{a,b*}(P)}} \left( \widehat{\mu}_T^{a,\text{AIPW}} - \widehat{\mu}_T^{b,\text{AIPW}} - \Delta^{a,b}(P) \right) \le -\sqrt{\frac{T}{V^{a,b*}(P)}}\Delta^{a,b}(P) \right)$$

$$= \mathbb{P}\left( \sum_{t=1}^{T} \xi_t^{a,b}(P) \le -\sqrt{\frac{T}{V^{a,b*}(P)}}\Delta^{a,b}(P) \right).$$

Thus, we are interested in $= \mathbb{P}\left( \sum_{t=1}^{T} \xi_t^{a,b}(P) \le -\sqrt{\frac{T}{V^{a,b*}(P)}}\Delta^{a,b}(P) \right)$ and show the bound by using the martingale CLT.

Under the following three conditions, we can apply the martingale CLT,

**(a)** $\sum_{t=1}^{T} \mathbb{E}_P[(\xi_t^{a,b}(P))^2] \to 1$, a positive value;

**(b)** $\mathbb{E}_P[|\sqrt{T}\xi_t^{a,b}(P)|^r] < \infty$ for some $r > 2$ and for all $t \in \mathbb{N}$;

**(c)** $\sum_{t=1}^{T} (\xi_t^{a,b}(P))^2 \xrightarrow{p} 1$.

By using the martingale CLT, as $T \to \infty$,

$$\sum_{t=1}^{T} \xi_t^{a,b}(P) \xrightarrow{d} \mathcal{N}(0,1).$$

$\square$

### I.3. Proof of Lemma I.3

We prove Lemma I.3. Our proof is inspired by Kato et al. (2020).

*Proof.* Recall that

$$\varphi_t^a(Y_t, A_t, X_t) = \frac{\mathbb{1}[A_t = a](Y_t^a - \widehat{\mu}_t^a(X_t))}{\widehat{w}_t(a|X_t)} + \widehat{\mu}_t^a(X_t)$$

$$\xi_t^{a,b}(P) = \frac{\varphi_t^a(Y_t, A_t, X_t) - \varphi^{a,b}(Y_t, A_t, X_t) - \Delta^b(P)}{\sqrt{TV^{a,b*}(P)}}.$$

### Step 1: Check of Condition (a)

Because $\sqrt{TV^{a,b*}(P)}$ is non-random variable, we consider the conditional expectation of $\varphi_t^a(Y_t, A_t, X_t) - \varphi^{a,b}(Y_t, A_t, X_t) - \Delta^{a,b}(P)$.

First, we rewrite $\mathbb{E}_P\big[(\xi_t^{a,b}(P))^2\big]$ as

$$\mathbb{E}_P\big[(\xi_t^{a,b}(P))^2\big]$$
$$= \frac{1}{TV^{a,b*}(P)}\mathbb{E}_P\left[\left(\frac{\mathbb{1}[A_t = a](Y_t - \widehat{\mu}_t^a(X_t))}{\widehat{w}_t(a|X_t)} - \frac{\mathbb{1}[A_t = b](Y_t - \widehat{\mu}_t^b(X_t))}{\widehat{w}_t(b|X_t)} + \widehat{\mu}_t^a(X_t) - \widehat{\mu}_t^b(X_t) - \Delta^{a,b}(P)\right)^2\right]$$
$$= \frac{1}{TV^{a,b*}(P)}\mathbb{E}_P\left[\left(\frac{\mathbb{1}[A_t = a](Y_t^a - \widehat{\mu}_t^a(X_t))}{\widehat{w}_t(a|X_t)} - \frac{\mathbb{1}[A_t = b](Y_t^b - \widehat{\mu}_t^b(X_t))}{\widehat{w}_t(b|X_t)} + \widehat{\mu}_t^a(X_t) - \widehat{\mu}_t^b(X_t) - \Delta^{a,b}(P)\right)^2\right]$$
$$= \frac{1}{TV^{a,b*}(P)}\mathbb{E}_P\left[\left(\frac{\mathbb{1}[A_t = a](Y_t^a - \widehat{\mu}_t^a(X_t))}{\widehat{w}_t(a|X_t)} - \frac{\mathbb{1}[A_t = b](Y_t^b - \widehat{\mu}_t^b(X_t))}{\widehat{w}_t(b|X_t)} + \widehat{\mu}_t^a(X_t) - \widehat{\mu}_t^b(X_t) - \Delta^{a,b}(P)\right)^2\right]$$
$$- \frac{1}{TV^{a,b*}(P)}\mathbb{E}_P\left[\frac{\big(Y_t^a - \mu^a(P)(X_t)\big)^2}{w^*(a|X_t)} + \frac{\big(Y_t^b - \mu^b(P)(X_t)\big)^2}{w^*(b|X_t)} + \big(\mu^a(P)(X_t) - \mu^b(P)(X_t) - \Delta^{a,b}(P)\big)^2\right]$$
$$+ \frac{1}{TV^{a,b*}(P)}\mathbb{E}_P\left[\frac{\big(Y_t^a - \mu^a(P)(X_t)\big)^2}{w^*(a|X_t)} + \frac{\big(Y_t^b - \mu^b(P)(X_t)\big)^2}{w^*(b|X_t)} + \big(\mu^a(P)(X_t) - \mu^b(P)(X_t) - \Delta^{a,b}(P)\big)^2\right].$$

Therefore, we prove that the RHS of the following equation varnishes asymptotically to show that the condition (a) holds.

$$\mathbb{E}_P\big[(\xi_t^{a,b}(P))^2\big] - \frac{1}{TV^{a,b*}(P)}\mathbb{E}_P\left[\frac{\big(Y_t^a - \mu^a(P)(X_t)\big)^2}{w^*(a|X_t)} + \frac{\big(Y_t^b - \mu^b(P)(X_t)\big)^2}{w^*(b|X_t)} + \big(\mu^a(P)(X_t) - \mu^b(P)(X_t) - \Delta^{a,b}(P)\big)^2\right]$$
$$= \frac{1}{TV^{a,b*}(P)}\mathbb{E}_P\left[\left(\frac{\mathbb{1}[A_t = a](Y_t^a - \widehat{\mu}_t^a(X_t))}{\widehat{w}_t(a|X_t)} - \frac{\mathbb{1}[A_t = b](Y_t^b - \widehat{\mu}_t^b(X_t))}{\widehat{w}_t(b|X_t)} + \widehat{\mu}_t^a(X_t) - \widehat{\mu}_t^b(X_t) - \Delta^{a,b}(P)\right)^2\right]$$
$$- \frac{1}{TV^{a,b*}(P)}\mathbb{E}_P\left[\frac{\big(Y_t^a - \mu^a(P)(X_t)\big)^2}{w^*(a|X_t)} + \frac{\big(Y_t^b - \mu^b(P)(X_t)\big)^2}{w^*(b|X_t)} + \big(\mu^a(P)(X_t) - \mu^b(P)(X_t) - \Delta^{a,b}(P)\big)^2\right]. \quad (11)$$

For simplicity, we omit $\frac{1}{TV^{a,b*}(P)}$. For the first term of the RHS of (11),

$$\mathbb{E}_P\left[\left(\frac{\mathbb{1}[A_t = a](Y_t^a - \widehat{\mu}_t^a(X_t))}{\widehat{w}_t(a|X_t)} - \frac{\mathbb{1}[A_t = b](Y_t^b - \widehat{\mu}_t^b(X_t))}{\widehat{w}_t(b|X_t)} + \widehat{\mu}_t^a(X_t) - \widehat{\mu}_t^b(X_t) - \Delta^{a,b}(P)\right)^2\right]$$
$$= \mathbb{E}_P\left[\left(\frac{\mathbb{1}[A_t = a](Y_t^a - \widehat{\mu}_t^a(X_t))}{\widehat{w}_t(a|X_t)}\right)^2\right]$$

$$+ \mathbb{E}_P \left[ \left( \frac{\mathbb{1}[A_t = b]\left(Y_t^b - \widehat{\mu}_t^b(X_t)\right)}{\widehat{w}_t(b|X_t)} \right)^2 \right]$$

$$+ \mathbb{E}_P \left[ \left( \widehat{\mu}_t^a(X_t) - \widehat{\mu}_t^b(X_t) - \Delta^{a,b}(P) \right)^2 \right]$$

$$- 2\mathbb{E}_P \left[ \left( \frac{\mathbb{1}[A_t = a]\left(Y_t^a - \widehat{\mu}_t^a(X_t)\right)}{\widehat{w}_t(a|X_t)} \right) \left( \frac{\mathbb{1}[A_t = b]\left(Y_t^b - \widehat{\mu}_t^b(X_t)\right)}{\widehat{w}_t(b|X_t)} \right) \right]$$

$$+ 2\mathbb{E}_P \left[ \left( \frac{\mathbb{1}[A_t = a]\left(Y_t^a - \widehat{\mu}_t^a(X_t)\right)}{\widehat{w}_t(a|X_t)} \right) \left( \widehat{\mu}_t^a(X_t) - \widehat{\mu}_t^b(X_t) - \Delta^{a,b}(P) \right) \right]$$

$$- 2\mathbb{E}_P \left[ \left( \frac{\mathbb{1}[A_t = b]\left(Y_t^b - \widehat{\mu}_t^b(X_t)\right)}{\widehat{w}_t(b|X_t)} \right) \left( \widehat{\mu}_t^a(X_t) - \widehat{\mu}_t^b(X_t) - \Delta^{a,b}(P) \right) \right].$$

Because $\mathbb{1}[A_t = a]\mathbb{1}[A_t = b] = 0$, $\mathbb{1}[A_t = k]\mathbb{1}[A_t = k] = \mathbb{1}[A_t = k]$ for $k \in \{a, b\}$, we have

$$\mathbb{E}_P \left[ \left( \frac{\mathbb{1}[A_t = k]\left(Y_t^k - \widehat{\mu}_t^k(X_t)\right)}{\widehat{w}_t(k|X_t)} \right)^2 \right] = \mathbb{E}_P \left[ \frac{\left(Y_t^k - \widehat{\mu}_t^k(X_t)\right)^2}{\widehat{w}_t(k|X_t)} \right],$$

$$\mathbb{E}_P \left[ \left( \frac{\mathbb{1}[A_t = a]\left(Y_t^a - \widehat{\mu}_t^a(X_t)\right)}{\widehat{w}_t(a|X_t)} \right) \left( \frac{\mathbb{1}[A_t = b]\left(Y_t^b - \widehat{\mu}_t^b(X_t)\right)}{\widehat{w}_t(b|X_t)} \right) \right] = 0,$$

$$\mathbb{E}_P \left[ \left( \frac{\mathbb{1}[A_t = a]\left(Y_t - \widehat{\mu}_t^a(X_t)\right)}{\widehat{w}_t(a|X_t)} - \frac{\mathbb{1}[A_t = b]\left(Y_t - \widehat{\mu}_t^b(X_t)\right)}{\widehat{w}_t(b|X_t)} \right) \left( \widehat{\mu}_t^a(X_t) - \widehat{\mu}_t^b(X_t) - \Delta^{a,b}(P) \right) \right]$$

$$= \mathbb{E}_P \left[ \mathbb{E}_P \left[ \frac{\mathbb{1}[A_t = a]\left(Y_t - \widehat{\mu}_t^a(X_t)\right)}{\widehat{w}_t(a|X_t)} - \frac{\mathbb{1}[A_t = b]\left(Y_t - \widehat{\mu}_t^b(X_t)\right)}{\widehat{w}_t(b|X_t)} \mid X_t, \mathcal{F}_{t-1} \right] \left( \widehat{\mu}_t^a(X_t) - \widehat{\mu}_t^b(X_t) - \Delta^{a,b}(P) \right) \right]$$

$$= \mathbb{E}_P \left[ \left( \mu^a(P)(X_t) - \mu^b(P)(X_t) - \widehat{\mu}_t^a(X_t) + \widehat{\mu}_t^b(X_t) \right) \left( \widehat{\mu}_t^a(X_t) - \widehat{\mu}_t^b(X_t) - \Delta^{a,b}(P) \right) \right].$$

Therefore, for the first term of the RHS of (11),

$$\mathbb{E}_P \left[ \left( \frac{\mathbb{1}[A_t = a]\left(Y_t^a - \widehat{\mu}_t^a(X_t)\right)}{\widehat{w}_t(a|X_t)} - \frac{\mathbb{1}[A_t = b]\left(Y_t^b - \widehat{\mu}_t^b(X_t)\right)}{\widehat{w}_t(b|X_t)} + \widehat{\mu}_t^a(X_t) - \widehat{\mu}_t^b(X_t) - \Delta^{a,b}(P) \right)^2 \right]$$

$$= \mathbb{E}_P \left[ \frac{\left(Y_t^a - \widehat{\mu}_t^a(X_t)\right)^2}{\widehat{w}_t(a|X_t)} + \frac{\left(Y_t^b - \widehat{\mu}_t^b(X_t)\right)^2}{\widehat{w}_t(b|X_t)} + \left( \widehat{\mu}_t^a(X_t) - \widehat{\mu}_t^b(X_t) - \Delta^{a,b}(P) \right)^2 \right.$$

$$\left. + 2\left( \mu^a(P)(X_t) - \mu^b(P)(X_t) - \widehat{\mu}_t^a(X_t) + \widehat{\mu}_t^b(X_t) \right) \left( \widehat{\mu}_t^a(X_t) - \widehat{\mu}_t^b(X_t) - \Delta^{a,b}(P) \right) \right].$$

Then, using these equations, the RHS of (11) can be calculated as

$$\frac{1}{TV^{a,b*}(P)} \mathbb{E}_P \left[ \left( \frac{\mathbb{1}[A_t = a]\left(Y_t^a - \widehat{\mu}_t^a(X_t)\right)}{\widehat{w}_t(a|X_t)} - \frac{\mathbb{1}[A_t = b]\left(Y_t^b - \widehat{\mu}_t^b(X_t)\right)}{\widehat{w}_t(b|X_t)} + \widehat{\mu}_t^a(X_t) - \widehat{\mu}_t^b(X_t) - \Delta^{a,b}(P) \right)^2 \right]$$

$$- \frac{1}{TV^{a,b*}(P)} \mathbb{E}_P \left[ \frac{\left(Y_t^a - \mu^a(P)(X_t)\right)^2}{w^*(a|X_t)} + \frac{\left(Y_t^b - \mu^b(P)(X_t)\right)^2}{w^*(b|X_t)} + \left( \mu^a(P)(X_t) - \mu^b(P)(X_t) - \Delta^{a,b}(P) \right)^2 \right]$$

$$= \frac{1}{TV^{a,b*}(P)} \mathbb{E}_P \left[ \frac{\left(Y_t^a - \widehat{\mu}_t^a(X_t)\right)^2}{\widehat{w}_t(a|X_t)} + \frac{\left(Y_t^b - \widehat{\mu}_t^b(X_t)\right)^2}{\widehat{w}_t(b|X_t)} + \left( \widehat{\mu}_t^a(X_t) - \widehat{\mu}_t^b(X_t) - \Delta^{a,b}(P) \right)^2 \right.$$

$$\left. + 2\left( \mu^a(P)(X_t) - \mu^b(P)(X_t) - \widehat{\mu}_t^a(X_t) + \widehat{\mu}_t^b(X_t) \right) \left( \widehat{\mu}_t^a(X_t) - \widehat{\mu}_t^b(X_t) - \Delta^{a,b}(P) \right) \right]$$

$$- \frac{1}{TV^{a,b*}(P)} \mathbb{E}_P \left[ \frac{\left(Y_t^a - \mu^a(P)(X_t)\right)^2}{w^*(a|X_t)} + \frac{\left(Y_t^b - \mu^b(P)(X_t)\right)^2}{w^*(b|X_t)} + \left(\mu^a(P)(X_t) - \mu^b(P)(X_t) - \Delta^{a,b}(P)\right)^2 \right].$$

By taking the absolute value, we can bound the RHS as

$$\frac{1}{TV^{a,b*}(P)} \mathbb{E}_P \left[ \frac{\left(Y_t^a - \widehat{\mu}_t^a(X_t)\right)^2}{\widehat{w}_t(a|X_t)} + \frac{\left(Y_t^b - \widehat{\mu}_t^b(X_t)\right)^2}{\widehat{w}_t(b|X_t)} + \left(\widehat{\mu}_t^a(X_t) - \widehat{\mu}_t^b(X_t) - \Delta^{a,b}(P)\right)^2 \right.$$

$$\left. + 2\left(\mu^a(P)(X_t) - \mu^b(P)(X_t) - \widehat{\mu}_t^a(X_t) + \widehat{\mu}_t^b(X_t)\right)\left(\widehat{\mu}_t^a(X_t) - \widehat{\mu}_t^b(X_t) - \Delta^{a,b}(P)\right) \right]$$

$$- \frac{1}{TV^{a,b*}(P)} \mathbb{E}_P \left[ \frac{\left(Y_t^a - \mu^a(P)(X_t)\right)^2}{w^*(a|X_t)} + \frac{\left(Y_t^b - \mu^b(P)(X_t)\right)^2}{w^*(b|X_t)} + \left(\mu^a(P)(X_t) - \mu^b(P)(X_t) - \Delta^{a,b}(P)\right)^2 \right]$$

$$\leq \frac{1}{TV^{a,b*}(P)} \mathbb{E}_P \left[ \left| \left\{ \frac{\left(Y_t^a - \widehat{\mu}_t^a(X_t)\right)^2}{\widehat{w}_t(a|X_t)} + \frac{\left(Y_t^b - \widehat{\mu}_t^b(X_t)\right)^2}{\widehat{w}_t(b|X_t)} + \left(\widehat{\mu}_t^a(X_t) - \widehat{\mu}_t^b(X_t) - \Delta^{a,b}(P)\right)^2 \right. \right. \right.$$

$$\left. + 2\left(\mu^a(P)(X_t) - \mu^b(P)(X_t) - \widehat{\mu}_t^a(X_t) + \widehat{\mu}_t^b(X_t)\right)\left(\widehat{\mu}_t^a(X_t) - \widehat{\mu}_t^b(X_t) - \Delta^{a,b}(P)\right) \right\}$$

$$\left. \left. - \left\{ \frac{\left(Y_t^a - \mu^a(P)(X_t)\right)^2}{w^*(a|X_t)} + \frac{\left(Y_t^b - \mu^b(P)(X_t)\right)^2}{w^*(b|X_t)} + \left(\mu^a(P)(X_t) - \mu^b(P)(X_t) - \Delta^{a,b}(P)\right)^2 \right\} \right| \right].$$

Then, from the triangle inequality, we have

$$\mathbb{E}_P \left[ \left| \left\{ \frac{\left(Y_t^a - \widehat{\mu}_t^a(X_t)\right)^2}{\widehat{w}_t(a|X_t)} + \frac{\left(Y_t^b - \widehat{\mu}_t^b(X_t)\right)^2}{\widehat{w}_t(b|X_t)} + \left(\widehat{\mu}_t^a(X_t) - \widehat{\mu}_t^b(X_t) - \Delta^{a,b}(P)\right)^2 \right. \right. \right.$$

$$\left. + 2\left(\mu^a(P)(X_t) - \mu^b(P)(X_t) - \widehat{\mu}_t^a(X_t) + \widehat{\mu}_t^b(X_t)\right)\left(\widehat{\mu}_t^a(X_t) - \widehat{\mu}_t^b(X_t) - \Delta^{a,b}(P)\right) \right\}$$

$$\left. \left. - \left\{ \frac{\left(Y_t^a - \mu^a(P)(X_t)\right)^2}{w^*(a|X_t)} + \frac{\left(Y_t^b - \mu^b(P)(X_t)\right)^2}{w^*(b|X_t)} + \left(\mu^a(P)(X_t) - \mu^b(P)(X_t) - \Delta^{a,b}(P)\right)^2 \right\} \right| \right]$$

$$\leq \mathbb{E}_P \left[ \left| \frac{\left(Y_t^a - \widehat{\mu}_t^a(X_t)\right)^2}{\widehat{w}_t(a|X_t)} - \frac{\left(Y_t^a - \mu^a(P)(X_t)\right)^2}{w^*(a|X_t)} \right| \right] + \mathbb{E}_P \left[ \left| \frac{\left(Y_t^b - \widehat{\mu}_t^b(X_t)\right)^2}{\widehat{w}_t(b|X_t)} - \frac{\left(Y_t^b - \mu^b(P)(X_t)\right)^2}{w^*(b|X_t)} \right| \right]$$

$$+ \mathbb{E}_P \left[ \left| \left(\widehat{\mu}_t^a(X_t) - \widehat{\mu}_t^b(X_t) - \Delta^{a,b}(P)\right)^2 - \left(\mu^a(P)(X_t) - \mu^b(P)(X_t) - \Delta^{a,b}(P)\right)^2 \right| \right]$$

$$+ 2\mathbb{E}_P \left[ \left| \left(\mu^a(P)(X_t) - \mu^b(P)(X_t) - \widehat{\mu}_t^a(X_t) + \widehat{\mu}_t^b(X_t)\right)\left(\widehat{\mu}_t^a(X_t) - \widehat{\mu}_t^b(X_t) - \Delta^{a,b}(P)\right) \right| \right].$$

Because all elements are assumed to be bounded and $b_1^2 - b_2^2 = (b_1 + b_2)(b_1 - b_2)$ for variables $b_1$ and $b_2$, there exist constants $\tilde{C}_0$, $\tilde{C}_1$, $\tilde{C}_2$, and $\tilde{C}_3$ such that

$$\sum_{k \in \{a,b\}} \mathbb{E}_P \left[ \left| \frac{\left(Y_t^k - \widehat{\mu}_t^k(X_t)\right)^2}{\widehat{w}_t(k|X_t)} - \frac{\left(Y_t^k - \mu^k(P)(X_t)\right)^2}{w^*(k|X_t)} \right| \right]$$

$$+ \mathbb{E}_P \left[ \left| \left(\widehat{\mu}_t^a(X_t) - \widehat{\mu}_t^b(X_t) - \Delta^{a,b}(P)\right)^2 - \left(\mu^a(P)(X_t) - \mu^b(P)(X_t) - \Delta^{a,b}(P)\right)^2 \right| \right]$$

$$+ 2\mathbb{E}_P \left[ \left| \left(\mu^a(P)(X_t) - \mu^b(P)(X_t) - \widehat{\mu}_t^a(X_t) + \widehat{\mu}_t^b(X_t)\right)\left(\widehat{\mu}_t^a(X_t) - \widehat{\mu}_t^b(X_t) - \Delta^{a,b}(P)\right) \right| \right]$$

$$\leq \tilde{C}_0 \sum_{k\in\{a,b\}} \mathbb{E}_P\left[\left|\frac{(Y_t^k - \widehat{\mu}_t^k(X_t))}{\sqrt{\widehat{w}_t(k|X_t)}} - \frac{(Y_t^k - \mu^k(P)(X_t))}{\sqrt{w^*(k|X_t)}}\right|\right]$$

$$+ \mathbb{E}_P\left[\left|\left(\widehat{\mu}_t^a(X_t) - \widehat{\mu}_t^b(X_t) - \Delta^{a,b}(P)\right)^2 - \left(\mu^a(P)(X_t) - \mu^b(P)(X_t) - \Delta^{a,b}(P)\right)^2\right|\right]$$

$$+ 2\mathbb{E}_P\left[\left|\left(\mu^a(P)(X_t) - \mu^b(P)(X_t) - \widehat{\mu}_t^a(X_t) + \widehat{\mu}_t^b(X_t)\right)\left(\widehat{\mu}_t^a(X_t) - \widehat{\mu}_t^b(X_t) - \Delta^{a,b}(P)\right)\right|\right]$$

$$\leq \tilde{C}_1 \sum_{k\in\{a,b\}} \mathbb{E}_P\left[\left|\sqrt{w^*(k|X_t)}\left(Y_t - \widehat{\mu}_t^k(X_t)\right) - \sqrt{\widehat{w}_t(k|X_t)}\left(Y_t - \mu^k(P)(X_t)\right)\right|\right]$$

$$+ \mathbb{E}_P\left[\left|\left(\widehat{\mu}_t^a(X_t) - \widehat{\mu}_t^b(X_t) - \Delta^{a,b}(P)\right)^2 - \left(\mu^a(P)(X_t) - \mu^b(P)(X_t) - \Delta^{a,b}(P)\right)^2\right|\right]$$

$$+ 2\mathbb{E}_P\left[\left|\left(\mu^a(P)(X_t) - \mu^b(P)(X_t) - \widehat{\mu}_t^a(X_t) + \widehat{\mu}_t^b(X_t)\right)\left(\widehat{\mu}_t^a(X_t) - \widehat{\mu}_t^b(X_t) - \Delta^{a,b}(P)\right)\right|\right]$$

$$\leq \widetilde{C}_1 \sum_{k=0}^1 \mathbb{E}_P\left[\left|\sqrt{w^*(k|X_t)}\widehat{\mu}_t^k(X_t) - \sqrt{\widehat{w}_t(k|X_t)}\mu^k(P)(X_t)\right|\right]$$

$$+ \widetilde{C}_2 \sum_{k=0}^1 \mathbb{E}_P\left[\left|\sqrt{w^*(k|X_t)} - \sqrt{\widehat{w}_t(k|X_t)}\right|\right] + \widetilde{C}_3 \sum_{k=0}^1 \mathbb{E}_P\left[\left|\widehat{\mu}_t^k(X_t) - \mu^k(P)(X_t)\right|\right].$$

Then, from $b_1b_2 - b_3b_4 = (b_1 - b_3)b_4 - (b_4 - b_2)b_1$ for variables $b_1$, $b_2$, $b_3$, and $b_4$, there exist $\tilde{C}_4$ and $\tilde{C}_5$ such that

$$\widetilde{C}_1 \sum_{k\in\{a,b\}} \mathbb{E}_P\left[\left|\sqrt{w^*(k|X_t)}\widehat{\mu}_t^k(X_t) - \sqrt{\widehat{w}_t(k|X_t)}\mu^k(P)(X_t)\right|\right]$$

$$+ \widetilde{C}_2 \sum_{k\in\{a,b\}} \mathbb{E}_P\left[\left|\sqrt{w^*(k|X_t)} - \sqrt{\widehat{w}_t(k|X_t)}\right|\right] + \widetilde{C}_3 \sum_{k\in\{a,b\}} \mathbb{E}_P\left[\left|\widehat{\mu}_t^k(X_t) - \mu^k(P)(X_t)\right|\right]$$

$$\leq \widetilde{C}_4 \sum_{k\in\{a,b\}} \mathbb{E}_P\left[\left|\sqrt{w^*(k|X_t)} - \sqrt{\widehat{w}_t(k|X_t)}\right|\right] + \widetilde{C}_5 \sum_{k\in\{a,b\}} \mathbb{E}_P\left[\left|\widehat{\mu}_t^k(X_t) - \mu^k(P)(X_t)\right|\right].$$

From $\widehat{w}_t(k|x) - w^*(k|x) \xrightarrow{\mathrm{P}} 0$, we have $\sqrt{\widehat{w}_t(k|x)} - \sqrt{w^*(k|x)} \xrightarrow{\mathrm{P}} 0$. From the assumption that the point convergences in probability, i.e., for all $x \in \mathcal{X}$ and $k \in \mathcal{A}$, $\sqrt{\widehat{w}_t(k|x)} - \sqrt{w^*(k|x)} \xrightarrow{\mathrm{P}} 0$ and $\widehat{\mu}_t^k(x) - \mu^k(P)(x) \xrightarrow{\mathrm{P}} 0$ as $t \to \infty$, if $\sqrt{\widehat{w}_t(k|x)}$, and $\widehat{\mu}_t^k(x)$ are uniformly integrable, for fixed $x \in \mathcal{X}$, we can prove that

$$\mathbb{E}_P\left[|\sqrt{\widehat{w}_t(k|X_t)} - \sqrt{w^*(k|X_t)}||X_t = x\right] = \mathbb{E}_P\left[|\sqrt{\widehat{w}_t(k|x)} - \sqrt{w^*(k|x)}|\right] \to 0,$$
$$\mathbb{E}_P\left[|\widehat{\mu}_t^k(X_t) - \mu^k(P)(X_t)||X_t = x\right] = \mathbb{E}_P\left[|\widehat{\mu}_t^k(x) - \mu^k(P)(x)|\right] \to 0,$$

as $t \to \infty$ using $L^r$-convergence theorem (Proposition B.3). Here, we used the fact that $\widehat{\mu}_t^k(x)$ and $\sqrt{\widehat{w}_t(k|x)}$ are independent from $X_t$. For fixed $x \in \mathcal{X}$, we can show that $\sqrt{\widehat{w}_t(k|x)}$, and $\widehat{\mu}_t^k(x)$ are uniformly integrable from the boundedness of $\sqrt{\widehat{w}_t(k|x)}$, and $\widehat{\mu}_t^k(x)$ (Proposition B.2). From the point convergence of $\mathbb{E}[|\sqrt{\widehat{w}_t(k|X_t)} - \sqrt{w^*(k|X_t)}| \mid X_t = x]$ and $\mathbb{E}[|\widehat{\mu}_t^k(X_t) - \mu^k(P)(X_t)| \mid X_t = x]$, by using Lebesgue's dominated convergence theorem, we can show that

$$\mathbb{E}_{X_t}\left[\mathbb{E}_P\left[|\sqrt{\widehat{w}_t(k|X_t)} - \sqrt{w^*(k|X_t)}| \mid X_t\right]\right] \to 0,$$
$$\mathbb{E}_{X_t}\left[\mathbb{E}[|\widehat{\mu}_t^k(X_t) - \mu^k(P)(X_t)| \mid X_t]\right] \to 0.$$

Then, as $t \to \infty$,

$$TV^{a,b*}(P)\mathbb{E}_P\left[(\xi_t^{a,b}(P))^2\right] - \mathbb{E}_P\left[\sum_{k\in\{a,b\}} \frac{(Y_t^k - \mu^k(P)(X_t))^2}{w^*(k|X_t)} + \left(\mu^a(P)(X_t) - \mu^b(P)(X_t) - \Delta^{a,b}(P)\right)^2\right]$$

$\to 0$.

Therefore, for any $\epsilon > 0$, there exists $\tilde{t} > 0$ such that

$$\frac{1}{TV^{a,b*}(P)} \sum_{t=1}^{T} \left( TV^{a,b*}(P)\mathbb{E}_P\big[(\xi_t^{a,b}(P))^2\big] - \mathbb{E}_P\left[ \sum_{k\in\{a,b\}} \frac{\big(Y_t^k - \mu^k(P)(X_t)\big)^2}{w^*(k|X_t)} + \Big(\mu^a(P)(X_t) - \mu^b(P)(X_t) - \Delta^{a,b}(P)\Big)^2 \right] \right)$$

$$\leq \frac{\tilde{t}}{TV^{a,b*}(P)} + \epsilon.$$

Here,

$$\mathbb{E}_P\left[ \sum_{k\in\{a,b\}} \frac{\big(Y_t^k - \mu^k(P)(X_t)\big)^2}{w^*(k|X_t)} + \Big(\mu^a(P)(X_t) - \mu^b(P)(X_t) - \Delta^{a,b}(P)\Big)^2 \right]$$

$$= \mathbb{E}_P\left[ \sum_{k\in\{a,b\}} \frac{\big(Y^k - \mu^k(P)(X)\big)^2}{w^*(k|X_t)} + \Big(\mu^a(P)(X) - \mu^b(P)(X) - \Delta^{a,b}(P)\Big)^2 \right]$$

$$= V^{a,b*}(P)$$

does not depend on $t$. Therefore, $\sum_{t=1}^{T} \mathbb{E}_P\big[(\xi_t^{a,b}(P))^2\big] - 1 \leq \frac{\tilde{t}}{TV^{a,b*}(P)} + \epsilon \to 0$ as $T \to \infty$.

**Step 2: check of condition (b).**   We directly assumed that the condition holds from Definition 3.2.

**Step 3: Check of Condition (c)**

Let $u_t$ be an MDS such that

$$u_t = (\xi_t^{a,b}(P))^2 - \mathbb{E}_P\big[(\xi_t^{a,b}(P))^2|\mathcal{F}_{t-1}\big]$$

$$= \frac{1}{TV^{a,b*}(P)} \left( \frac{\mathbb{1}[A_t = a]\big(Y_t^a - \widehat{\mu}_t^a(X_t)\big)}{\widehat{w}_t(a|X_t)} - \frac{\mathbb{1}[A_t = b]\big(Y_t^b - \widehat{\mu}_t^b(X_t)\big)}{\widehat{w}_t(b|X_t)} + \widehat{\mu}_t^a(X_t) - \widehat{\mu}_t^b(X_t) - \Delta^{a,b}(P) \right)^2$$

$$- \frac{1}{TV^{a,b*}(P)} \mathbb{E}_P \left[ \left( \frac{\mathbb{1}[A_t = a]\big(Y_t^a - \widehat{\mu}_t^a(X_t)\big)}{\widehat{w}_t(a|X_t)} - \frac{\mathbb{1}[A_t = b]\big(Y_t^b - \widehat{\mu}_t^b(X_t)\big)}{\widehat{w}_t(b|X_t)} + \widehat{\mu}_t^a(X_t) - \widehat{\mu}_t^b(X_t) - \Delta^{a,b}(P) \right)^2 \Big| \mathcal{F}_{t-1} \right].$$

We can apply the weak law of large numbers for an MDS (Proposition B.5 in Appendix B), and obtain

$$\sum_{t=1}^{T} u_t = \frac{1}{T} \sum_{t=1}^{T} T\left( (\xi_t^{a,b}(P))^2 - \mathbb{E}_P\big[(\xi_t^{a,b}(P))^2|\mathcal{F}_{t-1}\big] \right) \xrightarrow{\text{P}} 0.$$

The conditions in the weak law of large numbers for an MDS (Proposition B.5) can be confirmed from Definition 3.2.

Next, we show that

$$\sum_{t=1}^{T} \mathbb{E}_P\big[(\xi_t^{a,b}(P))^2 \mid \mathcal{F}_{t-1}\big] - 1 \xrightarrow{\text{P}} 0.$$

From Markov's inequality, for $\varepsilon > 0$, we have

$$\mathbb{P}\left( \left| \sum_{t=1}^{T} \mathbb{E}_P\big[(\xi_t^{a,b}(P))^2 \mid \mathcal{F}_{t-1}\big] - 1 \right| \geq \varepsilon \right)$$

$$\leq \frac{\mathbb{E}_P\left[ \left| \sum_{t=1}^{T} \mathbb{E}_P\big[(\xi_t^{a,b}(P))^2 \mid \mathcal{F}_{t-1}\big] - 1 \right| \right]}{\varepsilon}$$

$$\leq \frac{\frac{1}{TV^{a,b*}(P)} \sum_{t=1}^{T} \mathbb{E}_P \left[ \left| TV^{a,b*}(P) \mathbb{E}_P \left[ (\xi_t^{a,b}(P))^2 \mid \mathcal{F}_{t-1} \right] - V^{a,b*}(P) \right| \right]}{\varepsilon}.$$

Then, we consider showing $\mathbb{E}_P \left[ \left| TV^{a,b*}(P) \mathbb{E}_P \left[ (\xi_t^{a,b}(P))^2 \mid \mathcal{F}_{t-1} \right] - V^{a,b*}(P) \right| \right] \to 0$. Here, we have

$$\mathbb{E}_P \left[ \left| TV^{a,b*}(P) \mathbb{E}_P \left[ (\xi_t^{a,b}(P))^2 \mid \mathcal{F}_{t-1} \right] - V^{a,b*}(P) \right| \right]$$

$$= \mathbb{E}_P \Bigg[ \Bigg| \mathbb{E}_P \Bigg[ \frac{\left(Y_t^a - \widehat{\mu}_t^a(X_t)\right)^2}{\widehat{w}_t(a|X_t)} + \frac{\left(Y_t^b - \widehat{\mu}_t^b(X_t)\right)^2}{\widehat{w}_t(b|X_t)} + \left( \widehat{\mu}_t^a(X_t) - \widehat{\mu}_t^b(X_t) - \Delta^{a,b}(P) \right)^2$$

$$+ 2 \left( \mu^a(P)(X_t) - \mu^b(P)(X_t) - \widehat{\mu}_t^a(X_t) + \widehat{\mu}_t^b(X_t) \right) \left( \widehat{\mu}_t^a(X_t) - \widehat{\mu}_t^b(X_t) - \Delta^{a,b}(P) \right)$$

$$- \frac{\left(Y_t^a - \mu^a(P)(X_t)\right)^2}{w^*(a|X_t)} - \frac{\left(Y_t^b - \mu^b(P)(X_t)\right)^2}{w^*(b|X_t)} - \left( \mu^a(P)(X_t) - \mu^b(P)(X_t) - \Delta^{a,b}(P) \right)^2 \mid \mathcal{F}_{t-1} \Bigg] \Bigg| \Bigg]$$

$$= \mathbb{E}_P \Bigg[ \Bigg| \mathbb{E}_P \Bigg[ \mathbb{E}_P \Bigg[ \frac{\left(Y_t^a - \widehat{\mu}_t^a(X_t)\right)^2}{\widehat{w}_t(a|X_t)} + \frac{\left(Y_t^b - \widehat{\mu}_t^b(X_t)\right)^2}{\widehat{w}_t(b|X_t)} + \left( \widehat{\mu}_t^a(X_t) - \widehat{\mu}_t^b(X_t) - \Delta^{a,b}(P) \right)^2$$

$$+ 2 \left( \mu^a(P)(X_t) - \mu^b(P)(X_t) - \widehat{\mu}_t^a(X_t) + \widehat{\mu}_t^b(X_t) \right) \left( \widehat{\mu}_t^a(X_t) - \widehat{\mu}_t^b(X_t) - \Delta^{a,b}(P) \right)$$

$$- \frac{\left(Y_t^a - \mu^a(P)(X_t)\right)^2}{w^*(a|X_t)} - \frac{\left(Y_t^b - \mu^b(P)(X_t)\right)^2}{w^*(b|X_t)} - \left( \mu^a(P)(X_t) - \mu^b(P)(X_t) - \Delta^{a,b}(P) \right)^2 \mid X_t, \mathcal{F}_{t-1} \Bigg] \mid \mathcal{F}_{t-1} \Bigg] \Bigg| \Bigg].$$

Then, by using Jensen's inequality,

$$\mathbb{E}_P \left[ \left| TV^{a,b*}(P) \mathbb{E}_P \left[ (\xi_t^{a,b}(P))^2 \mid \mathcal{F}_{t-1} \right] - V^{a,b*}(P) \right| \right]$$

$$\leq \mathbb{E}_P \Bigg[ \mathbb{E}_P \Bigg[ \Bigg| \mathbb{E}_P \Bigg[ \frac{\left(Y_t^a - \widehat{\mu}_t^a(X_t)\right)^2}{\widehat{w}_t(a|X_t)} + \frac{\left(Y_t^b - \widehat{\mu}_t^b(X_t)\right)^2}{\widehat{w}_t(b|X_t)} + \left( \widehat{\mu}_t^a(X_t) - \widehat{\mu}_t^b(X_t) - \Delta^{a,b}(P) \right)^2$$

$$+ 2 \left( \mu^a(P)(X_t) - \mu^b(P)(X_t) - \widehat{\mu}_t^a(X_t) + \widehat{\mu}_t^b(X_t) \right) \left( \widehat{\mu}_t^a(X_t) - \widehat{\mu}_t^b(X_t) - \Delta^{a,b}(P) \right)$$

$$- \frac{\left(Y_t^a - \mu^a(P)(X_t)\right)^2}{w^*(a|X_t)} - \frac{\left(Y_t^b - \mu^b(P)(X_t)\right)^2}{w^*(b|X_t)} - \left( \mu^a(P)(X_t) - \mu^b(P)(X_t) - \Delta^{a,b}(P) \right)^2 \mid X_t, \mathcal{F}_{t-1} \Bigg] \Bigg| \mid \mathcal{F}_{t-1} \Bigg] \Bigg]$$

$$= \mathbb{E}_P \Bigg[ \Bigg| \mathbb{E}_P \Bigg[ \frac{\left(Y_t^a - \widehat{\mu}_t^a(X_t)\right)^2}{\widehat{w}_t(a|X_t)} + \frac{\left(Y_t^b - \widehat{\mu}_t^b(X_t)\right)^2}{\widehat{w}_t(b|X_t)} + \left( \widehat{\mu}_t^a(X_t) - \widehat{\mu}_t^b(X_t) - \Delta^{a,b}(P) \right)^2$$

$$+ 2 \left( \mu^a(P)(X_t) - \mu^b(P)(X_t) - \widehat{\mu}_t^a(X_t) + \widehat{\mu}_t^b(X_t) \right) \left( \widehat{\mu}_t^a(X_t) - \widehat{\mu}_t^b(X_t) - \Delta^{a,b}(P) \right)$$

$$- \frac{\left(Y_t^a - \mu^a(P)(X_t)\right)^2}{w^*(a|X_t)} - \frac{\left(Y_t^b - \mu^b(P)(X_t)\right)^2}{w^*(b|X_t)} - \left( \mu^a(P)(X_t) - \mu^b(P)(X_t) - \Delta^{a,b}(P) \right)^2 \mid X_t, \mathcal{F}_{t-1} \Bigg] \Bigg| \Bigg].$$

Because $\widehat{\mu}_t^a, \widehat{\mu}_t^b$ and $\widehat{w}_t$ are constructed from $\mathcal{F}_{t-1}$,

$$\mathbb{E}_P \left[ \left| TV^{a,b*}(P) \mathbb{E}_P \left[ (\xi_t^{a,b}(P))^2 \mid \mathcal{F}_{t-1} \right] - V^{a,b*}(P) \right| \right]$$

$$\leq \mathbb{E}_P \Bigg[ \Bigg| \mathbb{E}_P \Bigg[ \frac{\left(Y_t^a - \widehat{\mu}_t^a(X_t)\right)^2}{\widehat{w}_t(a|X_t)} + \frac{\left(Y_t^b - \widehat{\mu}_t^b(X_t)\right)^2}{\widehat{w}_t(b|X_t)} + \left( \widehat{\mu}_t^a(X_t) - \widehat{\mu}_t^b(X_t) - \Delta^{a,b}(P) \right)^2$$

$$+ 2 \left( \mu^a(P)(X_t) - \mu^b(P)(X_t) - \widehat{\mu}_t^a(X_t) + \widehat{\mu}_t^b(X_t) \right) \left( \widehat{\mu}_t^a(X_t) - \widehat{\mu}_t^b(X_t) - \Delta^{a,b}(P) \right)$$

$$- \frac{\left(Y_t^a - \mu^a(P)(X_t)\right)^2}{w^*(a|X_t)} - \frac{\left(Y_t^b - \mu^b(P)(X_t)\right)^2}{w^*(b|X_t)} - \left( \mu^a(P)(X_t) - \mu^b(P)(X_t) - \Delta^{a,b}(P) \right)^2 \mid X_t, \widehat{f}_{t-1}, \pi_t \Bigg] \Bigg| \Bigg].$$

From the results of Step 1, there exist $\widetilde{C}_4$ and $\widetilde{C}_5$ such that

$$\mathbb{E}_P \left[ \left| TV^{a,b*}(P) \mathbb{E}_P \left[ (\xi_t^{a,b}(P))^2 \mid \mathcal{F}_{t-1} \right] - V^{a,b*}(P) \right| \right]$$

$$\leq \mathbb{E}_P \left[ \left| \mathbb{E}_P \left[ \frac{\left(Y_t^a - \widehat{\mu}_t^a(X_t)\right)^2}{\widehat{w}_t(a|X_t)} + \frac{\left(Y_t^b - \widehat{\mu}_t^b(X_t)\right)^2}{\widehat{w}_t(b|X_t)} + \left(\widehat{\mu}_t^a(X_t) - \widehat{\mu}_t^b(X_t) - \Delta^{a,b}(P)\right)^2 \right. \right. \right.$$

$$+ 2 \left(\mu^a(P)(X_t) - \mu^b(P)(X_t) - \widehat{\mu}_t^a(X_t) + \widehat{\mu}_t^b(X_t)\right) \left(\widehat{\mu}_t^a(X_t) - \widehat{\mu}_t^b(X_t) - \Delta^{a,b}(P)\right) \Big\}$$

$$- \frac{\left(Y_t^a - \mu^a(P)(X_t)\right)^2}{w^*(a|X_t)} + \frac{\left(Y_t^b - \mu^b(P)(X_t)\right)^2}{w^*(b|X_t)} - \left(\mu^a(P)(X_t) - \mu^b(P)(X_t) - \Delta^{a,b}(P)\right)^2 \mid X_t, \widehat{f}_{t-1}, \pi_t \bigg] \bigg| \bigg]$$

$$\leq \widetilde{C}_4 \sum_{k=0}^1 \mathbb{E}_P \left[ \left| \sqrt{w^*(k|X_t)} - \sqrt{\widehat{w}_t(k|X_t)} \right| \right] + \widetilde{C}_5 \sum_{k=0}^1 \mathbb{E}_P \left[ \left| \widehat{\mu}_t^k(X_t) - \mu^k(P)(X_t) \right| \right].$$

Then, from $L^r$ convergence theorem, by using point convergence of $\widehat{\mu}_t^a$, $\widehat{\mu}_t^b$ and $\widehat{w}_t$, and the sub-Exponentiality of $(\xi_t^{a,b}(P))^2$, we have $\mathbb{E}_P \left[ \left| TV^{a,b*}(P)\mathbb{E}_P\left[(\xi_t^{a,b}(P))^2 \mid \mathcal{F}_{t-1}\right] - V^{a,b*}(P) \right| \right] \to 0$. Therefore,

$$\mathbb{P}\left( \left| \sum_{t=1}^T \mathbb{E}_P\left[(\xi_t^{a,b}(P))^2 \mid \mathcal{F}_{t-1}\right] - 1 \right| \geq \varepsilon \right)$$

$$\leq \frac{\frac{1}{TV^{a,b*}(P)} \sum_{t=1}^T \mathbb{E}_P \left[ \left| TV^{a,b*}(P)\mathbb{E}_P\left[(\xi_t^{a,b}(P))^2 \mid \mathcal{F}_{t-1}\right] - V^{a,b*}(P) \right| \right]}{\varepsilon} \to 0.$$

As a conclusion,

$$\sum_{t=1}^T \left((\xi_t^{a,b}(P))^2 - 1\right) = \sum_{t=1}^T \left((\xi_t^{a,b}(P))^2 - \mathbb{E}_P\left[(\xi_t^{a,b}(P))^2|\mathcal{F}_{t-1}\right] + \mathbb{E}_P\left[(\xi_t^{a,b}(P))^2|\mathcal{F}_{t-1}\right] - 1\right) \xrightarrow{\mathrm{P}} 0.$$

$\square$

# J. Direct Proof for the Upper Bound for the Probability of Misidentification of the AS-AIPW Strategy

For the AS-AIPW strategy, in the previous section, we discuss the asymptotic normality of the AS-AIPW strategy to obtain the worst-case upper bound for the expected simple regret. However, for the AS-AIPW estimator, we can directly derive the upper bound for the probability of misidentification more easily without going through the asymptotic normality, which requires some specific techniques of empirical process in the proof.

In this section, we derive the following upper bound for the probability of misidentification of the TS-HIR estimator

**Theorem J.1.** *Under Assumptions I.1, for any $P_0 \in \mathcal{P}^*$ and all $a, b$,*

$$- \liminf_{T \to \infty} \frac{1}{T} \log \mathbb{P}_0 \left( \widehat{\mu}_T^{\mathrm{AIPW},a^*(P)} \leq \widehat{\mu}_T^{\mathrm{AIPW},a} \right) \geq \frac{(\Delta^a(P))^2}{2V^{a*}(P)} - O\left((\Delta^a(P))^3\right).$$

## J.1. Proof of Theorem J.1

Now, we consider proving Theorem J.1. Let us define

$$\xi_t^a(P) = \frac{\varphi^{a^*(P)}\left(Y_t, A_t, X_t; \widehat{\mu}_t^{a^*(P)}, \widehat{w}_t\right) - \varphi^a\left(Y_t, A_t, X_t; \widehat{\mu}_t^a, \widehat{w}_t\right) - (\mu^a(P) - \mu^b(P))}{\sqrt{TV^{a*}(P)}},$$

$$V^{a*}(P) = V^{a^*(P),a*}(P)$$

The upper bound is derived from the Chernoff bound. Our proof is partially inspired by techniques in Hadad et al. (2021), and Kato et al. (2020).

**Step 1: the sequence $\{\xi_t^a(P)\}_{t=1}^T$ is an MDS.** We prove that $\{\xi_t^a(P)\}_{t=1}^T$ is an MDS. Although this fact is well-known in the literature of causal inference (van der Laan, 2008; Hadad et al., 2021; Kato et al., 2020), we show the proof for the sake of completeness.

**Lemma J.2.** *Under Assumptions I.1, for any $P_0 \in \mathcal{P}^*$, $\{\xi_t^a(P)\}_{t=1}^T$ is an MDS; that is,*

$$\mathbb{E}_P\left[\xi_t^a(P)|\mathcal{F}_{t-1}\right] = 0.$$

*Proof.* For each $t \in [T]$,

$$\mathbb{E}_P\left[\xi_t^a(P)|X_t, \mathcal{F}_{t-1}\right] = \frac{1}{\sqrt{T}V^{a*}(P)}\mathbb{E}_P\left[\varphi^{a^*(P)}\left(Y_t, A_t, X_t; \widehat{\mu}_t^{a^*(P)}, \widehat{w}_t\right) - \varphi^a\left(Y_t, A_t, X_t; \widehat{\mu}_t^a, \widehat{w}_t\right) - (\mu^{a^*(P)}(P) - \mu^a(P))\Big|X_t, \mathcal{F}_{t-1}\right]$$

$$= \frac{1}{\sqrt{T}V^{a*}(P)}\frac{\mathbb{E}_P[\mathbb{1}[A_t = a^*(P)]|X_t, \mathcal{F}_{t-1}]\mathbb{E}_P\left[Y_t^* - \widehat{\mu}_t^{a^*(P)}(X_t)|X_t, \mathcal{F}_{t-1}\right]}{\widehat{w}_t(a^*(P)|X_t)} + \widehat{\mu}_t^{a^*(P)}(X_t)$$

$$- \frac{\mathbb{E}_P[\mathbb{1}[A_t = a]|X_t, \mathcal{F}_{t-1}]\mathbb{E}_P\left[Y_t^a - \widehat{\mu}_t^a(X_t)|X_t, \mathcal{F}_{t-1}\right]}{\widehat{w}_t(a|X_t)} - \widehat{\mu}_t^a(X_t) - (\mu^{a^*(P)}(P) - \mu^a(P))$$

$$= \frac{1}{\sqrt{T}V^{a*}(P)}\left\{\left(\mu^{a^*(P)}(P)(X_t) - \mu^a(P)(X_t)\right) - \left(\mu^{a^*(P)}(P) - \mu^a(P)\right)\right\}.$$

Therefore,

$$\mathbb{E}_P\left[\xi_t^a(P)|\mathcal{F}_{t-1}\right] = \mathbb{E}_P\left[\mathbb{E}_P\left[\xi_t^a(P)|X_t, \mathcal{F}_{t-1}\right]|\mathcal{F}_{t-1}\right] = \frac{1}{\sqrt{T}V^{a*}(P)}\left\{\left(\mu^{a^*(P)}(P) - \mu^a(P)\right) - \left(\mu^{a^*(P)}(P) - \mu^a(P)\right)\right\} = 0.$$

$\square$

**Step 2: the Chernoff bound.** By applying the Chernoff bound, for any $v \geq 0$ and any $\lambda < 0$,

$$\mathbb{P}_0\left(\sum_{t=1}^T \xi_t^a(P) \leq v\right) \leq \mathbb{E}_P\left[\exp\left(\lambda\sum_{t=1}^T \xi_t^a(P)\right)\right]\exp\left(-\lambda v\right).$$

From the Chernoff bound and a property of an MDS, we have

$$\mathbb{E}_P\left[\exp\left(\lambda\sum_{t=1}^T \xi_t^a(P)\right)\right] = \mathbb{E}_P\left[\prod_{t=1}^T \mathbb{E}_P\left[\exp\left(\lambda\xi_t^a(P)\right)|\mathcal{F}_{t-1}\right]\right] = \mathbb{E}_P\left[\exp\left(\sum_{t=1}^T \log\mathbb{E}_P\left[\exp\left(\lambda\xi_t^a(P)\right)|\mathcal{F}_{t-1}\right]\right)\right] \tag{12}$$

By applying the Taylor series expansion around $\lambda = 0$,

$$\log\mathbb{E}_P\left[\exp\left(\lambda\xi_t^a(P)\right)|\mathcal{F}_{t-1}\right] = \frac{\lambda^2}{2}\mathbb{E}_P\left[(\xi_t^a(P))^2|\mathcal{F}_{t-1}\right] + O\left(\left(\lambda/\sqrt{T}\right)^3\right). \tag{13}$$

Here, $\mathbb{E}_P\left[\exp\left(\lambda\xi_t^a(P)\right)|\mathcal{F}_{t-1}\right] = 1+\sum_{k=1}^\infty (\lambda/\sqrt{T})^k\mathbb{E}_P\left[(\sqrt{T}\xi_t^a(P))^k/k!|\mathcal{F}_{t-1}\right]$. Because $\mathbb{E}_P\left[(\sqrt{T}\xi_t^a(P))^k/k!|\mathcal{F}_{t-1}\right]$ is bounded by a constant that is independent from $T$[7] for all $k \geq 1$, $\mathbb{E}_P\left[\exp\left(\lambda\xi_t^a(P)\right)|\mathcal{F}_{t-1}\right] = 1 + \sum_{k=1}^2 (\lambda/\sqrt{T})^k\mathbb{E}_P\left[(\sqrt{T}\xi_t^a(P))^k/k!|\mathcal{F}_{t-1}\right] + O\left(\left(\lambda/\sqrt{T}\right)^3\right)$. Note that the Taylor series expansion of $\log(1 + z)$ around $z = 0$ is given as $\log(1 + z) = z - z^2/2 + z^3/3 - \cdots$. Therefore,

$$\log\mathbb{E}_P\left[\exp\left(\lambda\xi_t^a(P)\right)|\mathcal{F}_{t-1}\right]$$

$$= \left\{\frac{\lambda}{\sqrt{T}}\mathbb{E}_P\left[\sqrt{T}\xi_t^a(P)|\mathcal{F}_{t-1}\right] + \frac{\lambda^2}{T}\mathbb{E}_P\left[(\sqrt{T}\xi_t^a(P))^2/2!|\mathcal{F}_{t-1}\right] + O\left(\left(\lambda/\sqrt{T}\right)^3\right)\right\}$$

---

[7]Note that $\sqrt{T}\xi_t^a(P) = \dfrac{\varphi^{a^*(P)}\left(Y_t, A_t, X_t; \widehat{\mu}_t^{a^*(P)}, \widehat{w}_t\right) - \varphi^a\left(Y_t, A_t, X_t; \widehat{\mu}_t^a, \widehat{w}_t\right) - (\mu^{a^*(P)}(P) - \mu^a(P))}{\sqrt{V^{a*}(P)}}$.

$$- \frac{1}{2} \left\{ \frac{\lambda}{\sqrt{T}} \mathbb{E}_P \left[ \sqrt{T} \xi_t^a(P) | \mathcal{F}_{t-1} \right] + O\left( \left( \lambda / \sqrt{T} \right)^2 \right) \right\}^2$$

$$= \frac{\lambda^2}{T} \mathbb{E}_P \left[ (\sqrt{T} \xi_t^a(P))^2 / 2! | \mathcal{F}_{t-1} \right] + O\left( \left( \lambda / \sqrt{T} \right)^3 \right).$$

Here, we used $\mathbb{E}_P \left[ \xi_t^a(P) | \mathcal{F}_{t-1} \right] = 0$. Thus, the (13) holds.

**Step 3: convergence of the second moment.** We next show that $T \mathbb{E}_P \left[ (\xi_t^a(P))^2 | \mathcal{F}_{t-1} \right] - 1 \xrightarrow{\text{a.s}} 0$.

**Lemma J.3.** *Under Assumptions I.1, for any $P_0 \in \mathcal{P}^*$,*

$$T \mathbb{E}_P \left[ (\xi_t^a(P))^2 | \mathcal{F}_{t-1} \right] - 1 \xrightarrow{\text{a.s}} 0 \qquad \text{as } t \to \infty.$$

Note that $T(\xi_t^a(P))^2$ does not depend on $T$. The proof is shown in Appendix K.

This lemma immediately yields the following lemma.

**Lemma J.4.** *Under Assumptions I.1, for any $P_0 \in \mathcal{P}^*$,*

$$\sum_{t=1}^{T} \mathbb{E}_P \left[ (\xi_t^a(P))^2 | \mathcal{F}_{t-1} \right] - 1 \xrightarrow{\text{P}} 0 \qquad \text{as } T \to \infty.$$

Our proof refers to the proof of Lemma 10 in Hadad et al. (2021).

*Proof.* Let $u_t$ be $u_t = T \mathbb{E}_P \left[ (\xi_t^a(P))^2 | \mathcal{F}_{t-1} \right] - 1$. Note that $\sum_{t=1}^{T} \mathbb{E}_P \left[ (\xi_t^a(P))^2 | \mathcal{F}_{t-1} \right] - 1 = \frac{1}{T} \sum_{t=1}^{T} u_t$.

From the proof of Lemma J.3, we can find that $u_t$ is a bounded random variable. Recall that

$$TV^{a*}(P) \mathbb{E}_P \left[ (\xi_t^a(P))^2 | \mathcal{F}_{t-1} \right] = \mathbb{E}_P \left[ \frac{(\sigma^*(X_t))^2 + (\mu^{a^*(P)}(P)(X_t) - \widehat{\mu}_t^{a^*(P)}(X_t))^2}{\widehat{w}_t(a^*(P)|X_t)} | \mathcal{F}_{t-1} \right]$$

$$+ \mathbb{E}_P \left[ \frac{(\sigma^a(X_t))^2 + (\mu^a(P)(X_t) - \widehat{\mu}_t^a(X_t))^2}{\widehat{w}_t(a|X_t)} | \mathcal{F}_{t-1} \right] - \mathbb{E}_P \left[ \left( \widehat{\mu}_t^{a^*(P)}(X_t) + \widehat{\mu}_t^a(X_t) - (\mu^{a^*(P)}(P) - \mu^a(P)) \right)^2 | \mathcal{F}_{t-1} \right].$$

We assumed that $(\mu^{a^*(P)}(P)(X_t), \mu^a(P)(X_t), \widehat{\mu}_t^{a^*(P)}(X_t), \widehat{\mu}_t^a(X_t), \widehat{w}_t(a^*(P)|X_t), \widehat{w}_t(a|X_t))$ are all bounded random variables. Let $C$ be a constant independent from $T$ such that $|u_t| < C$ for all $t \in \mathbb{N}$.

Fix some positive $\epsilon > 0$ and $\delta > 0$. Almost-sure convergence of $u_t$ to zero as $t \to \infty$ implies that we can find a large enough $t_\epsilon$ such that $|u_t| < \epsilon$ for all $t \geq t_\epsilon$ with probability at least $1 - \delta$. Let $\mathcal{E}(\epsilon)$ denote the event in which this happens; that is, $\mathcal{E}(\epsilon) = \{|u_t| < \epsilon \quad \forall t \geq t_\epsilon\}$. Under this event, for $T > t_\epsilon$,

$$\sum_{t=1}^{T} |u_t| \leq \sum_{t=1}^{t_\epsilon} C + \sum_{t=t_\epsilon+1}^{T} \epsilon = t_\epsilon C + T\epsilon.$$

Therefore,

$$\mathbb{P}\left( \frac{1}{T} \sum_{t=1}^{T} |u_t| > 2\epsilon \right) = \mathbb{P}\left( \left\{ \frac{1}{T} \sum_{t=1}^{T} |u_t| > 2\epsilon \right\} \cap \mathcal{E}(\epsilon) \right) + \mathbb{P}\left( \left\{ \frac{1}{T} \sum_{t=1}^{T} |u_t| > 2\epsilon \right\} \cap \mathcal{E}^c(\epsilon) \right)$$

$$\leq \mathbb{P}\left( \frac{t_\epsilon}{T} C + \epsilon > 2\epsilon \right) + \mathbb{P}\left( \mathcal{E}^c(\epsilon) \right)$$

$$= \mathbb{P}\left( \frac{t_\epsilon}{T} C > \epsilon \right) + \mathbb{P}\left( \mathcal{E}^c(\epsilon) \right).$$

Letting $T \to \infty$, for arbitrarily small $\delta > 0$, the statement follows. $\qquad \square$

Then, from the continuous mapping theorem, $\sum_{t=1}^{T} \mathbb{E}_P \left[ (\xi_t^a(P))^2 | \mathcal{F}_{t-1} \right] - 1 \xrightarrow{\text{P}} 0$ as $T \to \infty$ implies

$$\exp\left( \frac{\lambda^2}{2} \left\{ \sum_{t=1}^{T} \mathbb{E}_P \left[ (\xi_t^a(P))^2 | \mathcal{F}_{t-1} \right] - 1 \right\} \right) \xrightarrow{\text{P}} \exp(0) = 1 \qquad \text{as } T \to \infty.$$

Then, from $L^r$-convergence theorem, we obtain the following lemma.

**Lemma J.5.** *Under Assumptions I.1, for any $P_0 \in \mathcal{P}^*$, for any $\varepsilon > 0$, there exist $T_0 > 0$ such that for all $T > T_0$,*

$$\mathbb{E}_P \left[ \exp\left( \lambda \sum_{t=1}^{T} \xi_t^a(P) \right) \right] \exp(-\lambda v) \leq (1+\varepsilon) \exp\left( \frac{\lambda^2}{2} + O(\lambda^3/\sqrt{T}) - \lambda v \right).$$

*Proof.* Because $\mathbb{E}_P \left[ (\xi_t^a(P))^2 | \mathcal{F}_{t-1} \right]$ is bounded, $\exp\left( \frac{\lambda^2}{2} \left\{ \sum_{t=1}^{T} \mathbb{E}_P \left[ (\xi_t^a(P))^2 | \mathcal{F}_{t-1} \right] - 1 \right\} \right)$ is uniformly integrable (Proposition B.2 in Appendix B). Therefore, from $L^r$-convergence theorem (Proposition B.3 in Appendix B),

$$\mathbb{E}_P \left[ \exp\left( \frac{\lambda^2}{2} \left\{ \sum_{t=1}^{T} \mathbb{E}_P \left[ (\xi_t^a(P))^2 | \mathcal{F}_{t-1} \right] - 1 \right\} \right) \right] \to 1$$

From (12) and (13), for any $\varepsilon > 0$, there exist $T_0 > 0$ such that for all $T > T_0$,

$$\mathbb{E}_P \left[ \exp\left( \lambda \sum_{t=1}^{T} \xi_t^a(P) \right) \right] \exp\left( -\frac{\lambda^2}{2} \right)$$

$$= \mathbb{E}_P \left[ \exp\left( \frac{\lambda^2}{2} \left\{ \sum_{t=1}^{T} \mathbb{E}_P \left[ (\xi_t^a(P))^2 | \mathcal{F}_{t-1} \right] - 1 \right\} + O(\lambda^3/\sqrt{T}) \right) \right]$$

$$\leq (1+\varepsilon) \exp\left( O(\lambda^3/\sqrt{T}) \right)$$

The proof is complete. □

This lemma immediately yields the following lemma.

**Lemma J.6.** *Under Assumptions I.1, for any $P_0 \in \mathcal{P}^*$ and any $v, \varepsilon > 0$, there exist $T_0 > 0$ such that for all $T > T_0$,*

$$\mathbb{P}_0 \left( \sum_{t=1}^{T} \xi_t^a(P) \leq v \right) \leq (1+\varepsilon) \exp\left( -\frac{v^2}{2} + O(-v^3/\sqrt{T}) \right)$$

*Proof.* For any $v, \varepsilon > 0$, there exist $T_0 > 0$ such that for all $T > T_0$, from the Chernoff bound,

$$\mathbb{P}_0 \left( \sum_{t=1}^{T} \xi_t^a(P) \leq v \right) \leq (1+\varepsilon) \exp\left( \frac{\lambda^2}{2} + O(\lambda^3/\sqrt{T}) - \lambda v \right).$$

By substituting $\lambda = -u < 0$, the claim follows. □

Then, by substituting $u = -\frac{\sqrt{T}(\mu^{a^*(P)}(P) - \mu^a(P))}{\sqrt{V^{a*}(P)}} < 0$, we obtain

$$-\frac{1}{T} \log \mathbb{P}_0 \left( \sum_{t=1}^{T} \xi_t^a(P) \leq -\frac{\sqrt{T}(\mu^{a^*(P)}(P) - \mu^a(P))}{\sqrt{V^{a*}(P)}} \right)$$

$$\geq -\frac{1}{T} \log \left( \exp\left( -\frac{T(\mu^{a^*(P)}(P) - \mu^a(P))^2}{2V^{a*}(P)} + O\left( \frac{T(\mu^{a^*(P)}(P) - \mu^a(P))^3}{(V^{a*}(P))^{3/2}} \right) \right) \right) - \frac{1}{T} \log(1+\varepsilon).$$

Letting $T \to \infty$ and $\varepsilon \to 0$,

$$-\liminf_{T \to \infty} \frac{1}{T} \log \mathbb{P}_0 \left( \widehat{\mu}_T^{\text{AIPW}, a^*(P)} \leq \widehat{\mu}_T^{\text{AIPW}, a} \right) \geq \frac{(\mu^{a^*(P)}(P) - \mu^a(P))^2}{2V^{a*}(P)} - O\left( (\mu^{a^*(P)}(P) - \mu^a(P))^3 \right).$$

Thus, Theorem J.1 holds.

## K. Proof of Lemma J.3

*Proof.*

$$TV^{a*}(P)\mathbb{E}_P\left[(\xi_t^a(P))^2|\mathcal{F}_{t-1}\right] = \mathbb{E}_P\left[\left(\varphi^{a*(P)}\left(Y_t, A_t, X_t; \widehat{\mu}_t^{a*(P)}, \widehat{w}_t\right) - \varphi^a\left(Y_t, A_t, X_t; \widehat{\mu}_t^a, \widehat{w}_t\right) - (\mu^{a*(P)}(P) - \mu^a(P))\right)^2 \Big|\mathcal{F}_{t-1}\right]$$

$$= \mathbb{E}_P\left[\left(\frac{\mathbb{1}[A_t = a^*(P)]\left(Y_t^* - \widehat{\mu}_t^{a*(P)}(X_t)\right)}{\widehat{w}_t(a^*(P)|X_t)} - \frac{\mathbb{1}[A_t = a]\left(Y_t^a - \widehat{\mu}_t^a(X_t)\right)}{\widehat{w}_t(a|X_t)} + \widehat{\mu}_t^{a*(P)}(X_t) - \widehat{\mu}_t^a(X_t) - (\mu^{a*(P)}(P) - \mu^a(P))\right)^2 \Big|\mathcal{F}_{t-1}\right]$$

$$= \mathbb{E}_P\left[\left(\frac{\mathbb{1}[A_t = a^*(P)]\left(Y_t^* - \widehat{\mu}_t^{a*(P)}(X_t)\right)}{\widehat{w}_t(a^*(P)|X_t)} - \frac{\mathbb{1}[A_t = a]\left(Y_t^a - \widehat{\mu}_t^a(X_t)\right)}{\widehat{w}_t(a|X_t)}\right)^2\right.$$

$$+ 2\left(\frac{\mathbb{1}[A_t = a^*(P)]\left(Y_t^* - \widehat{\mu}_t^{a*(P)}(X_t)\right)}{\widehat{w}_t(a^*(P)|X_t)} - \frac{\mathbb{1}[A_t = a]\left(Y_t^a - \widehat{\mu}_t^a(X_t)\right)}{\widehat{w}_t(a|X_t)}\right)\left(\widehat{\mu}_t^{a*(P)}(X_t) - \widehat{\mu}_t^a(X_t) - (\mu^{a*(P)}(P) - \mu^a(P))\right)$$

$$\left.+ \left(\widehat{\mu}_t^{a*(P)}(X_t) - \widehat{\mu}_t^a(X_t) - (\mu^{a*(P)}(P) - \mu^a(P))\right)^2 |\mathcal{F}_{t-1}\right]$$

$$= \mathbb{E}_P\left[\frac{\mathbb{1}[A_t = a^*(P)]\left(Y_t^* - \widehat{\mu}_t^{a*(P)}(X_t)\right)^2}{\widehat{w}_t(a^*(P)|X_t)} + \frac{\mathbb{1}[A_t = a]\left(Y_t^a - \widehat{\mu}_t^a(X_t)\right)^2}{\widehat{w}_t(a|X_t)}\right.$$

$$+ 2\left(\frac{\mathbb{1}[A_t = a^*(P)]\left(Y_t^* - \widehat{\mu}_t^{a*(P)}(X_t)\right)}{\widehat{w}_t(a^*(P)|X_t)} - \frac{\mathbb{1}[A_t = a]\left(Y_t^a - \widehat{\mu}_t^a(X_t)\right)}{\widehat{w}_t(a|X_t)}\right)\left(\widehat{\mu}_t^{a*(P)}(X_t) - \widehat{\mu}_t^a(X_t) - (\mu^{a*(P)}(P) - \mu^a(P))\right)$$

$$\left.+ \left(\widehat{\mu}_t^{a*(P)}(X_t) - \widehat{\mu}_t^a(X_t) - (\mu^{a*(P)}(P) - \mu^a(P))\right)^2 |\mathcal{F}_{t-1}\right]$$

$$= \mathbb{E}_P\left[\frac{\left(Y_t^* - \widehat{\mu}_t^{a*(P)}(X_t)\right)^2}{\widehat{w}_t(a^*(P)|X_t)}|\mathcal{F}_{t-1}\right] + \mathbb{E}_P\left[\frac{\left(Y_t^a - \widehat{\mu}_t^a(X_t)\right)^2}{\widehat{w}_t(a|X_t)}|\mathcal{F}_{t-1}\right]$$

$$- \mathbb{E}_P\left[\left(\widehat{\mu}_t^{a*(P)}(X_t) + \widehat{\mu}_t^a(X_t) - (\mu^{a*(P)}(P) - \mu^a(P))\right)^2 |\mathcal{F}_{t-1}\right].$$

Here, we used

$$\mathbb{E}_P\left[\frac{\mathbb{1}[A_t = a]\left(Y_t^a - \widehat{\mu}_t^a(X_t)\right)^2}{(\widehat{w}_t(a|X_t))^2}|\mathcal{F}_{t-1}\right] = \mathbb{E}_P\left[\mathbb{E}_P\left[\frac{\widehat{w}_t(a|X_t)\left(Y_t^a - \widehat{\mu}_t^a(X_t)\right)^2}{(\widehat{w}_t(a|X_t))^2}|X_t\mathcal{F}_{t-1}\right]\right]$$

$$= \mathbb{E}_P\left[\frac{\left(Y_t^a - \widehat{\mu}_t^a(X_t)\right)^2}{\widehat{w}_t(a|X_t)}|\mathcal{F}_{t-1}\right]$$

and

$$\mathbb{E}_P\left[\frac{\mathbb{1}[A_t = a]\left(Y_t^a - \widehat{\mu}_t^a(X_t)\right)}{\widehat{w}_t(a|X_t)}\left(\widehat{\mu}_t^{a*(P)}(X_t) - \widehat{\mu}_t^a(X_t) - (\mu^{a*(P)}(P) - \mu^a(P))\right)|\mathcal{F}_{t-1}\right]$$

$$= \mathbb{E}_P\left[\left(\widehat{\mu}_t^{a*(P)}(X_t) - \widehat{\mu}_t^a(X_t) - (\mu^{a*(P)}(P) - \mu^a(P))\right)\mathbb{E}_P\left[\frac{\widehat{w}_t(a|X_t)\left(Y_t^a - \widehat{\mu}_t^a(X_t)\right)}{\widehat{w}_t(a|X_t)}|X_t, \mathcal{F}_{t-1}\right]|\mathcal{F}_{t-1}\right].$$

For $d \in \{a, a^*(P)\}$, we also have

$$\mathbb{E}_P\left[\frac{\left(Y_t^d - \widehat{\mu}_t^d(X_t)\right)^2}{\widehat{w}_t(d|X_t)}|X_t, \mathcal{F}_{t-1}\right] = \frac{\mathbb{E}_P[(Y_t^d)^2|X_t] - 2\mu^d(P)(X_t)\widehat{\mu}_t^d(X_t) + (\widehat{\mu}_t^d(X_t))^2}{\widehat{w}_t(d|X_t)}$$

$$= \frac{\mathbb{E}_P[(Y_t^d)^2|X_t] - (\mu^d(P)(X_t))^2 + (\mu^d(P)(X_t) - \widehat{\mu}_t^d(X_t))^2}{\widehat{w}_t(d|X_t)}$$

$$= \frac{\left(\sigma^d(X_t)\right)^2 + (\mu^d(P)(X_t) - \widehat{\mu}_t^d(X_t))^2}{\widehat{w}_t(d|X_t)},$$

where we used $\mathbb{E}_P[(Y_t^d)^2|x] - (\mu^d(P)(x))^2 = \left(\sigma^d(x)\right)^2$. Therefore,

$$\mathbb{E}_P\left[\frac{\left(Y_t^* - \widehat{\mu}_t^{a^*(P)}(X_t)\right)^2}{\widehat{w}_t(a^*(P)|X_t)}\Big|\mathcal{F}_{t-1}\right] + \mathbb{E}_P\left[\frac{\left(Y_t^a - \widehat{\mu}_t^a(X_t)\right)^2}{\widehat{w}_t(a|X_t)}\Big|\mathcal{F}_{t-1}\right]$$

$$- \mathbb{E}_P\left[\left(\widehat{\mu}_t^{a^*(P)}(X_t) + \widehat{\mu}_t^a(X_t) - (\mu^{a^*(P)}(P) - \mu^a(P))\right)^2\Big|\mathcal{F}_{t-1}\right]$$

$$= \mathbb{E}_P\left[\frac{(\sigma^*(X_t))^2 + (\mu^{a^*(P)}(P)(X_t) - \widehat{\mu}_t^{a^*(P)}(X_t))^2}{\widehat{w}_t(a^*(P)|X_t)}\right] + \mathbb{E}_P\left[\frac{(\sigma^a(X_t))^2 + (\mu^a(P)(X_t) - \widehat{\mu}_t^a(X_t))^2}{\widehat{w}_t(a|X_t)}\right]$$

$$- \mathbb{E}_P\left[\left(\widehat{\mu}_t^{a^*(P)}(X_t) + \widehat{\mu}_t^a(X_t) - (\mu^{a^*(P)}(P) - \mu^a(P))\right)^2\right].$$

Because $\widehat{\mu}_t^a(x) \xrightarrow{\text{a.s.}} \mu^a(P)(x)$ and $\widehat{w}_t(a|x) \xrightarrow{\text{a.s.}} w^*(a|x)$, for each $x \in \mathcal{X}$, with probability 1,

$$\lim_{t\to\infty}\left|\left(\frac{(\sigma^*(x))^2 + (\mu^{a^*(P)}(P)(x) - \widehat{\mu}_t^{a^*(P)}(x))^2}{\widehat{w}_t(a^*(P)|x)}\right)\right.$$

$$+ \left(\frac{(\sigma^a(x))^2 + (\mu^a(P)(x) - \widehat{\mu}_t^a(x))^2}{\widehat{w}_t(a|x)}\right) - \left(\widehat{\mu}_t^{a^*(P)}(x) + \widehat{\mu}_t^a(x) - (\mu^{a^*(P)}(P) - \mu^a(P))\right)^2$$

$$\left.- \left(\frac{(\sigma^*(x))^2}{w^*(a^*(P)|x)} + \frac{(\sigma^a(x))^2}{w^*(a|x)} + \left(\mu^{a^*(P)}(P)(x) - \mu^a(P)(x) - (\mu^{a^*(P)}(P) - \mu^a(P))\right)^2\right)\right|$$

$$\leq \lim_{t\to\infty}\left|\frac{(\sigma^*(x))^2}{\widehat{w}_t(a^*(P)|x)} - \frac{(\sigma^*(x))^2}{w^*(a^*(P)|x)}\right| + \lim_{t\to\infty}\left|\frac{(\sigma^a(x))^2}{\widehat{w}_t(a|x)} - \frac{(\sigma^a(X)^2)}{w^*(a|x)}\right|$$

$$+ \lim_{t\to\infty}\frac{(\mu^{a^*(P)}(P)(x) - \widehat{\mu}_t^{a^*(P)}(x))^2}{\widehat{w}_t(a^*(P)|x)} + \lim_{t\to\infty}\frac{(\mu^a(P)(x) - \widehat{\mu}_t^a(x))^2}{\widehat{w}_t(a|x)}$$

$$+ \lim_{t\to\infty}\left|\left(\widehat{\mu}_t^{a^*(P)}(x) + \widehat{\mu}_t^a(x) - (\mu^{a^*(P)}(P) - \mu^a(P))\right)^2 - \left(\mu^{a^*(P)}(P)(x) - \mu^a(P)(x) - (\mu^{a^*(P)}(P) - \mu^a(P))\right)^2\right|$$

$$= 0.$$

Therefore, from Lebesgue's dominated convergence theorem,

$$TV^{a*}(P)\mathbb{E}_P\left[(\xi_t^a(P))^2|\mathcal{F}_{t-1}\right] - V^{a*}(P)$$

$$= \mathbb{E}_P\left[\frac{(\sigma^*(x))^2 + (\mu^{a^*(P)}(P)(X_t) - \widehat{\mu}_t^{a^*(P)}(X_t))^2}{\widehat{w}_t(a^*(P)|X_t)}\Big|\mathcal{F}_{t-1}\right]$$

$$+ \mathbb{E}_P\left[\frac{(\sigma^a(x))^2 + (\mu^a(P)(X_t) - \widehat{\mu}_t^a(X_t))^2}{\widehat{w}_t(a|X_t)}\Big|\mathcal{F}_{t-1}\right]$$

$$- \mathbb{E}_P\left[\left(\widehat{\mu}_t^{a^*(P)}(X_t) + \widehat{\mu}_t^a(X_t) - (\mu^{a^*(P)}(P) - \mu^a(P))\right)^2\Big|\mathcal{F}_{t-1}\right]$$

$$- \mathbb{E}_P\left[\frac{(\sigma^*(X_t))^2}{w^*(a^*(P)|X_t)} + \frac{(\sigma^a(X_t))^2}{w^*(a|X_t)} + \left(\mu^{a^*(P)}(P)(X_t) - \mu^a(P)(X_t) - (\mu^{a^*(P)}(P) - \mu^a(P))\right)^2\Big|\mathcal{F}_{t-1}\right]$$

$$\xrightarrow{\text{a.s.}} 0.$$

$\square$

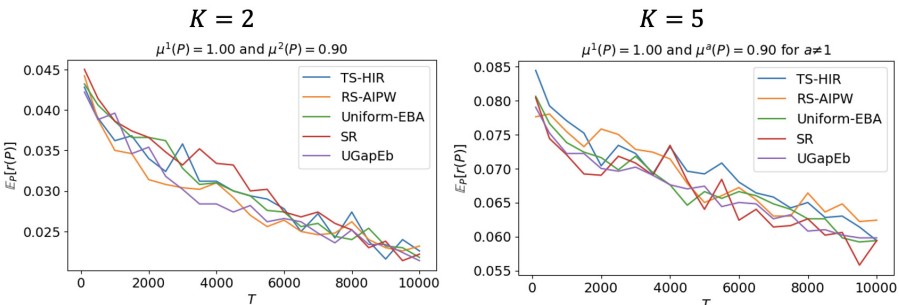

Figure 2: Experimental results. The $y$-axis and $x$-axis denote the expected simple regret $\mathbb{E}_P[r_T(\pi)(P)]$ under each strategy and $T$, respectively.

## L. Non-asymptotic Upper Bound

When a specific convergence rate is assumed, a non-asymptotic convergence rate for the CLT can be derived for the AS-AIPW strategy.

**Corollary L.1.** *For all $a, b \in [K]^2$, suppose that $\xi_t^{a,b}(P)$ is conditionally sub-Gaussian; that is, there exits an absolute constant $C_\xi > 0$ such that for all $P \in \mathcal{P}$ and all $\lambda \in \mathbb{R}$,*

$$\mathbb{E}_P\left[\exp\left(\lambda \xi_t^{a,b}(P)\right) | \mathcal{F}_{t-1}\right] \leq \exp\left(\frac{\lambda^2 C_\xi}{2}\right).$$

*Also suppose that some $\alpha > 0$ and constants $M$, $C$ and $D$,*

$$\max_{t \in \mathbb{N}} \mathbb{E}_P\left[\exp\left(\left|\sqrt{T}\xi_t^{a,b}(P)\right|^\alpha\right)\right] < M,$$

*and*

$$\mathbb{P}\left(|\Omega_t^{a,b}(P) - 1| > D/\sqrt{t}(\log t)^{2+2/\alpha}\right) \leq Ct^{-1/4}(\log t)^{1+1/\alpha}.$$

*Then, for $a, b \in [K]$ and $T \geq 2$,*

$$\mathbb{P}\left(\widehat{\mu}_T^{a,\text{AIPW}} - \widehat{\mu}_T^{b,\text{AIPW}} \leq 0\right) \leq \begin{cases} \exp\left(-\frac{T\left(\Delta^{a,b}(P)\right)^2}{V^{a,b*}(P)}\right) + AT^{-1/4}(\log T)^{1+1/\alpha} & \text{if } E_0 < \sqrt{T}\Delta^{a,b}(P) \leq E; \\ \exp\left(-\frac{T\left(\Delta^{a,b}(P)\right)^2}{2C_\xi^2}\right) & \text{if } E_0 < \sqrt{T}\Delta^{a,b}(P), \end{cases}$$

*where the constant $A$ depends only on $\alpha$, $M$, $C$, and $D$, and $E_0 > E > 0$ are some constants independent from $T$ and $\Delta^{a,b}(P)$.*

## M. Additional Experimental Results

We show addition experimental results. In Appendix M.1, we show results with variances different from those in Section 7. In Appendix M.2, we show the result with continuous contextual information.

### M.1. Addition Experimental Results without Contextual Information

Under the same setting with that in Section 7, we draw the variances from a uniform distribution with support $[10, 100]$. We show the result in Figure 2.

### M.2. Continuous Contextual Information

We compare our TS-HIR and AS-AIPW strategies with the Uniform-EBA (Uniform, Bubeck et al., 2011), and Successive Rejection (SR, Audibert et al., 2010), and UGapEb (Gabillon et al., 2012).

We consider cases with $K = 2, 3, 5, 10$ and $D = 2$-dimensional contextual information. We consider contextual information; therefore, we only investigate the AS-AIPW strategy. Because we cannot obtain a closed-form solution for $K \geq 3$, for simplicity, we fix $w^*(a|x) = \frac{(\sigma^a(x))^2}{\sum_{b \in [K]} (\sigma^b(x))^2}$ for $K \geq 3$, which still reduces the expected simple regret better than $w^*(a) = \frac{(\sigma^a)^2}{\sum_{b \in [K]} (\sigma^b)^2}$.

In each set up, the best treatment arm is arm $1$. The expected outcomes of suboptimal treatment arms are equivalent and denoted by $\widetilde{\mu} = \mu^2(P) = \mu^K(P)$. We use $\widetilde{\mu} = 0.80, 0.90$. We generate the variance from a uniform distribution with a support $[0.1, 5]$ and contextual information $X_t = (X_{t1}, X_{t2})$ from a multinomial distribution with mean $(1, 1)$ and variance $\begin{pmatrix} 1 & 0.1 \\ 0.1 & 1 \end{pmatrix}$. Let $(\theta_1, \theta_2)$ be random variables generated from a uniform distribution with a support $[0, 1]$. We then generate $\mu^a(P)(X_t) = \theta_1 X_{t1}^2 + \theta_2 X_{t2}^2 / c_\mu^a$ and $(\sigma^a(X_t))^2 = (\theta_1 X_{t1}^2 + \theta_2 X_{t2}^2) / c_\sigma^a$, where $c_\mu^a, c_\sigma^a$ are values that adjust the expectation to align with $\mu^a(P)$ and $(\sigma^a)^2$. We continue the experiments until $T = 5,000$ when $\widetilde{\mu} = 0.80$ and $T = 10,000$ when $\widetilde{\mu} = 0.90$. We conduct $100$ independent trials for each setting. At each $t \in [T]$, we plot the empirical simple regret in Figure 1. Additional results are presented in Appendix M.

From Figure 1 and Appendix M, we can observe that the AS-AIPW performs well when $K = 2$. When $K \geq 3$, although the AS-AIPW tends to outperform the Uniform, other strategies also perform well. We conjecture that the AS-AIPW exhibits superiority against other methods when $K$ is small (mismatching term in the upper bound), the gap between the best and suboptimal arms is small, and the variances significantly vary across arms. As the superiority depends on the situation, we recommend a practitioner to use the AS-AIPW with several strategies in a hybrid way.

We show experimental results with $K = 2, 3, 5, 10$ in Figures 4–6, respectively.

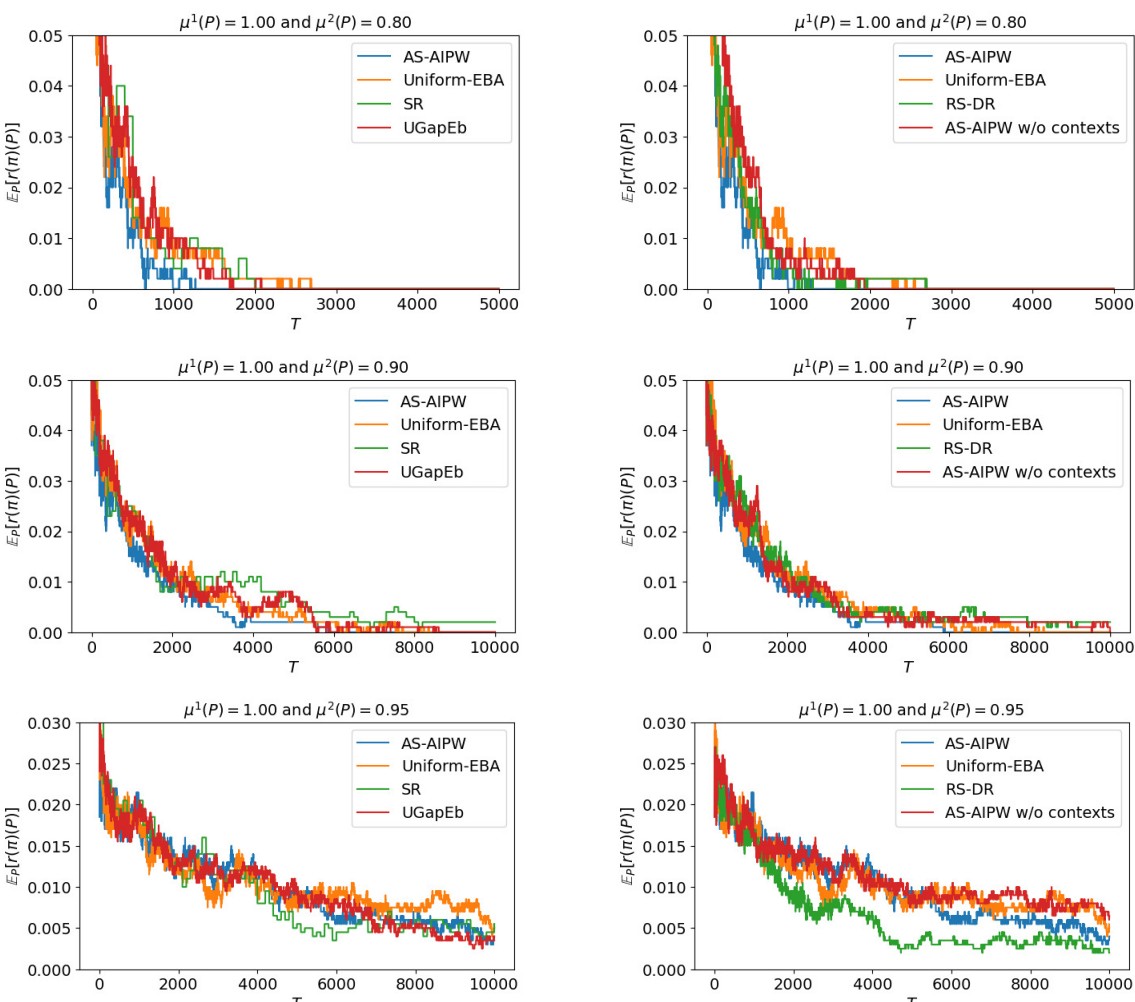

Figure 3: Results when $K = 2$.

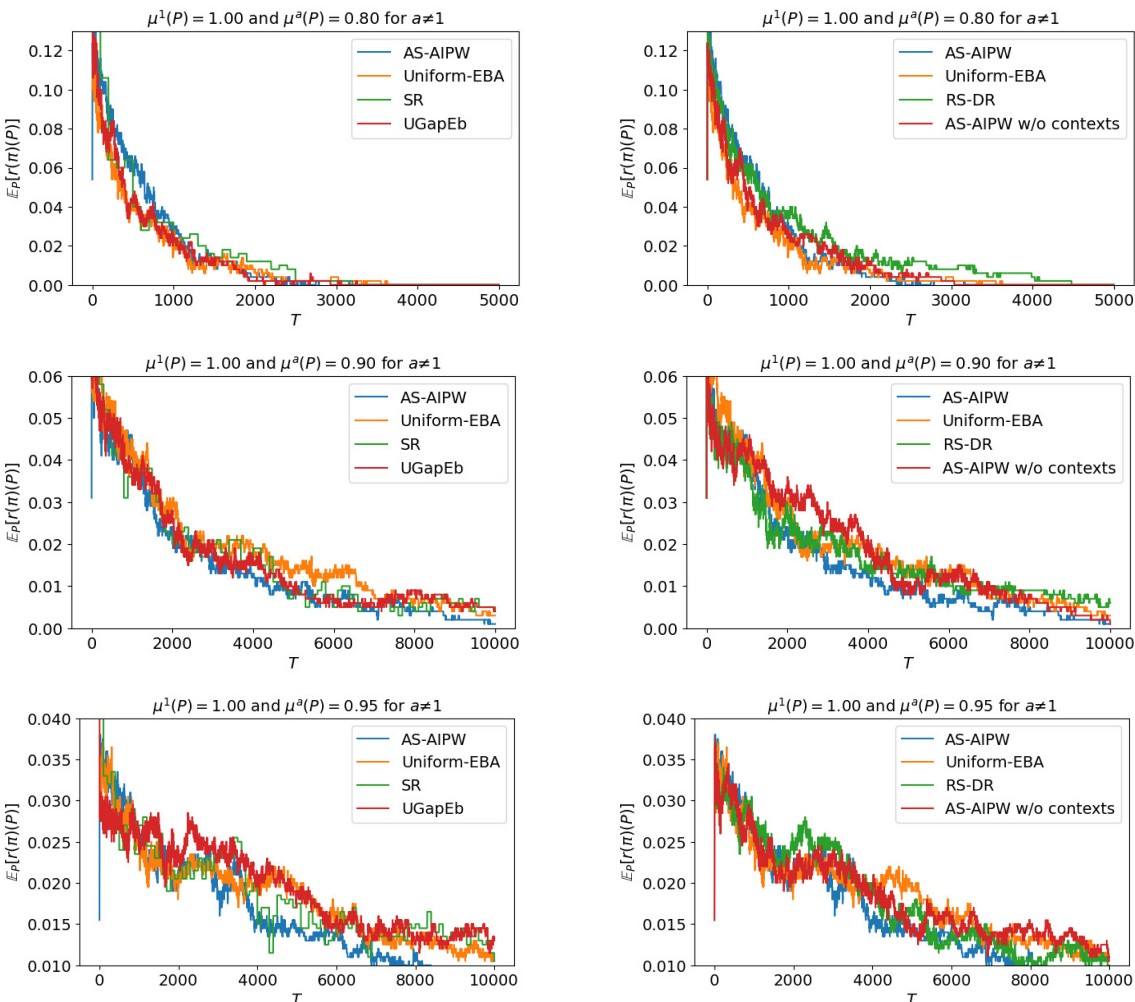

Figure 4: Results when $K = 3$.

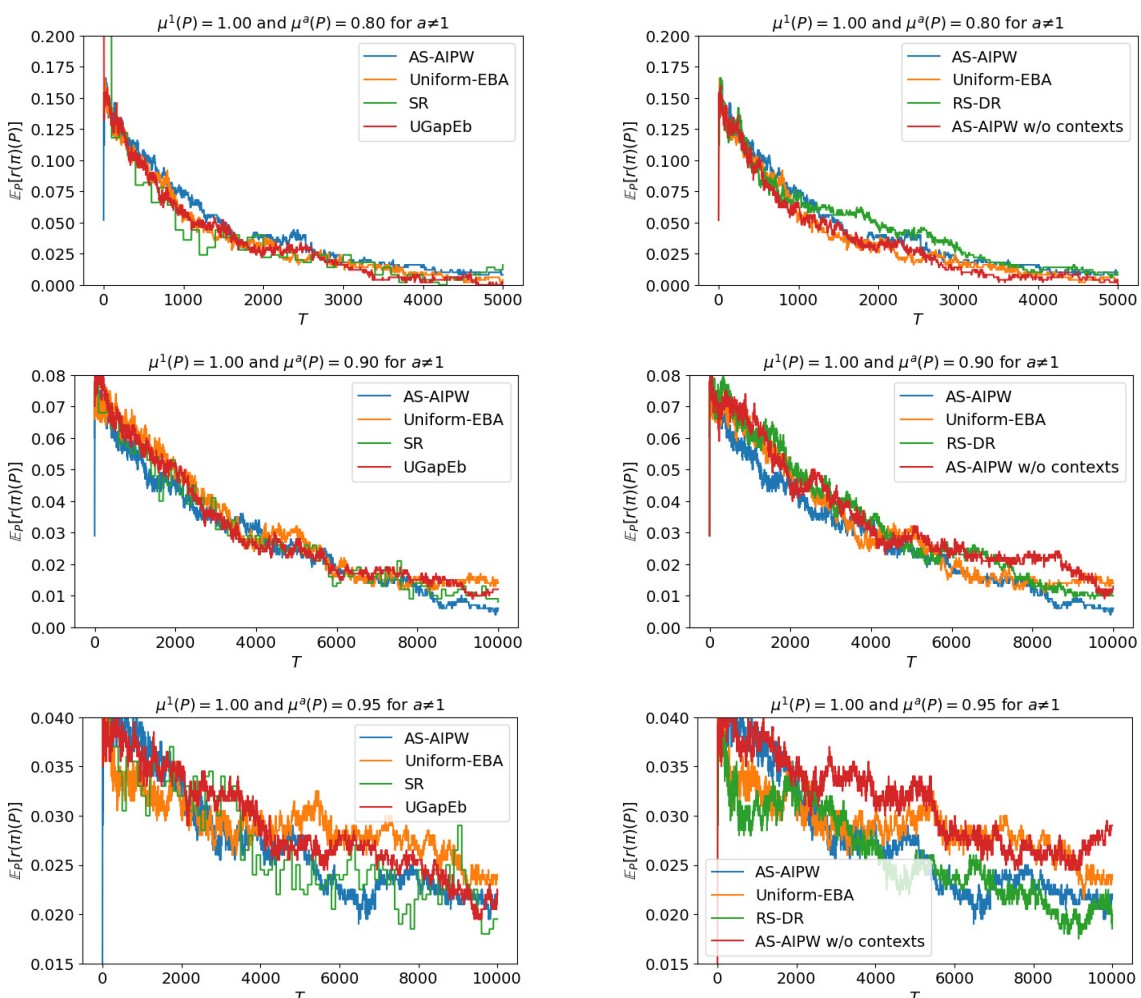

Figure 5: Results when $K = 5$.

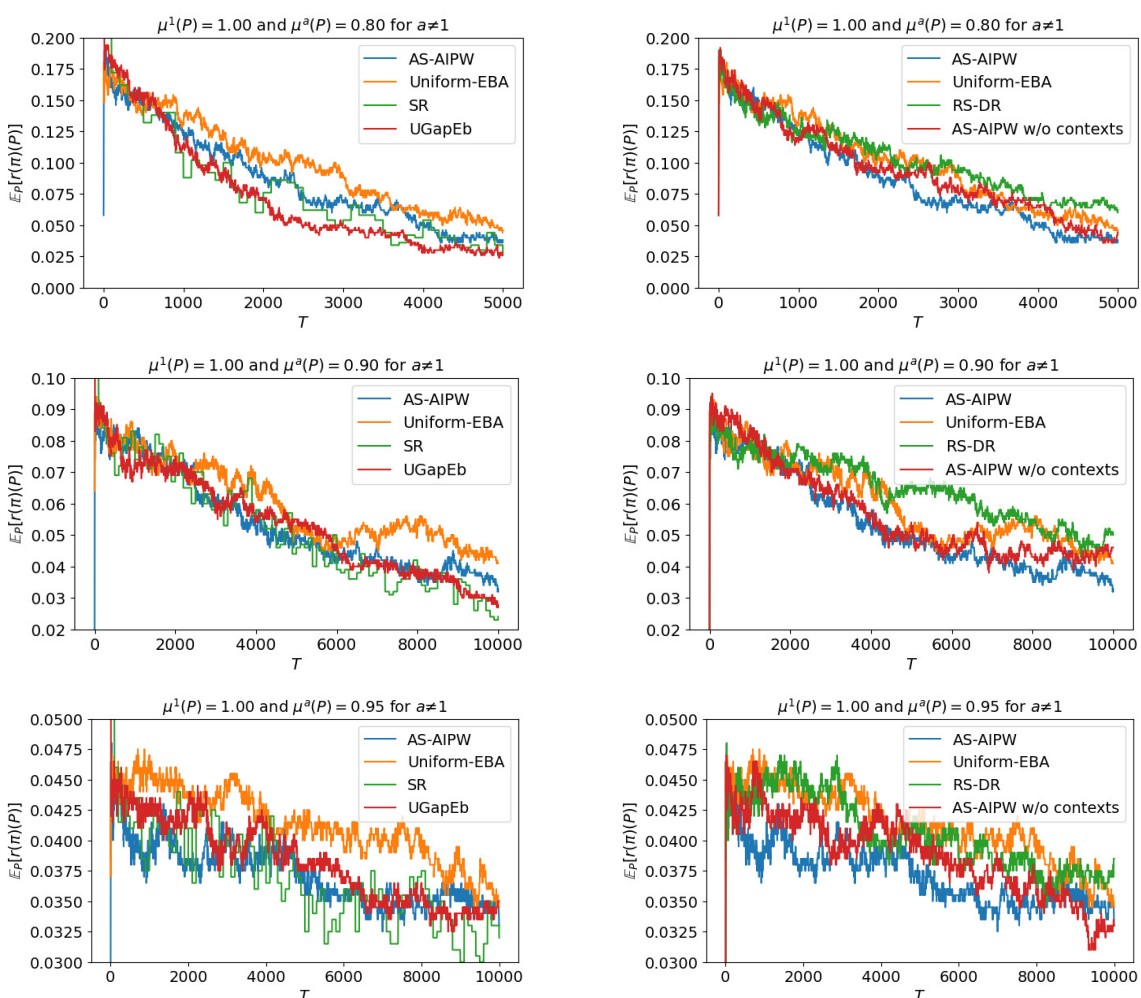

Figure 6: Results when $K = 10$.