# OpenReview forum: "Asymptotically Optimal Fixed-Budget Best Arm Identification with Variance-Dependent Bounds"
_ICML.cc/2023/Workshop/ILHF — ILHF Workshop ICML 2023_

### Official Review · Reviewer_97ZG · 2023-06-09
**Review for Submission 45**

**Rating:** 6
**Confidence:** 3

**Review:**

## Summary
This paper studied the "fixed-budge best arm identification" problem, where for the first T rounds, a decision maker will repeatedly observe contextual information, take an arm and observe its outcome, and return an recommendation rule for arm selection. The authors established lower bound and proposed a TS-HIR algorithm. Besides, they provided upper bounds for the algorithm and showed its asymptotic minimax optimality.

## Strength

The problem studied by this paper is important and realistic I believed. The theoretical contribution is clear.

## Weakness

I did not find the appendix, so can not check the correctness of the proof.

This paper seems not very related to the topic of this workshop. I can not see which part of the setting can be interpreted as the human feedback.


### Others
There are some typos of notations and etc.
e.g.
line 083-084 Y^K_a
line 114: b\in[b]
line 115: \Delta^a should be \Delta^b?
and etc…

---

### Official Review · Reviewer_j9Ld · 2023-06-16

**Rating:** 6
**Confidence:** 4

**Review:**

This paper studies the asymptotically optimality problem for the BAI and obtained the variance-dependence bounds for both the lower bound and the upper bound. This is an interesting endeavor, and I like it. Meanwhile, there are some existing works that studies something similar in the reinforcement learning setting for the policy evaluation purpose such as [1],[2]. it would nice to incorporate those discussions to understand how novel the obtained variance-structure are. Of course, I admit the current BAI (learning) problem is slightly different from the evaluation problem.

[1] Asymptotically Efficient Off-policy Evaluation for Tabular Reinforcement Learning, AISTATS20 (upper bound)

[2] Doubly Robust Off-policy Value Evaluation for Reinforcement Learning, ICML16 (lower bound)


Some small:

1. Y^K_a in line 83 should be Y_t^K;

2. no explanation for why the worst-case simple regret is approximated in the formula displayed in line 163;

3. def of sigma^a in line 199 not found;

4. should explain more why the definition 3.3 is a rich enough for constructing the lower bound?

3.4 seems to be a worst-case lower bound. Is it possible to have some instance-dependent lower bound?

---

### Decision · Program_Chairs · 2023-06-20

Accept